# Riemannian Projection-free Online Learning

**Zihao Hu**[†]**, Guanghui Wang**[†]**, Jacob Abernethy**[†,⋆]
College of Computing, Georgia Institute of Technology[†]
Google Research[⋆]
{zihaohu,gwang369}@gatech.edu, abernethyj@google.com

## Abstract

The projection operation is a critical component in a wide range of optimization algorithms, such as online gradient descent (OGD), for enforcing constraints and achieving optimal regret bounds. However, it suffers from computational complexity limitations in high-dimensional settings or when dealing with ill-conditioned constraint sets. *Projection-free* algorithms address this issue by replacing the projection oracle with more efficient optimization subroutines. But to date, these methods have been developed primarily in the *Euclidean setting*, and while there has been growing interest in optimization on *Riemannian manifolds*, there has been essentially no work in trying to utilize projection-free tools here. An apparent issue is that non-trivial affine functions are generally non-convex in such domains. In this paper, we present methods for obtaining sub-linear regret guarantees in *online geodesically convex optimization* on curved spaces for two scenarios: when we have access to (a) a *separation oracle* or (b) a *linear optimization oracle*. For geodesically convex losses, and when a separation oracle is available, our algorithms achieve $O(T^{1/2})$, $O(T^{3/4})$ and $O(T^{1/2})$ adaptive regret guarantees in the full information setting, the bandit setting with one-point feedback and the bandit setting with two-point feedback, respectively. When a linear optimization oracle is available, we obtain regret rates of $O(T^{3/4})$ for geodesically convex losses and $O(T^{2/3} \log T)$ for strongly geodesically convex losses.

## 1 Introduction

Online convex optimization (OCO) offers a framework for modeling sequential decision-making problems (Hazan et al., 2016). The standard setting depicts the learning process as a zero-sum game between a learner and an adversary. At round $t$, the learner selects a decision $\mathbf{x}_t$ from a convex set $\mathcal{K}$ and observes the encountered convex loss function $f_t$. The learner's goal is to minimize *regret*, defined as

$$\text{Regret}_T := \sum_{t=1}^{T} f_t(\mathbf{x}_t) - \min_{\mathbf{x} \in \mathcal{K}} \sum_{t=1}^{T} f_t(\mathbf{x}).$$

In Euclidean space, OCO boasts a robust theoretical foundation and numerous real-world applications, such as online load balancing (Molinaro, 2017), optimal control (Li et al., 2019), revenue maximization (Lin et al., 2019), and portfolio management (Jézéquel et al., 2022). The standard approach for OCO is online gradient descent (OGD), which performs

$$\mathbf{x}_{t+1} = \Pi_{\mathcal{K}}(\mathbf{x}_t - \eta_t \nabla f_t(\mathbf{x}_t)),$$

where $\Pi_{\mathcal{K}}$ represents the orthogonal projection onto $\mathcal{K}$, ensuring the sequence $\{\mathbf{x}_t\}_{t=1}^{T}$ remains feasible. However, the projection operation can be computationally expensive in high-dimensional or complex feasible sets. Projection-free online learning provides a reasonable way to handle this situation. At the heart of this approach is the understanding that, in many cases, the complexity of

37th Conference on Neural Information Processing Systems (NeurIPS 2023).

Table 1: Summary of main results. Our approach allows us to invoke either a Separation Oracle (SO) or a Linear Optimization Oracle (LOO) for $O(T)$ times throughout the $T$ rounds. Notably, our results match those presented by Garber & Kretzu (2022).

| Oracle | Losses | Feedback | Measure | Regret |
|--------|--------|----------|---------|--------|
| SO | gsc-convex | full information | adaptive regret | $O(T^{1/2})$, Thm. 1 |
| | gsc-convex | bandit, one-point | expected adaptive regret | $O(T^{3/4})$, Thm. 2 |
| | gsc-convex | bandit, two-point | expected adaptive regret | $O(T^{1/2})$, Thm. 3 |
| LOO | gsc-convex | full information | adaptive regret | $O(T^{3/4})$, Thm. 4 |
| | strongly gsc-convex | full information | regret | $O(T^{2/3} \log T)$, Thm. 5 |

implementing an optimization oracle can be significantly lower than that of the orthogonal projection. As a result, both practical and theoretical interests lie in replacing the projection operation with these optimization oracles. The two most well-known projection-free optimization oracles are:

$$\text{(Linear optimization oracle) } \underset{\mathbf{x} \in \mathcal{K}}{\operatorname{argmin}} \langle \mathbf{g}, \mathbf{x} \rangle$$

and

$$\text{(Separation oracle) } \mathbf{g} : \ \langle \mathbf{y} - \mathbf{x}, \mathbf{g} \rangle > 0, \quad \forall \mathbf{x} \in \mathcal{K}.$$

Whereas much attention has been given to the development of sub-linear regret guarantees using optimization oracles in Euclidean space (Hazan & Kale, 2012; Levy & Krause, 2019; Hazan & Minasyan, 2020; Wan et al., 2022; Garber & Kretzu, 2022; Mhammedi, 2022a; Wan et al., 2023), there is considerably less research on projection-free optimization on *Riemannian manifolds*. There are numerous scenarios where the space of feasible parameters is not a convex subset of Euclidean space, but instead has a manifold structure with a Riemannian metric; examples include non-negative PCA (Montanari & Richard, 2015), $K$-means clustering (Carson et al., 2017) and computing Wasserstein-Barycenters (Weber & Sra, 2022b). However, there has been significantly less research on efficiently computing projections or projection-free optimization in the Riemannian setting, even for highly-structured geodesically convex (gsc-convex) feasible sets like:

$$\mathcal{K} = \{\mathbf{x} | \max_{1 \le i \le m} h_i(\mathbf{x})\}$$

where each $h_i(\mathbf{x})$ is gsc-convex. In this paper we focus on projection-free optimization on Riemannian manifolds in the online setting. A Riemannian version of Online Gradient Descent (R-OGD) was given by Wang et al. (2021):

$$\mathbf{x}_{t+1} = \Pi_{\mathcal{K}} \mathrm{Exp}_{\mathbf{x}_t} \left( -\eta_t \nabla f_t(\mathbf{x}_t) \right),$$

where $\mathcal{K}$ is now a gsc-convex set, and $\Pi_{\mathcal{K}}$ is the metric projection onto $\mathcal{K}$. This method achieves sub-linear regret guarantees for Riemannian OCO. The metric projection is often the most expensive operation, and it is not always clear how to implement it on a manifold. In the present work, we propose the use of two alternative operations: a separation oracle (SO) or a linear optimization oracle (LOO) on the Riemannian manifold, with definitions deferred to Section 3. It is important to note that, in Euclidean space, both oracles rely on the definition of a hyperplane. In the realm of Riemannian manifolds, there are two natural extensions of a hyperplane: one is the sub-level set of a Busemann function, known as a horosphere (Bridson & Haefliger, 2013); the other relies on the inverse exponential map, as outlined in (2). While a horosphere is gsc-convex, the corresponding separation oracle exists if every boundary point of the feasible gsc-convex set has a locally supporting horosphere (Borisenko, 2002). The existence of a separation oracle is assured for any gsc-convex sets on Hadamard manifolds (Silva Louzeiro et al., 2022), if defined via the inverse exponential map. Hence, we adopt this latter definition. It is worth noting that this object is typically non-convex (Kristály et al., 2016), leading to geometric complexities that necessitate careful management. Also, our work builds upon the findings of Garber & Kretzu (2022) and inherits the adaptive regret guarantee for gsc-convex losses, as defined by Hazan & Seshadhri (2009):

$$\text{Adaptive Regret}_T := \sup_{[s,e] \subseteq [T]} \left\{ \sum_{t=s}^e f_t(\mathbf{x}_t) - \min_{\mathbf{x} \in \mathcal{K}} \sum_{t=s}^e f_t(\mathbf{x}) \right\}.$$

Our main contributions are summarized in Table 1. More specifically,

- Given a separation oracle, we attain adaptive regret bounds of $O(T^{1/2})$, $O(T^{3/4})$ and $O(T^{1/2})$ for gsc-convex losses in the full information setting, the bandit convex optimization setting with one-point feedback[1], and the bandit convex optimization setting with two-point feedback, respectively.

- Assuming access to a linear optimization oracle, we provide algorithms that enjoy $O(T^{3/4})$ adaptive regret for gsc-convex losses and $O(T^{2/3} \log T)$ regret for strongly gsc-convex losses.

- We highlight some key differences between convex sets on a Hadamard manifold and in Euclidean space. In particular, shrinking a gsc-convex set towards an interior point does not preserve convexity, and the Minkowski functional on a Hadamard manifold is non-convex. These results, which may not be well-known within the machine learning community, could be of independent interest.

The technical challenges of this paper can be divided into two parts.

Firstly, for the separation oracle, we need to bound the "thickness" of part of the feasible set cut by the separating hyperplane. While in Euclidean space, this can be achieved through a convex combination argument (Garber & Kretzu, 2022), the task becomes challenging on manifolds due to the varying metric at different points and the non-convex nature of the separating hyperplane. Fortunately, this problem can be addressed using the Jacobi field comparison technique. Also, unlike in Euclidean space, one cannot directly construct a separation oracle for $(1 - \delta)\mathcal{K}$ using a separation oracle for $\mathcal{K}$. This poses significant challenges in the bandit setting. Nonetheless, we have identified a novel solution to this issue in the two-point feedback setting.

Secondly, in Euclidean space, the linear optimization oracle is typically invoked by online Frank-Wolfe (OFW) (Hazan & Kale, 2012; Kretzu & Garber, 2021) to achieve no-regret online learning. The analysis of OFW relies on the fact that the Hessian of a linear function is zero everywhere. But on Hadamard manifolds, such functions' existence implies that the manifold has zero sectional curvature everywhere (Kristály et al., 2016). The algorithms in Garber & Kretzu (2022) do not require affinity and serve as a starting point for our results. However, the analysis still needs to be conducted carefully due to the non-convexity of the separating hyperplane on manifolds.

## 2 Related Work

In this section, we briefly review previous work on projection-free online learning in Euclidean space as well as online and projection-free optimization on Riemannian manifolds.

### 2.1 Projection-free OCO in Euclidean Space

**Linear Optimization Oracle.** The pioneering work of Hazan & Kale (2012) first introduced an online variant of the Frank-Wolfe algorithm (OFW) and achieved $O(T^{3/4})$ regret for convex functions. Hazan & Minasyan (2020) proposed a randomized algorithm that leverages smoothness to achieve $O(T^{2/3})$ expected regret. The insightful analysis of Wan & Zhang (2021) and Kretzu & Garber (2021) demonstrated that OFW indeed attains $O(T^{2/3})$ regret for strongly convex functions. Garber & Kretzu (2022) showed that it is possible to achieve $O(T^{3/4})$ adaptive regret and $O(T^{2/3} \log T)$ regret for convex and strongly convex functions, respectively. Mhammedi (2022b) illustrated how to obtain $\tilde{O}(T^{2/3})$ regret for convex functions, where $\tilde{O}(\cdot)$ hides logarithmic terms. Our results are in line with those of Garber & Kretzu (2022) and inherit the adaptive regret guarantee.

**Separation Oracle and Membership Oracle.** Levy & Krause (2019) demonstrated that it is possible to achieve $O(\sqrt{T})$ and $O(\log T)$ regret bounds for convex and strongly convex functions when the feasible set is a sublevel set of a smooth and convex function. Mhammedi (2022a) generalized the idea of Levy & Krause (2019) and showed how to obtain $O(\sqrt{T})$ and $O(\log T)$ regret guarantees for general convex sets. Garber & Kretzu (2022) provided algorithms that ensure $O(\sqrt{T})$ and $O(T^{3/4})$ adaptive regret guarantees for convex losses in the full information and bandit settings, respectively.

---

[1]For the one-point feedback setting, we allow the point at which we play and the point where we receive feedback to differ, thereby bypassing a fundamental challenge posed by Riemannian geometry. In the case of two-point feedback, however, we eliminate this non-standard setting.

## 2.2 Online and Projection-free Optimization on Manifolds

**Online Optimization on Manifolds.** Bécigneul & Ganea (2019) demonstrated that a series of adaptive optimization algorithms can be implemented on a product of Riemannian manifolds, with each factor manifold being assigned a learning rate. Antonakopoulos et al. (2020) proposed using Follow the Regularized Leader with a strongly gsc-convex regularizer to achieve $O(\sqrt{T})$ regret when the loss satisfies Riemannian Lipschitzness. Wang et al. (2021) introduced Riemannian OGD (R-OGD) and showed regret guarantees in full information and bandit convex optimization settings. Hu et al. (2023) considered achieving optimistic and dynamic regret on Riemannian manifolds.

**Projection-free Optimization on Manifolds.** Rusciano (2018) provided a non-constructive cutting hyperplane method on Hadamard manifolds. By comparison, our algorithms are constructive and deterministic. Weber & Sra (2022b) proposed Riemannian Frank-Wolfe (RFW) for gsc-convex optimization and showed some practical applications on the manifold of SPD matrices. In a subsequent work, Weber & Sra (2022a) generalized RFW to the stochastic and non-convex setting. We use RFW as a subroutine to invoke the linear optimization oracle and establish sub-linear regret guarantees. Hirai et al. (2023) implemented the interior point method on Riemannian manifolds and used a self-concordant barrier to enforce the constraint.

## 3 Preliminaries and Assumptions

In this section, we lay the groundwork for our study by presenting an overview of Riemannian manifolds, the separation oracle and the linear optimization oracle within these spaces. Additionally, we establish key definitions and assumptions that will be integral to the following sections.

**Riemannian Manifolds.** We provide key notations in Riemannian geometry that will be employed throughout this paper. Readers looking for a more comprehensive treatment are encouraged to consult Petersen (2006); Lee (2018). Our proof also relies on the concept of the *Jacobi field*, and we provide some backgrounds in Appendix E.1. We consider an $n$-dimensional smooth manifold $\mathcal{M}$ equipped with a Riemannian metric $g$. This metric confers a point-wise inner product $\langle \mathbf{u}, \mathbf{v} \rangle_{\mathbf{x}}$ at every point $\mathbf{x} \in \mathcal{M}$, where $\mathbf{u}, \mathbf{v}$ are vectors in the tangent space $T_{\mathbf{x}}\mathcal{M}$ at $\mathbf{x}$. This tangent space, a vector space of dimension $n$, encompasses all vectors tangent to $\mathbf{x}$. The Riemannian metric also determines the norm of a tangent vector $\mathbf{u}$ as: $\|\mathbf{u}\| := \sqrt{\langle \mathbf{u}, \mathbf{u} \rangle}$. A geodesic $\gamma(t) : [0, c] \to \mathcal{M}$ is a piecewise smooth curve with a constant velocity that locally minimizes the distance between its endpoints, say, $\mathbf{x}$ and $\mathbf{y}$. The Riemannian distance between these two points is given by $d(\mathbf{x}, \mathbf{y}) := \int_0^c \|\dot{\gamma}(t)\| dt = \int_0^c \|\dot{\gamma}(0)\| dt$. It's important to note that the Riemannian distance remains invariant under reparameterizations of $\gamma(t)$. Consider a geodesic $\gamma(t) : [0, 1] \to \mathcal{M}$ with $\gamma(0) = \mathbf{x}$, $\gamma(1) = \mathbf{y}$ and $\dot{\gamma}(0) = \mathbf{v}$. The exponential map $\text{Exp}_{\mathbf{x}}(\mathbf{v})$ transforms $\mathbf{v} \in T_{\mathbf{x}}\mathcal{M}$ to $\mathbf{y} \in \mathcal{M}$, and the inverse exponential map $\text{Exp}_{\mathbf{x}}^{-1}\mathbf{y}$ performs the inverse operation, mapping $\mathbf{y} \in \mathcal{M}$ to $\mathbf{v} \in T_{\mathbf{x}}\mathcal{M}$. The inverse exponential map also offers a handy way to express the Riemannian distance: $d(\mathbf{x}, \mathbf{y}) = \|\text{Exp}_{\mathbf{x}}^{-1}\mathbf{y}\|$.

The sectional curvature at a point $\mathbf{x} \in \mathcal{M}$ is contingent on two-dimensional subspaces of $T_{\mathbf{x}}\mathcal{M}$, and describes the curvature near $\mathbf{x}$. Generally, geodesics diverge on manifolds with negative sectional curvature, converge on manifolds with positive sectional curvature, and manifolds with zero sectional curvature are locally isometric to Euclidean space. In line with Zhang & Sra (2016); Wang et al. (2021), we primarily explore Hadamard manifolds, which are simply connected manifolds with non-positive curvature that admit a unique global minimizing geodesic between any pair of points. A set $\mathcal{K} \subseteq \mathcal{M}$ is geodesically convex (gsc-convex) if it includes the geodesic connecting $\mathbf{x}$ and $\mathbf{y}$ for any $\mathbf{x}, \mathbf{y} \in \mathcal{K}$. A $\mu$-strongly gsc-convex (or gsc-convex, when $\mu = 0$) function $f : \mathcal{M} \to \mathbb{R}$ fulfills

$$f(\mathbf{y}) \geq f(\mathbf{x}) + \left\langle \nabla f(\mathbf{x}), \text{Exp}_{\mathbf{x}}^{-1}\mathbf{y} \right\rangle + \frac{\mu}{2} d(\mathbf{x}, \mathbf{y})^2, \tag{1}$$

for any $\mathbf{x}, \mathbf{y} \in \mathcal{M}$, where $\nabla f(\mathbf{x}) \in T_{\mathbf{x}}\mathcal{M}$ is the Riemannian gradient.

**Optimization Oracles on Riemannian Manifolds.** In this part, we introduce the concept of a separation oracle and a linear optimization oracle on Riemannian manifolds.

A separation oracle, given a point $\mathbf{y}$ not in the gsc-convex set $\mathcal{K}$, returns a non-convex separating hyperplane that satisfies the following condition:

$$\left\langle -\text{Exp}_{\mathbf{y}}^{-1}\mathbf{x}, \mathbf{g} \right\rangle > 0, \quad \forall \mathbf{x} \in \mathcal{K}, \tag{2}$$

where $\mathbf{g} \in T_{\mathbf{y}}\mathcal{M}$. Even with the non-convexity of the separating hyperplane, for certain gsc-convex sets like $\mathcal{K} = \{\mathbf{x}| \max_{1 \leq i \leq m} h_i(\mathbf{x}) \leq 0\}$ where each $h_i(\mathbf{x})$ is gsc-convex, a separation oracle can be efficiently implemented by Lemma 18 and Remark 4.[2]

On the other hand, a linear optimization oracle is responsible for solving the following problem:

$$\operatorname*{argmin}_{\mathbf{x} \in \mathcal{K}} \left\langle \mathbf{g}, \operatorname{Exp}_{\mathbf{x}_0}^{-1} \mathbf{x} \right\rangle$$

on a gsc-convex set $\mathcal{K}$, where $\mathbf{x}_0 \in \mathcal{K}$ and $\mathbf{g} \in T_{\mathbf{x}_0}\mathcal{M}$. Although this objective is not gsc-convex, it can still be solved in closed form for certain problems. Examples include computing the geometric mean and the Bures-Wasserstein barycenter on the manifold of SPD matrices (Weber & Sra, 2022b).

In this paper, we rely on a series of definitions and assumptions, which we introduce here for clarity and reference in subsequent sections.

**Assumption 1.** *The manifold $\mathcal{M}$ is Hadamard with sectional curvature bounded below by $\kappa$, so the sectional curvature of $\mathcal{M}$ lies in the interval $[\kappa, 0]$.*

**Assumption 2.** *The manifold $\mathcal{M}$ is a homogeneous Hadamard manifold with sectional curvature bounded below by $\kappa$. For every $t \in [T]$, the inequality $|f_t(\mathbf{x})| \leq M$ holds. For the bandit setting with two-point feedback, we additionally require $\mathcal{M}$ to be symmetric.*

**Assumption 3.** *The set $\mathcal{K} \subseteq \mathcal{M}$ is a gsc-convex decision set and satisfies $\mathbb{B}_{\mathbf{p}}(r) \subseteq \mathcal{K} \subseteq \mathbb{B}_{\mathbf{p}}(R)$. Here, $\mathbb{B}_{\mathbf{p}}(r)$ represents the geodesic ball centered at $\mathbf{p} \in \mathcal{K}$ with radius $r$.*

**Assumption 4.** *For every $t \in [T]$ and $\mathbf{x} \in \mathbb{B}_{\mathbf{p}}(R)$, the function $f_t(\mathbf{x})$ is gsc-convex (or strongly gsc-convex), and the norm of its gradient is bounded by $G$, i.e., $\|\nabla f_t(\mathbf{x})\| \leq G$.*

Let us make two important comments. First, the homogeneity and the symmetry of $\mathcal{M}$ allows us to employ the unbiased estimator presented in Wang et al. (2023) for the bandit setting. It should be noted that homogeneous and symmetric Hadamard manifolds include two of the most commonly used ones: the hyperbolic space and the manifold of SPD matrices. Second, the projection onto a geodesic ball, denoted as $\Pi_{\mathbb{B}_{\mathbf{p}}(R)}(\cdot)$, is considered projection-free as it can be computed to an $\epsilon$ precision using $\log(1/\epsilon)$ bisections [3].

**Definition 1.** *We define a geometric constant $\zeta$ as $\zeta := 2R\sqrt{-\kappa} \coth(2R\sqrt{-\kappa})$.*

**Definition 2.** *Fixing $\mathbf{p} \in \mathcal{K}$, for any $c \in (0, \infty)$, we define $c\mathcal{K} = \{\operatorname{Exp}_{\mathbf{p}}(c\operatorname{Exp}_{\mathbf{p}}^{-1}\mathbf{x})|\mathbf{x} \in \mathcal{K}\}$.*

**Definition 3.** *We call $\tilde{\mathbf{y}} \in \mathcal{M}$ an infeasible projection of $\mathbf{y} \in \mathcal{M}$ onto a simply connected closed set $\tilde{\mathcal{K}} \subseteq \mathcal{M}$ if for every point $\mathbf{z} \in \tilde{\mathcal{K}}$, the inequality $d(\tilde{\mathbf{y}}, \mathbf{z}) \leq d(\mathbf{y}, \mathbf{z})$ holds. We define $\mathcal{O}_{IP}(\tilde{\mathcal{K}}, \mathbf{y})$ as an infeasible oracle for $\tilde{\mathcal{K}}$ which, given any $\mathbf{y}$, returns an infeasible projection of $\mathbf{y}$ onto $\tilde{\mathcal{K}}$.*

We note that, in Euclidean space where the sectional curvature is zero everywhere, we have $\zeta = \lim_{\kappa \to 0} 2R\sqrt{-\kappa} \coth(2R\sqrt{-\kappa}) = 1$. We also observe that the definition of the infeasible projection in Garber & Kretzu (2022) requires $\tilde{\mathcal{K}}$ to be convex, which is indeed unnecessary. This distinction is essential because, in the case of the separation-oracle-based OCO, we construct an infeasible projection oracle onto $\tilde{\mathcal{K}} = (1 - \delta)\mathcal{K}$, which may be non-convex (Theorem 6).

## 4   Warm-up: the High-level Idea

We briefly illustrate the overarching strategy of achieving regret guarantees. Our basic algorithm is Algorithm 1, which generates a sequence $\{\mathbf{y}_t\}_{t=1}^{T}$ by R-OGD that does not necessarily fall within the feasible set. A key insight is that we can build an infeasible projection oracle using either a separation oracle or a linear optimization oracle, resulting in a sequence $\{\tilde{\mathbf{y}}_t\}_{t=1}^{T}$ that exhibits a desirable regret guarantee (as shown in Lemma 1). The design of the infeasible projection oracle rests on a straightforward fact: whenever $\mathbf{y}_t$ deviates significantly from $\mathcal{K}$, we can call upon either oracle to produce a descent direction and then apply Lemma 2 to gauge the progress. Additional error terms, arising from the fact that $\tilde{\mathbf{y}}_t$ may not necessarily lie in $\mathcal{K}$, can be quantified by leveraging the boundedness of the gradient in Assumption 4.

---

[2]This fact is well-known in Euclidean space (Grötschel et al., 2012), but it appears we are the first to observe its counterpart on manifolds.

[3]A comprehensive explanation of this observation can be found in Remark 5.

**Algorithm 1:** Infeasible Riemannian OGD

---

**Data:** horizon $T$, feasible set $\tilde{\mathcal{K}}$, step-sizes $\{\eta_t\}_{t=1}^T$, infeasible projection oracle $\mathcal{O}_{IP}(\tilde{\mathcal{K}}, \cdot)$.

**for** $t = 1, \ldots, T$ **do**

    Play $\tilde{\mathbf{y}}_t$, and observe $f_t(\tilde{\mathbf{y}}_t)$

    Update $\mathbf{y}_{t+1} = \mathrm{Exp}_{\tilde{\mathbf{y}}_t}(-\eta_t \nabla f_t(\tilde{\mathbf{y}}_t))$, and set $\tilde{\mathbf{y}}_{t+1} = \mathcal{O}_{IP}(\tilde{\mathcal{K}}, \mathbf{y}_{t+1})$

**end**

---

We have the following guarantee for Algorithm 1.

**Lemma 1.** *(Proof in Appendix A.1) Assume $\tilde{\mathbf{y}}_t \in \mathbb{B}_{\mathbf{p}}(R)$ and let $\nabla_t = \nabla f_t(\tilde{\mathbf{y}}_t)$ and $\mathcal{K} \subseteq \mathbb{B}_{\mathbf{p}}(R)$ be a gsc-convex subset of $\mathcal{M}$. Consider $\tilde{\mathcal{K}} \subseteq \mathcal{K}$ as a simply connected and compact set, and $\mathcal{O}_{IP}(\tilde{\mathcal{K}}, \cdot)$ be an infeasible projection oracle as in Definition 3.*

*1) Suppose all losses are gsc-convex on $\mathbb{B}_{\mathbf{p}}(R)$. Fix some $\eta > 0$ and let $\eta_t = \eta$ for all $t \geq 1$. Algorithm 1 guarantees that the adaptive regret is upper-bounded by:*

$$\forall I = [s, e] \subseteq [T] : \sum_{t=s}^{e} f_t(\tilde{\mathbf{y}}_t) - \min_{\mathbf{x}_I \in \tilde{\mathcal{K}}} \sum_{t=s}^{e} f_t(\mathbf{x}_I) \leq \frac{d(\tilde{\mathbf{y}}_s, \mathbf{x}_I)^2}{2\eta} + \frac{\eta \zeta \sum_{t=s}^{e} \|\nabla_t\|^2}{2}.$$

*2) Suppose all losses are $\alpha$-strongly gsc-convex on $\mathbb{B}_{\mathbf{p}}(R)$ for some $\alpha > 0$. Let $\eta_t = \frac{1}{\alpha t}$ for all $t \geq 1$. Algorithm 1 guarantees that the static regret is upper bounded by:*

$$\sum_{t=1}^{T} f_t(\tilde{\mathbf{y}}_t) - \min_{\mathbf{x} \in \tilde{\mathcal{K}}} \sum_{t=1}^{T} f_t(\mathbf{x}) \leq \sum_{t=1}^{T} \frac{\zeta \|\nabla_t\|^2}{2\alpha t}.$$

**Remark 1.** *To apply Lemma 1, we need to ensure that $\tilde{\mathbf{y}}_t \in \mathbb{B}_{\mathbf{p}}(R)$ for any $t \in [T]$. In the case of a separation oracle, we have $\tilde{\mathcal{K}} = (1 - \delta)\mathcal{K}$ for some $\delta \in (0, 1)$, and $\tilde{\mathbf{y}}_t \in \mathcal{K} \subseteq \mathbb{B}_{\mathbf{p}}(R)$ by Lemma 4. With a linear optimization oracle, we have $\tilde{\mathcal{K}} = \mathcal{K}$, and $\tilde{\mathbf{y}}_t \in \mathbb{B}_{\mathbf{p}}(R)$ is guaranteed by Lemma 6.*

**Lemma 2.** *(Proof in Appendix A.2) Consider $\tilde{\mathcal{K}} \subseteq \mathcal{K} \subseteq \mathbb{B}_{\mathbf{p}}(R)$ as a simply connected and compact subset of $\mathcal{M}$. If $\mathbf{y} \notin \tilde{\mathcal{K}}$ and $\mathbf{g} \in T_{\mathbf{y}}\mathcal{M}$ satisfies $-\left\langle \mathrm{Exp}_{\mathbf{y}}^{-1}\mathbf{z}, \mathbf{g} \right\rangle \geq Q$, where $Q > 0$, then consider $\tilde{\mathbf{y}} = \mathrm{Exp}_{\mathbf{y}}(-\gamma \mathbf{g})$. For $\gamma = \frac{Q}{\zeta C^2}$ and $\|\mathbf{g}\| \leq C$, assume $d(\mathbf{y}, \mathbf{z}) \leq 2R$, then we have*

$$d(\tilde{\mathbf{y}}, \mathbf{z})^2 \leq d(\mathbf{y}, \mathbf{z})^2 - \frac{Q^2}{\zeta C^2}.$$

Unlike Garber & Kretzu (2022), in Lemma 2, we also do not require $\tilde{\mathcal{K}}$ to be gsc-convex.

## 5 Riemannian OCO with a Separation Oracle

In this section, we show how to use a separation oracle to construct an infeasible projection oracle and achieve sublinear regret guarantees. We note that we rely on an infeasible projection oracle onto $(1 - \delta)\mathcal{K}$ rather than directly onto $\mathcal{K}$. While we have a separation oracle that results in $\left\langle -\mathrm{Exp}_{\mathbf{y}}^{-1}\mathbf{x}, \mathbf{g} \right\rangle > 0$, using Lemma 2 on this separating hyperplane may lead to minuscule progress, given that $Q$ can be arbitrarily small. Consequently, achieving sublinear regret with only $O(T)$ oracle calls becomes unfeasible. In contrast, constructing an infeasible projection onto $(1 - \delta)\mathcal{K}$ always ensures meaningful progress, as quantified by Lemmas 3 and 4.

---

**Algorithm 2:** Infeasible Projection onto $(1 - \delta)\mathcal{K}$ with a Riemannian Separation Oracle

---

**Data:** feasible gsc-convex set $\mathcal{K}$, radius $r$, squeeze parameter $\delta$, initial point $\mathbf{y}_0$.

$\mathbf{y}_1 = \Pi_{\mathbb{B}_{\mathbf{p}}(R)} \mathbf{y}_0$

**for** $i = 1, \ldots$ **do**

    **if** $\mathbf{y}_i \notin \mathcal{K}$ **then**

        $\mathrm{SO}_{\mathcal{K}}$ returns $\mathbf{g}_i$ satisfying $\left\langle -\mathrm{Exp}_{\mathbf{y}_i}^{-1}\mathbf{x}, \mathbf{g}_i \right\rangle > 0$

        $\mathbf{y}_{i+1} = \mathrm{Exp}_{\mathbf{y}_i}(-\gamma_i \mathbf{g}_i)$ where $\gamma_i = \frac{\delta \bar{r}}{\|\mathbf{g}\|}$ and $\bar{r}$ is defined in Equation (3)

    **else**

        **return** $\mathbf{y} = \mathbf{y}_i$

    **end**

**end**

---

**Lemma 3.** *(Proof in Appendix B.1) Let* $\mathbf{y} \in \mathbb{B}_{\mathbf{p}}(R) \setminus \mathcal{K}$ *and let* $\mathbf{g} \in T_{\mathbf{y}}\mathcal{M}$ *be the output of the separation oracle for* $\mathbf{y}$. *Then, under Assumptions 1 and 3, we have that* $\langle -\mathrm{Exp}_{\mathbf{y}}^{-1}\mathbf{z}, \mathbf{g} \rangle > \delta\bar{r}\|\mathbf{g}\|$ *for any* $\mathbf{z} \in (1-\delta)\mathcal{K}$, *where*

$$\bar{r} := \frac{\sqrt{-\kappa}(2R+r)}{\sinh(\sqrt{-\kappa}(2R+r))} \cdot \frac{\sqrt{-\kappa}(R+r)}{\sinh(\sqrt{-\kappa}(R+r))} \cdot r. \tag{3}$$

In Euclidean space, we can establish that $\langle \mathbf{y} - \mathbf{z}, \mathbf{g} \rangle > \delta r\|\mathbf{g}\|$(Garber & Kretzu, 2022, Lemma 11). However, as indicated in Lemma 3, the result on manifolds is significantly worse with respect to $R$, given the exponential nature of $\sinh$. It is an interesting line of inquiry to explore whether this dependence is unavoidable.[4]

Based on Lemma 3, to implement an infeasible projection oracle onto $(1-\delta)\mathcal{K}$, the number of calls to the separation oracle is bounded in Lemma 4.

**Lemma 4.** *(Proof in Appendix B.2) Under Assumptions 1 and 3. Let* $0 < \delta < 1$ *and set* $\gamma_i = \frac{\delta\bar{r}}{\|\mathbf{g}\|}$. *Algorithm 2 executes at most* $\frac{\zeta(d(\mathbf{y}_0,(1-\delta)\mathcal{K})^2 - d(\mathbf{y},(1-\delta)\mathcal{K})^2)}{\delta^2\bar{r}^2} + 1$ *iterations and returns* $\mathbf{y} \in \mathcal{K}$ *such that* $d(\mathbf{y}, \mathbf{z})^2 \leq d(\mathbf{y}_0, \mathbf{z})^2$ *holds for any* $\mathbf{z} \in (1-\delta)\mathcal{K}$.

In the full information setting, with a separation oracle, infeasible R-OGD is shown in Algorithm 3.

---

**Algorithm 3:** Infeasible R-OGD with a separation oracle

---

**Data:** feasible gsc-convex set $\mathcal{K}$, radius $r$, step-size $\eta$ and squeeze parameter $\delta$.
$\tilde{\mathbf{y}}_1 = \mathbf{p} \in (1-\delta)\mathcal{K}$
**for** $t = 1, \ldots, T$ **do**
    Play $\tilde{\mathbf{y}}_t$ and receive $f_t(\tilde{\mathbf{y}}_t)$
    $\mathbf{y}_{t+1} = \mathrm{Exp}_{\tilde{\mathbf{y}}_t}(-\eta\nabla f_t(\tilde{\mathbf{y}}_t))$
    Update $\tilde{\mathbf{y}}_{t+1}$ as the output of Algorithm 2 with set $\mathcal{K}$, radius $r$, squeeze parameter $\delta$ and
    initial point $\mathbf{y}_{t+1}$.
**end**

---

We can show the following regret guarantee for Algorithm 3.

**Theorem 1.** *(Proof in Appendix B.3) Under Assumptions 1, 3 and 4. Set* $\eta = \frac{2R}{G\sqrt{\zeta T}}$ *and* $\delta = \frac{1}{2\sqrt{T}}$, *then the regret of Algorithm 3 is upper bounded by*

$$\sup_{[s,e]\subseteq[T]} \left\{ \sum_{t=s}^{e} f_t(\tilde{\mathbf{y}}_t) - \min_{\mathbf{x}_I \in \mathcal{K}} \sum_{t=s}^{e} f_t(\mathbf{x}_I) \right\} \leq \tfrac{5}{2} GR\sqrt{\zeta T},$$

*and the number of calls to the separation oracle is* $O(T)$.

Moving on, we demonstrate how to achieve a sublinear regret guarantee in the bandit convex optimization setting. A major challenge is that, while in Euclidean space, we can construct a separation oracle on $(1-\delta)\mathcal{K}$ using the separation oracle on $\mathcal{K}$ (Garber & Kretzu, 2022, Lemma 11.). On Hadamard manifolds, $(1-\delta)\mathcal{K}$ can even be non-convex (Theorem 6), thus a separation oracle for $(1-\delta)\mathcal{K}$ may not exist. For Riemannian BCO with one-point feedback, in Algorithm 4, we address this by resorting to a non-standard setting: we play $\tilde{\mathbf{z}}_t \in \mathcal{K}$ but we receive feedback at $\mathbf{z}_t$ where $\mathbf{z}_t$, $\tilde{\mathbf{z}}_t$ are nearby points. We present the algorithm and the corresponding regret guarantee in Algorithm 4 and Theorem 2.

---

**Algorithm 4:** One-point bandit convex optimization on manifolds with a separation oracle

---

**Data:** feasible gsc-convex set $\mathcal{K}$, radii $(R, r)$, step-size $\eta$, squeeze parameters $(\delta, \delta', \tau)$, $\tilde{\mathbf{y}}_1 = \mathbf{p}$.
**for** $t = 1, \ldots, T$ **do**
    Sample $\mathbf{z}_t \sim \mathbb{S}_{\tilde{\mathbf{y}}_t}(\delta')$; play $\tilde{\mathbf{z}}_t := \mathrm{Exp}_{\mathbf{p}}\left(\frac{\mathrm{Exp}_{\mathbf{p}}^{-1}\mathbf{z}_t}{1+\tau}\right)$ // $\delta' = \frac{\sqrt{-\kappa}(R+r)}{\sinh(\sqrt{-\kappa}(R+r))}\tau r$
    Observe $f_t(\mathbf{z}_t)$; $\mathbf{g}_t = f_t(\mathbf{z}_t) \cdot \frac{\mathrm{Exp}_{\tilde{\mathbf{y}}_t}^{-1}\mathbf{z}_t}{\|\mathrm{Exp}_{\tilde{\mathbf{y}}_t}^{-1}\mathbf{z}_t\|}$; $\mathbf{y}_{t+1} = \mathrm{Exp}_{\tilde{\mathbf{y}}_t}(-\eta\mathbf{g}_t)$
    $\tilde{\mathbf{y}}_{t+1} \leftarrow$ Output of Algorithm 2 with $\mathcal{K}$, radius $r$, squeeze parameter $\delta$ and initial point $\mathbf{y}_{t+1}$.
**end**

---

[4]An exponential dependence on the diameter is common on manifolds with strictly negative curvature. For instance, the volume of a geodesic ball grows exponentially with its radius in hyperbolic space, whereas this dependence is only polynomial in Euclidean space. This property has been leveraged to construct lower bounds for convex optimization on manifolds (Criscitiello & Boumal, 2022).

**Theorem 2.** *(Proof in Appendix B.4) Under Assumptions 2, 3 and 4. Set $\eta = T^{-\frac{1}{2}}$, $\delta = T^{-\frac{1}{4}}$, $\tau = T^{-\frac{1}{4}}$. Then regret of Algorithm 4 is upper bounded by*

$$\sup_{[s,e]\subseteq[T]} \left\{ \sum_{t=s}^{e} f_t(\widetilde{\mathbf{z}}_t) - \min_{\mathbf{x}_I \in \mathcal{K}} \sum_{t=s}^{e} f_t(\mathbf{x}_I) \right\} = O\left(T^{\frac{3}{4}}\right),$$

*and the number of calls to the separation oracle is $O(T)$.*

**Remark 2.** *An acute reader may notice a discrepancy between the step-size $\eta = O\left(T^{-\frac{1}{2}}\right)$ in our work and the step-size $\eta = O\left(T^{-\frac{3}{4}}\right)$ in Garber & Kretzu (2022, Theorem 15). It is important to highlight that our $\eta$ is equivalent to $\frac{n\eta}{\delta'}$ as per Garber & Kretzu (2022), ensuring that the parameters are in fact consistent. This reasoning is also applicable to Theorem 3.*

Interestingly, in the context of the two-point feedback setting, we can get rid of the non-standard setting by adapting the algorithm. In Algorithm 5, we adhere to the loop invariants: $\mathbf{x}_t \in \beta\mathcal{K}$ and $\mathbf{y}_t \in \mathcal{K}$. Each round involves constructing an unbiased gradient estimator $\mathbf{g}_t$ at $\mathbf{x}_t$ using the two-point feedback. Subsequently, we map $\mathbf{g}_t$ to the tangent space of $\mathbf{y}_t$ and execute online gradient descent in that space. A key advantage of this parallel transport is the ability to employ the separation oracle on $\mathcal{K}$ to construct an infeasible projection onto $(1-\delta)\mathcal{K}$. Upon a meticulous analysis, we observe that, at each round, an additional distortion arises from this parallel transport:

$$\frac{S_{\delta'}}{V_{\delta'}} \mathbb{E}[\langle \mathbf{g}_t, \Gamma_{\mathbf{y}_t}^{\mathbf{x}_t} \mathrm{Exp}_{\mathbf{y}_t}^{-1}\mathbf{x} - \mathrm{Exp}_{\mathbf{x}_t}^{-1}\mathbf{x}\rangle] \leq \frac{S_{\delta'}}{V_{\delta'}} \mathbb{E}[\|\mathbf{g}_t\|] \cdot O(\delta') = O(\delta'),$$

This distortion accumulates to $O(\sqrt{T})$ when choosing $\delta' = O\left(\frac{1}{\sqrt{T}}\right)$, ensuring the regret bound remains unaffected. The regret assurance of Algorithm 5 is detailed in Theorem 3.

---

**Algorithm 5:** Two-point bandit convex optimization on manifolds with a separation oracle

---

**Data:** feasible gsc-convex set $\mathcal{K}$, radii $(R, r)$, step-size $\eta$, parameters $(\delta, \delta', \beta)$:
$\quad \delta \in (0,1), \beta \in (0,1), \delta' = (1-\beta)\frac{\sqrt{-\kappa}(R+r)}{\sinh(\sqrt{-\kappa}(R+r))}r.$

Initialize $\mathbf{x}_1 \in \beta\mathcal{K}$, $\mathbf{y}_1 = \mathrm{Exp}_{\mathbf{p}}\left(\frac{\mathrm{Exp}_{\mathbf{p}}^{-1}\mathbf{x}_1}{\beta}\right)$ // $\mathbf{y}_1 \in \mathcal{K}$

**for** $t = 1, \ldots, T$ **do**
    Sample $\mathbf{z}_t \sim \mathbb{S}_{\mathbf{x}_t}(\delta')$
    Play $\mathbf{z}_t$ and its antipodal point $\tilde{\mathbf{z}}_t$
    Observe $f_t$ at $\mathbf{z}_t$ and $\tilde{\mathbf{z}}_t$
    Construct $\mathbf{g}_t$ by $\mathbf{g}_t = \frac{1}{2}(f_t(\mathbf{z}_t) - f_t(\tilde{\mathbf{z}}_t))\frac{\mathrm{Exp}_{\mathbf{x}_t}^{-1}\mathbf{z}_t}{\|\mathrm{Exp}_{\mathbf{x}_t}^{-1}\mathbf{z}_t\|}$
    $\mathbf{y}_{t+1}' = \mathrm{Exp}_{\mathbf{y}_t}(-\eta\Gamma_{\mathbf{x}_t}^{\mathbf{y}_t}\mathbf{g}_t)$
    $\mathbf{y}_{t+1} \leftarrow$ Output of Algorithm 2 with $\mathcal{K}$, radius $r$, squeeze parameter $\delta$ and initial point $\mathbf{y}_{t+1}'$
    $\mathbf{x}_{t+1} = \mathrm{Exp}_{\mathbf{p}}\left(\beta\mathrm{Exp}_{\mathbf{p}}^{-1}\mathbf{y}_{t+1}\right)$ // $\mathbf{x}_{t+1} \in \beta\mathcal{K}$
**end**

---

**Theorem 3.** *(Proof in Appendix B.5) Under Assumptions 2, 3 and 4. Set $\eta = 1$, $\delta = 1 - \beta = T^{-\frac{1}{2}}$, then the regret of Algorithm 5 is upper bounded by*

$$\sup_{[s,e]\subseteq[T]} \left\{ \mathbb{E}\left[\sum_{t=s}^{e} f_t(\mathbf{z}_t) - \min_{\mathbf{x}^* \in \mathcal{K}} \sum_{t=s}^{e} f_t(\mathbf{x}^*)\right] \right\} = O\left(\sqrt{T}\right)$$

*and the number of calls to the separation oracle is $O(T)$.*

## 6 Riemannian OCO with a Linear Optimization Oracle

In this section, we focus on performing projection-free OCO on Riemannian manifolds utilizing a linear oracle. The linear oracle is invoked inside the Riemannian Frank-Wolfe (RFW) algorithm (Weber & Sra, 2022b). We outline how to obtain a separating hyperplane by RFW, as detailed in Algorithm 6 and Lemma 5.

---

**Algorithm 6:** Separating Hyperplane via RFW

---

**Data:** feasible gsc-convex set $\mathcal{K}$, error tolerance $\epsilon$, initial point $\mathbf{x}_1 \in \mathcal{K}$, target vector $\mathbf{y}$.

**for** $i = 1, \dots$ **do**

     $\mathbf{v}_i = \operatorname{argmin}_{\mathbf{x} \in \mathcal{K}} \{ \langle -\operatorname{Exp}_{\mathbf{x}_i}^{-1} \mathbf{y}, \operatorname{Exp}_{\mathbf{x}_i}^{-1} \mathbf{x} \rangle \}$

     **if** $\{ \langle \operatorname{Exp}_{\mathbf{x}_i}^{-1} \mathbf{y}, \operatorname{Exp}_{\mathbf{x}_i}^{-1} \mathbf{v}_i \rangle \} \leq \epsilon$ *or* $d(\mathbf{x}_i, \mathbf{y})^2 \leq 3\epsilon$ **then**

         **return** $\tilde{\mathbf{x}} \leftarrow \mathbf{x}_i$

     **end**

     $\sigma_i = \operatorname{argmin}_{\sigma \in [0,1]} \{ d(\mathbf{y}, \operatorname{Exp}_{\mathbf{x}_i}(\sigma \operatorname{Exp}_{\mathbf{x}_i}^{-1} \mathbf{v}_i))^2 \}$

     $\mathbf{x}_{i+1} = \operatorname{Exp}_{\mathbf{x}_i}(\sigma_i \operatorname{Exp}_{\mathbf{x}_i}^{-1} \mathbf{v}_i)$

**end**

---

**Lemma 5.** *(Proof in Appendix C.1) Under Assumptions 1 and 3. For any* $\mathbf{y} \in \mathbb{B}_{\mathbf{p}}(R)$, *Algorithm 6 terminates after at most* $\zeta \lceil (27R^2/\epsilon) - 2 \rceil$ *iterations and returns* $\tilde{\mathbf{x}} \in \mathcal{K}$ *satisfies:*

*1)* $d(\tilde{\mathbf{x}}, \mathbf{y})^2 \leq d(\mathbf{x}_1, \mathbf{y})^2$.

*2) At least one of the following holds:* $d(\tilde{\mathbf{x}}, \mathbf{y})^2 \leq 3\epsilon$ *or* $\forall \mathbf{z} \in \mathcal{K} : \langle \operatorname{Exp}_{\mathbf{y}}^{-1} \mathbf{z}, \operatorname{Exp}_{\mathbf{y}}^{-1} \tilde{\mathbf{x}} \rangle \geq 2\epsilon$.

*3) If* $d(\mathbf{y}, \mathcal{K}) \leq \epsilon$ *then* $d(\tilde{\mathbf{x}}, \mathbf{y})^2 \leq 3\epsilon$.

**Remark 3.** *Note that the second item of Lemma 5 provides a separating hyperplane between* $\mathbf{y}$ *and* $\mathcal{K}$. *One of the challenges in its proof is to find an analog of the Euclidean identity* $\langle \mathbf{z} - \mathbf{y}, \tilde{\mathbf{x}} - \mathbf{y} \rangle = \|\tilde{\mathbf{x}} - \mathbf{y}\|_2^2 - \langle \mathbf{z} - \tilde{\mathbf{x}}, \mathbf{y} - \tilde{\mathbf{x}} \rangle$ *on manifolds, which initially appears to be a daunting task. However, a clever application of Lemma 30 (Appendix E) provides a solution.*

---

**Algorithm 7:** Closer Infeasible Projection via LOO

---

**Data:** feasible gsc-convex set $\mathcal{K}$, $\mathbf{x}_0 \in \mathcal{K}$, initial point $\mathbf{y}_0$, error tolerance $\epsilon$, step size $\gamma$.

$\mathbf{y}_1 = \Pi_{\mathbb{B}_{\mathbf{p}}(R)} \mathbf{y}_0$

**if** $d(\mathbf{x}_0, \mathbf{y}_0)^2 \leq 3\epsilon$ **then**

     **return** $\mathbf{x} \leftarrow \mathbf{x}_0, \mathbf{y} \leftarrow \mathbf{y}_1$.

**end**

**for** $i = 1, \dots, T$ **do**

     $\mathbf{x}_i \leftarrow$ Output of Algorithm 6 with set $\mathcal{K}$, feasible point $\mathbf{x}_{i-1}$, initial point $\mathbf{y}_i$ and tolerance $\epsilon$.

     **if** $d(\mathbf{x}_i, \mathbf{y}_i)^2 > 3\epsilon$ **then**

         $\mathbf{y}_{i+1} = \operatorname{Exp}_{\mathbf{x}_i}((1 - \gamma) \operatorname{Exp}_{\mathbf{x}_i}^{-1} \mathbf{y}_i)$

     **else**

         **return** $\mathbf{x} \leftarrow \mathbf{x}_i, \mathbf{y} \leftarrow \mathbf{y}_i$.

     **end**

**end**

---

Algorithm 7 demonstrates how to "pull" an initial point $\mathbf{y}_0$ towards $\mathcal{K}$ using RFW, while Lemma 6 verifies that the output of Algorithm 7 is indeed an infeasible projection onto $\mathcal{K}$.

**Lemma 6.** *(Proof in Appendix C.2) Under Assumptions 1 and 3. Fix* $\epsilon > 0$. *Setting* $\gamma = \frac{2\epsilon}{d(\mathbf{x}_0, \mathbf{y}_0)^2}$, *Algorithm 7 stops after at most* $\max \left\{ \frac{\zeta d(\mathbf{x}_0, \mathbf{y}_0)^2 (d(\mathbf{x}_0, \mathbf{y}_0)^2 - \epsilon)}{4\epsilon^2} + 1, 1 \right\}$ *iterations, and returns* $(\mathbf{x}, \mathbf{y}) \in \mathcal{K} \times \mathbb{B}_{\mathbf{p}}(R)$ *such that*

$$\forall \mathbf{z} \in \mathcal{K} : \ d(\mathbf{y}, \mathbf{z})^2 \leq d(\mathbf{y}_0, \mathbf{z})^2 \quad and \quad d(\mathbf{x}, \mathbf{y})^2 \leq 3\epsilon.$$

Given that Lemma 6 provides an infeasible projection oracle, we can combine Algorithms 1 and 7 to achieve sublinear regret by setting the error tolerance as $\epsilon = o(1)$. However, RFW requires $\Theta(1/\epsilon) = \omega(1)$ iterations in the worst-case scenario (Lemma 5), and the resulting algorithm necessitates $\omega(T)$ calls to the linear optimization oracle. Garber & Kretzu (2022) utilize a block trick to address this challenge: the time horizon $T$ is broken into $B$ blocks, and the infeasible projection is computed once in each block. We demonstrate that this trick can be implemented on Riemannian manifolds in Algorithm 8. We present the regret guarantees for gsc-convex and strongly gsc-convex losses in Theorem 4 and Theorem 5, respectively.

**Theorem 4.** *(Proof in Appendix C.3) Under Assumptions 1, 3 and 4. Fixing* $\eta_i$ *and* $\epsilon_i$ *as* $\eta = \frac{R\zeta}{G} T^{-\frac{3}{4}}$ *and* $\epsilon = 60 R^2 \zeta^2 T^{-\frac{1}{2}}$, *respectively, for any* $i \in \left\{ 1, \dots, \frac{T}{B} \right\}$. *Setting* $B = 5 T^{1/2}$. *Then the regret of*

---

**Algorithm 8:** Block OGD on manifolds with a linear optimization oracle

---

**Data:** horizon $T$, feasible gsc-convex set $\mathcal{K}$, number of blocks $B$, step-sizes $\{\eta_i\}_{i=1}^{\frac{T}{B}}$, error
  tolerances $\{\epsilon_i\}_{i=1}^{\frac{T}{B}}$.
Choose $\mathbf{x}_0, \mathbf{x}_1 \in \mathcal{K}$
$\tilde{\mathbf{y}}_0 = \mathbf{x}_0, \mathbf{y}_1 = \mathbf{x}_0, \tilde{\mathbf{y}}_1 = \mathbf{x}_1, \nabla_1 = \mathbf{0} \in T_{\tilde{\mathbf{y}}_0}\mathcal{M}$
**for** $t = 1, \ldots, B$ **do**
  Play $\mathbf{x}_0$; observe $f_t(\mathbf{x}_0)$
  $\nabla_1 = \nabla_1 + \nabla f_t(\tilde{\mathbf{y}}_0)$
**end**
$\mathbf{y}_{B+1} = \mathrm{Exp}_{\tilde{\mathbf{y}}_0}(-\eta_1 \nabla_1)$.
**for** $i = 2, \ldots, \frac{T}{B}$ **do**
  $(\mathbf{x}_i, \tilde{\mathbf{y}}_i) \in \mathcal{K} \times \mathbb{B}_{\mathbf{p}}(R) \leftarrow$ output of Algorithm 7 with input $\mathcal{K}, \mathbf{x}_{i-2}, \mathbf{y}_{(i-1)B+1}$ and $\epsilon_i$.
  $\mathbf{y}_{(i-1)B+1} = \tilde{\mathbf{y}}_{i-1}; \nabla_i = \mathbf{0} \in T_{\tilde{\mathbf{y}}_{i-1}}\mathcal{M}$.
  **for** $s = 1, \ldots, B$ **do**
    Play $\mathbf{x}_{i-1}$ and observe $f_t(\mathbf{x}_{i-1})$
    $\nabla_i = \nabla_i + \nabla f_{(i-1)B+s}(\tilde{\mathbf{y}}_{i-1})$
  **end**
  $\mathbf{y}_{iB+1} = \mathrm{Exp}_{\tilde{\mathbf{y}}_{i-1}}(-\eta_i \nabla_i)$.
**end**

---

*Algorithm 8 for gsc-convex losses is bounded by*

$$\sup_{I=[s,e]\subseteq[T]} \left\{ \sum_{t=s}^{e} f_t(\mathbf{x}_t) - \min_{\mathbf{x}_I \in \mathcal{K}} \sum_{t=s}^{e} f_t(\mathbf{x}_I) \right\} \le GR\left( \frac{5}{2}T^{3/4}\zeta^2 + \zeta\sqrt{180}T^{3/4} + 4T^{3/4}/\zeta + 20T^{1/2} \right),$$

*and the number of calls to the linear optimization oracle is bounded by $T$.*

**Theorem 5.** *(Proof in Appendix C.4) Under Assumptions 1, 3 and 4. Suppose all losses $\{f_t\}_{t=1}^{T}$ are $\alpha$-strongly gsc-convex for some known $\alpha > 0$ and $T \ge 3B$. Choosing $\epsilon_i = \left(\frac{20G}{\alpha(i+3)}\right)^2 \zeta$ and $\eta_i = \frac{2}{\alpha i B}$ for any $i = 1, \ldots, \frac{T}{B}$. With $B = \left(\frac{\alpha R}{G}\right)^{\frac{2}{3}} T^{2/3}$ and assume $T \ge 3B$, the regret guarantee of Algorithm 8 is bounded by*

$$\sum_{t=1}^{T} f_t(\mathbf{x}_t) - \min_{\mathbf{x} \in \mathcal{K}} \sum_{t=1}^{T} f_t(\mathbf{x}) \le (20\sqrt{3}\zeta + 1)(G^4 R^2/\alpha)^{\frac{1}{3}} T^{2/3} \left( 1 + \frac{2}{3}\ln\left(\frac{\sqrt{T}G}{\alpha R}\right) \right).$$

*And the number of total calls to the linear optimization oracle is bounded by $\zeta T$.*

## 7 Conclusion and Perspective

This paper pioneers the exploration of projection-free online optimization on Riemannian manifolds. The primary technical challenges originate from the non-convex nature of the Riemannian hyperplane and the variable metric. These challenges are tackled effectively with the aid of the Jacobi field comparison, enabling us to establish a spectrum of sub-linear regret guarantees. Interested readers may question the difficulty of generalizing these techniques from Hadamard manifolds to $\mathrm{CAT}(\kappa)$ manifolds. Some hints toward this generalization are provided in Appendix E.3.

There are several promising directions for future research. First, there exists the potential to refine the regret bounds, particularly for strongly gsc-convex losses via a separation oracle. In the context of the separation oracle, reducing the dependence on the number of calls about the diameter of the decision set would be an intriguing objective. Moreover, devising an efficient method to optimize the linear optimization oracle objective, $\mathrm{argmin}_{\mathbf{x} \in \mathcal{K}} \langle \mathbf{g}, \mathrm{Exp}_{\mathbf{x}_0}^{-1}\mathbf{x} \rangle$, remains a notable open problem. This paper does not discuss the membership oracle, primarily because related work (Mhammedi, 2022a; Lu et al., 2023) heavily relies on the convexity of the Minkowski functional in Euclidean space, a property not guaranteed to hold on Hadamard manifolds (Theorem 7). However, this does not rule out the potential for executing OCO or convex optimization on manifolds using a membership oracle. Thus, uncovering alternative strategies to tackle this issue remains a compelling research question.

## Acknowledgments

We would like to thank five anonymous referees for constructive comments and suggestions. We gratefully thank the AI4OPT Institute for funding, as part of NSF Award 2112533. We also acknowledge the NSF for their support through Award IIS-1910077. GW would like to acknowledge an ARC-ACO fellowship provided by Georgia Tech. We would also like to thank Xi Wang from UCAS and Andre Wibisono from Yale for helpful discussions.

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

# A Omitted Proofs in Section 4

## A.1 Proof of Lemma 1

*Proof.* Fix $t$, since $\tilde{\mathbf{y}}_{t+1}$ is an infeasible projection of $\mathbf{y}_{t+1}$ onto $\tilde{\mathcal{K}}$, and $\mathbf{y}_{t+1} = \text{Exp}_{\tilde{\mathbf{y}}_t}(-\eta_t \nabla_t)$, by Lemma 29, we have for any $\mathbf{x} \in \tilde{\mathcal{K}}$,

$$d(\tilde{\mathbf{y}}_{t+1}, \mathbf{x})^2 \leq d(\mathbf{y}_{t+1}, \mathbf{x})^2 \leq \zeta d(\mathbf{y}_{t+1}, \tilde{\mathbf{y}}_t)^2 + d(\tilde{\mathbf{y}}_t, \mathbf{x})^2 - 2\left\langle \text{Exp}_{\tilde{\mathbf{y}}_t}^{-1} \mathbf{y}_{t+1}, \text{Exp}_{\tilde{\mathbf{y}}_t}^{-1} \mathbf{x} \right\rangle$$

$$\leq \zeta \eta_t^2 \|\nabla_t\|^2 + d(\tilde{\mathbf{y}}_t, \mathbf{x})^2 + 2\eta_t \left\langle \nabla_t, \text{Exp}_{\tilde{\mathbf{y}}_t}^{-1} \mathbf{x} \right\rangle. \tag{4}$$

To verify that $\zeta$ represents the correct geometric distortion, we need to show $d(\tilde{\mathbf{y}}_t, \mathbf{x}) \leq 2R$ holds for any $\tilde{\mathbf{y}}_t \in \mathbb{B}_{\mathbf{p}}(R)$ and $\mathbf{x} \in \tilde{\mathcal{K}}$. Since $\tilde{\mathcal{K}} \subseteq \mathcal{K} \subseteq \mathbb{B}_{\mathbf{p}}(R)$, we can demonstrate this by the triangle inequality:

$$d(\tilde{\mathbf{y}}_t, \mathbf{x}) \leq d(\tilde{\mathbf{y}}_t, \mathbf{p}) + d(\mathbf{p}, \mathbf{x}) \leq 2R.$$

Rearranging Equation (4), we have

$$\left\langle \nabla_t, -\text{Exp}_{\tilde{\mathbf{y}}_t}^{-1} \mathbf{x} \right\rangle \leq \frac{d(\tilde{\mathbf{y}}_t, \mathbf{x})^2}{2\eta_t} - \frac{d(\tilde{\mathbf{y}}_{t+1}, \mathbf{x})^2}{2\eta_t} + \frac{\zeta \eta_t \|\nabla_t\|^2}{2}. \tag{5}$$

Fix $1 \leq s \leq e \leq T$ and sum over $[s, e]$, we have

$$\sum_{t=s}^{e} \left\langle \nabla_t, -\text{Exp}_{\tilde{\mathbf{y}}_t}^{-1} \mathbf{x} \right\rangle \leq \frac{d(\tilde{\mathbf{y}}_s, \mathbf{x})^2}{2\eta_s} + \sum_{t=s+1}^{e} \left( \frac{1}{2\eta_t} - \frac{1}{2\eta_{t-1}} \right) d(\tilde{\mathbf{y}}_t, \mathbf{x})^2 + \sum_{t=s}^{e} \frac{\zeta \eta_t \|\nabla_t\|^2}{2}.$$

Using gsc-convexity of $f_t(\cdot)$ and set $\eta_t = \eta$, we have

$$\sum_{t=s}^{e} f_t(\tilde{\mathbf{y}}_t) - f_t(\mathbf{x}) \leq \frac{d(\tilde{\mathbf{y}}_s, \mathbf{x})^2}{2\eta} + \frac{\eta \zeta \sum_{t=s}^{e} \|\nabla_t\|^2}{2}.$$

In case all losses are $\alpha$-strongly gsc-convex, using $f_t(\mathbf{y}) - f_t(\mathbf{x}) \leq \left\langle \nabla f_t(\mathbf{y}), -\text{Exp}_{\mathbf{y}}^{-1} \mathbf{x} \right\rangle - \frac{\alpha d(\mathbf{x}, \mathbf{y})^2}{2}$ and set $(s, e) = (1, T)$, we have

$$\sum_{t=1}^{T} f_t(\tilde{\mathbf{y}}_t) - f_t(\mathbf{x}) \leq \sum_{t=1}^{T} \frac{\eta_t \|\nabla_t\|^2}{2} + \left( \frac{1}{2\eta_1} - \frac{\alpha}{2} \right) d(\tilde{\mathbf{y}}_1, \mathbf{x})^2 + \sum_{t=2}^{T} \zeta \left( \frac{1}{2\eta_t} - \frac{1}{2\eta_{t-1}} - \frac{\alpha}{2} \right) d(\tilde{\mathbf{y}}_t, \mathbf{x})^2.$$

Setting $\eta_t = \frac{1}{\alpha t}$, we get the guarantee for the strongly gsc-convex case. $\qquad\square$

## A.2 Proof of Lemma 2

*Proof.* By Lemma 29,

$$d(\tilde{\mathbf{y}}, \mathbf{z})^2 \leq \zeta d(\mathbf{y}, \tilde{\mathbf{y}})^2 + d(\mathbf{y}, \mathbf{z})^2 - 2\left\langle \text{Exp}_{\mathbf{y}}^{-1} \tilde{\mathbf{y}}, \text{Exp}_{\mathbf{y}}^{-1} \mathbf{z} \right\rangle$$

$$\leq \zeta \gamma^2 C^2 + d(\mathbf{y}, \mathbf{z})^2 - 2\left\langle -\gamma \mathbf{g}, \text{Exp}_{\mathbf{y}}^{-1} \mathbf{z} \right\rangle$$

$$\leq \zeta \gamma^2 C^2 + d(\mathbf{y}, \mathbf{z})^2 - 2\gamma Q$$

$$= d(\mathbf{y}, \mathbf{z})^2 - \frac{Q^2}{\zeta C^2},$$

where the first inequality relies on Lemma 29 and $d(\mathbf{y}, \mathbf{z}) \leq 2R$, the third inequality holds because $-\left\langle \text{Exp}_{\mathbf{y}}^{-1} \mathbf{z}, \mathbf{g} \right\rangle \geq Q$, and for the last line, we plug in $\gamma = \frac{Q}{\zeta C^2}$. $\qquad\square$

# B Omitted Proofs in Section 5

The technical difficulty lies in constructing an infeasible projection oracle using a separation oracle. The following lemma is instrumental to the later results.

## B.1 Proof of Lemma 3

*Proof.* Let $\mathbf{x} = \text{Exp}_{\mathbf{y}}\left(\delta\bar{r}\frac{\mathbf{g}}{\|\mathbf{g}\|} + \text{Exp}_{\mathbf{y}}^{-1}\mathbf{z}\right)$, if we can show $\mathbf{x} \in \mathcal{K}$, then by the definition of the separation oracle, we have

$$\left\langle -\text{Exp}_{\mathbf{y}}^{-1}\mathbf{x}, \mathbf{g}\right\rangle = -\delta\bar{r}\|\mathbf{g}\| - \left\langle \text{Exp}_{\mathbf{y}}^{-1}\mathbf{z}, \mathbf{g}\right\rangle > 0,$$

which in turn implies $\left\langle -\text{Exp}_{\mathbf{y}}^{-1}\mathbf{z}, \mathbf{g}\right\rangle > \delta\bar{r}\|\mathbf{g}\|$. Note that when $\bar{r} = 0$, $\mathbf{x} = \mathbf{z} \in (1-\delta)\mathcal{K} \subset \mathcal{K}$, thus there exists positive $\bar{r}$ such that $\mathbf{x} \in \mathcal{K}$ for any $\mathbf{z} \in (1-\delta)\mathcal{K}$. Let $\tilde{r} = \frac{\sqrt{-\kappa}(R+r)}{\sinh(\sqrt{-\kappa}(R+r))} \cdot r$. By Lemma 19, we have $\mathbb{B}_{\mathbf{z}}(\delta\tilde{r}) \subset \mathcal{K}$. Thus we only need to ensure $d(\mathbf{x}, \mathbf{z}) \leq \delta\tilde{r}$ because this implies $\mathbf{x} \in \mathbb{B}_{\mathbf{z}}(\delta\tilde{r}) \subset \mathcal{K}$. We define an admissible family of curves:

$$\Gamma : [0,1] \times [0,1] \to \mathcal{M} \qquad (t,s) \to \text{Exp}_{\mathbf{y}}(t(\text{Exp}_{\mathbf{y}}^{-1}\mathbf{z} + s\delta\bar{r}\mathbf{v}))$$

where $\mathbf{v} = \frac{\mathbf{g}}{\|\mathbf{g}\|}$. Then $\Gamma(1,0) = \mathbf{z}$ and $\Gamma(1,1) = \mathbf{x}$. Let $J_s(t) := \frac{\partial\Gamma(t,s)}{\partial s}$, which is a variation field of the variation $\Gamma$, and is thus a Jacobi field. We can compute $J_s(0) = \frac{\partial\Gamma(0,s)}{\partial s} = 0$ and by Lemma 14,

$$D_t J_s(t)|_{t=0} = \delta\bar{r}\mathbf{v}.$$

To apply Lemma 15, we need to reparametrize $\Gamma$ to make it unit speed. Let $\Gamma(t,s) = \tilde{\Gamma}\left(\frac{t}{R(s)}, s\right)$ where

$$R(s) = d(\Gamma(1,s), \Gamma(0,s)) = \|\text{Exp}_{\mathbf{y}}^{-1}\mathbf{z} + s\delta\bar{r}\mathbf{v}\| \leq \|\text{Exp}_{\mathbf{y}}^{-1}\mathbf{z}\| + \delta\bar{r} \leq 2R + \delta r \leq 2R + r.$$

Then we have $J_s(t) = \tilde{J}_s(R(s)t)$ and $D_t J_s(t) = R(s)D_t\tilde{J}_s(R(s)t)$. Now we can apply Lemma 15:

$$
\begin{aligned}
\|J_s(1)\| &= \|\tilde{J}_s(R(s))\| \leq \frac{\sinh\left(\sqrt{-\kappa}R(s)\right)\|D_t\tilde{J}_s(0)\|}{\sqrt{-\kappa}} \\
&= \frac{\sinh\left(\sqrt{-\kappa}R(s)\right)\|D_t J_s(0)\|}{\sqrt{-\kappa}R(s)} = \frac{\sinh\left(\sqrt{-\kappa}R(s)\right)\delta\bar{r}}{\sqrt{-\kappa}R(s)}
\end{aligned}
\tag{6}
$$

We would like to have $d(\mathbf{x}, \mathbf{z}) \leq \delta\tilde{r}$, and it suffices to choose $\bar{r}$ such that

$$d(\mathbf{x}, \mathbf{z}) \leq \int_0^1 \|J_s(1)\|ds \leq \frac{\sinh(\sqrt{-\kappa}(2R+r))}{\sqrt{-\kappa}(2R+r)}\delta\bar{r} \leq \delta\tilde{r}.$$

We then find a valid $\bar{r}$ is

$$\bar{r} = \frac{\sqrt{-\kappa}(2R+r)}{\sinh(\sqrt{-\kappa}(2R+r))}\tilde{r} = \frac{\sqrt{-\kappa}(2R+r)}{\sinh(\sqrt{-\kappa}(2R+r))} \cdot \frac{\sqrt{-\kappa}(R+r)}{\sinh(\sqrt{-\kappa}(R+r))} \cdot r.$$

$\square$

## B.2 Proof of Lemma 4

*Proof.* We use $k$ to denote the number of iterations in Algorithm 2, then for any $i < k$, we have $\mathbf{y}_i \notin \mathcal{K}$. By Lemma 3, we have $\left\langle -\text{Exp}_{\mathbf{y}_i}^{-1}\mathbf{z}, \mathbf{g}_i\right\rangle \geq \delta\bar{r}\|\mathbf{g}_i\|$ for any $\mathbf{z} \in (1-\delta)\mathcal{K}$. We then invoke Lemma 2 with $C = \|\mathbf{g}_i\|$, $Q = \delta\bar{r}\|\mathbf{g}_i\|$ to get

$$d(\mathbf{y}_{i+1}, \mathbf{z})^2 \leq d(\mathbf{y}_i, \mathbf{z})^2 - \frac{\delta^2\bar{r}^2}{\zeta}
\tag{7}$$

holds for any $\mathbf{z} \in (1-\delta)\mathcal{K}$. To ensure the geometric distortion $\zeta$ is valid here, we need to prove $d(\mathbf{y}_i, \mathbf{z}) \leq 2R$ holds for any $i \geq 1$ and $\mathbf{z} \in (1-\delta)\mathcal{K}$, which can be guaranteed by induction. The case of $i = 1$ is straightforward. As $\mathbf{y}_1 \in \mathbb{B}_{\mathbf{p}}(R)$, $\mathbf{z} \in (1-\delta)\mathcal{K} \subseteq \mathbb{B}_{\mathbf{p}}(R)$, we have

$$d(\mathbf{y}_1, \mathbf{z}) \leq d(\mathbf{y}_1, \mathbf{p}) + d(\mathbf{z}, \mathbf{p}) \leq 2R.$$

Now assume $d(\mathbf{y}_i, \mathbf{z}) \le 2R$ holds for some $i \ge 1$ and any $\mathbf{z} \in (1-\delta)\mathcal{K}$, then by Lemma 29 and Lemma 2,

$$
\begin{aligned}
d(\mathbf{y}_{i+1}, \mathbf{z})^2 &\le \zeta(\kappa, d(\mathbf{y}_i, \mathbf{z})) \cdot d(\mathbf{y}_{i+1}, \mathbf{y}_i)^2 + d(\mathbf{y}_i, \mathbf{z})^2 - 2\left\langle \mathrm{Exp}^{-1}_{\mathbf{y}_i}\mathbf{y}_{i+1}, \mathrm{Exp}^{-1}_{\mathbf{y}_i}\mathbf{z}\right\rangle \\
&\le \zeta(\kappa, 2R) \cdot d(\mathbf{y}_{i+1}, \mathbf{y}_i)^2 + d(\mathbf{y}_i, \mathbf{z})^2 - 2\left\langle \mathrm{Exp}^{-1}_{\mathbf{y}_i}\mathbf{y}_{i+1}, \mathrm{Exp}^{-1}_{\mathbf{y}_i}\mathbf{z}\right\rangle \\
&= \zeta d(\mathbf{y}_{i+1}, \mathbf{y}_i)^2 + d(\mathbf{y}_i, \mathbf{z})^2 - 2\left\langle \mathrm{Exp}^{-1}_{\mathbf{y}_i}\mathbf{y}_{i+1}, \mathrm{Exp}^{-1}_{\mathbf{y}_i}\mathbf{z}\right\rangle \\
&\le d(\mathbf{y}_i, \mathbf{z})^2 - \frac{\delta^2 \bar{r}^2}{\zeta} \le (2R)^2.
\end{aligned}
$$

Thus, we know $d(\mathbf{y}_i, \mathbf{z}) \le 2R$ holds for any $i \ge 1$ and $\mathbf{z} \in (1-\delta)\mathcal{K}$.

By Equation (7), $d(\mathbf{y}, \mathbf{z})^2 \le d(\mathbf{y}_1, \mathbf{z})^2$ for all $\mathbf{z} \in (1-\delta)\mathcal{K}$. Since $\mathbf{y}_1$ is the projection of $\mathbf{y}_0$ onto $\mathbb{B}_{\mathbf{p}}(R)$, we have $d(\mathbf{y}_1, \mathbf{z})^2 \le d(\mathbf{y}_0, \mathbf{z})^2$ holds for $\mathbf{z} \in \mathbb{B}_{\mathbf{p}}(R)$. Thus, we indeed have $d(\mathbf{y}, \mathbf{z})^2 \le d(\mathbf{y}_0, \mathbf{z})^2$ for any $\mathbf{z} \in (1-\delta)\mathcal{K}$.

It remains to bound the number of iterations in Algorithm 2. We must be careful because, on manifolds, $(1-\delta)\mathcal{K}$ is not guaranteed to be gsc-convex (Theorem 6). Note that $\mathrm{argmin}_{\mathbf{x} \in (1-\delta)\mathcal{K}} d(\mathbf{y}_i, \mathbf{x})^2$ is consistently non-empty because $(1-\delta)\mathcal{K}$ is a closed set. Let $\mathbf{x}_i^* \in \mathrm{argmin}_{\mathbf{x} \in (1-\delta)\mathcal{K}} d(\mathbf{y}_i, \mathbf{x})^2$ and invoke Equation (7), we have

$$
\begin{aligned}
d(\mathbf{y}_{i+1}, (1-\delta)\mathcal{K})^2 &= d(\mathbf{y}_{i+1}, \mathbf{x}_{i+1}^*)^2 \\
&\le d(\mathbf{y}_{i+1}, \mathbf{x}_i^*)^2 \le d(\mathbf{y}_i, \mathbf{x}_i^*)^2 - \frac{\delta^2 \bar{r}^2}{\zeta} = d(\mathbf{y}_i, (1-\delta)\mathcal{K})^2 - \frac{\delta^2 \bar{r}^2}{\zeta}.
\end{aligned}
$$

The first inequality is due to the definition of $\mathbf{x}_{i+1}^*$ and the second one follows from Equation (7). We also need to show $d(\mathbf{y}_1, (1-\delta)\mathcal{K}) \le d(\mathbf{y}_0, (1-\delta)\mathcal{K})$ to finish the proof. We have

$$
d(\mathbf{y}_1, \mathbf{x}_1^*) \le d(\mathbf{y}_1, \mathbf{x}_0^*) \le d(\mathbf{y}_0, \mathbf{x}_0^*), \tag{8}
$$

where the first inequality is again due to the definition of $\mathbf{x}_1^*$, while the second one comes from $\mathbf{y}_1$ is the metric projection of $\mathbf{y}_0$ onto $\mathbb{B}_{\mathbf{p}}(R)$ and $\mathbf{x}_0^* \in (1-\delta)\mathcal{K} \subseteq \mathbb{B}_{\mathbf{p}}(R)$. Reminding $\mathbf{y} = \mathbf{y}_k$ and unrolling the recurrence, we have

$$
d(\mathbf{y}, (1-\delta)\mathcal{K})^2 \le d(\mathbf{y}_0, (1-\delta)\mathcal{K})^2 - \frac{(k-1)\delta^2 \bar{r}^2}{\zeta}.
$$

Thus, in the worst case, we need

$$
k = \frac{\zeta\left(d(\mathbf{y}_0, (1-\delta)\mathcal{K})^2 - d(\mathbf{y}, (1-\delta)\mathcal{K})^2\right)}{\delta^2 \bar{r}^2} + 1
$$

iterations to stop. $\qquad\square$

### B.3 Proof of Theorem 1

*Proof.* Combining Algorithm 3 with Lemma 4, we know $\tilde{\mathbf{y}}_t \in \mathcal{K}$ thus Algorithm 3 plays a feasible point at each round. For a fixed interval $I = [s, e] \subseteq [T]$, let $\mathbf{x}_I = \mathrm{argmin}_{\mathbf{x} \in \mathcal{K}} \sum_{t=s}^{e} f_t(\mathbf{x})$ and $\tilde{\mathbf{x}}_I = \mathrm{Exp}_{\mathbf{p}}((1-\delta)\mathrm{Exp}^{-1}_{\mathbf{p}}\mathbf{x}_I)$. Then

$$
d(\tilde{\mathbf{x}}_I, \mathbf{x}_I) = d(\mathrm{Exp}_{\mathbf{p}}((1-\delta)\mathrm{Exp}^{-1}_{\mathbf{p}}\mathbf{x}_I), \mathbf{x}_I) = d(\mathrm{Exp}_{\mathbf{x}_I}(\delta \mathrm{Exp}^{-1}_{\mathbf{x}_I}\mathbf{p}), \mathbf{x}_I) = \delta d(\mathbf{p}, \mathbf{x}_I)^2 \le \delta R.
$$

By Lemma 4, $\tilde{\mathbf{y}}_t \in \mathcal{K}$ is an infeasible projection of $\mathbf{y}_t$ over $(1-\delta)\mathcal{K}$. Then by Lemma 1

$$
\sum_{t=s}^{e} (f_t(\tilde{\mathbf{y}}_t) - f_t(\tilde{\mathbf{x}}_I)) \le \frac{d(\tilde{\mathbf{y}}_s, \mathbf{x}_I)^2}{2\eta} + \frac{\eta\zeta \sum_{t=s}^{e} \|\nabla f_t(\tilde{\mathbf{y}}_t)\|^2}{2} \le \frac{2R^2}{\eta} + \frac{\eta\zeta G^2 T}{2}.
$$

On the other hand, by the gradient-Lipschitzness, we have

$$
f_t(\tilde{\mathbf{x}}_I) - f_t(\mathbf{x}_I) \le G \cdot d(\tilde{\mathbf{x}}_I, \mathbf{x}_I) \le G\delta R.
$$

Combining the above two equations, we have

$$
\sum_{t=s}^{e} (f_t(\tilde{\mathbf{y}}_t) - f_t(\mathbf{x}_I^*)) \le \left(GR\delta + \frac{G^2 \eta\zeta}{2}\right) T + \frac{2R^2}{\eta}.
$$

Setting $\eta = \frac{2R}{G\sqrt{\zeta T}}$ and $\delta = \frac{1}{2\sqrt{T}} < 1$, then we have

$$\sum_{t=s}^{e} (f_t(\tilde{\mathbf{y}}_t) - f_t(\mathbf{x}_I^*)) \le \frac{5}{2} GR\sqrt{\zeta T},$$

where we use $\zeta \ge 1$. It remains to bound the number of calls to the separation oracle. Denote $\mathbf{x}_t^* \in \arg\min_{\mathbf{x} \in (1-\delta)\mathcal{K}} d(\mathbf{x}, \tilde{\mathbf{y}}_t)$. Due to $\mathbf{y}_{t+1} = \mathrm{Exp}_{\tilde{\mathbf{y}}_t}(-\eta \nabla f_t(\tilde{\mathbf{y}}_t))$,

$$d(\mathbf{y}_{t+1}, (1-\delta)\mathcal{K}) \le d(\mathbf{x}_t^*, \mathbf{y}_{t+1}) \le d(\mathbf{x}_t^*, \tilde{\mathbf{y}}_t) + d(\tilde{\mathbf{y}}_t, \mathbf{y}_{t+1}) \le d(\tilde{\mathbf{y}}_t, (1-\delta)\mathcal{K}) + \eta\|\nabla f_t(\tilde{\mathbf{y}}_t)\|,$$

where the first inequality is because $\mathbf{x}_t^* \in (1-\delta)\mathcal{K}$, the second is due to the triangle inequality, while the third one follows from the definition of $\mathbf{x}_t^*$ and $\mathbf{y}_{t+1} = \mathrm{Exp}_{\tilde{\mathbf{y}}_t}(-\eta \nabla f_t(\tilde{\mathbf{y}}_t))$. Squaring both sides, we have

$$\begin{aligned}
d(\mathbf{y}_{t+1}, (1-\delta)\mathcal{K})^2 &\le d(\tilde{\mathbf{y}}_t, (1-\delta)\mathcal{K})^2 + 2d(\tilde{\mathbf{y}}_t, (1-\delta)\mathcal{K})\eta G + \eta^2 G^2 \\
&\le d(\tilde{\mathbf{y}}_t, (1-\delta)\mathcal{K})^2 + 2\delta R \eta G + \eta^2 G^2,
\end{aligned} \tag{9}$$

where the second inequality is because

$$d(\tilde{\mathbf{y}}_t, (1-\delta)\mathcal{K}) \le d(\tilde{\mathbf{y}}_t, \mathrm{Exp}_{\mathbf{p}}((1-\delta)\mathrm{Exp}_{\mathbf{p}}^{-1}\tilde{\mathbf{y}}_t)) = d(\tilde{\mathbf{y}}_t, \mathrm{Exp}_{\tilde{\mathbf{y}}_t}(\delta \mathrm{Exp}_{\tilde{\mathbf{y}}_t}^{-1}\mathbf{p})) = \delta d(\tilde{\mathbf{y}}_t, \mathbf{p}) \le \delta R.$$

Combining Equation (9) with Lemma 4, we can bound the number of separation oracle calls as:

$$\begin{aligned}
N_{calls} &\le \sum_{t=1}^{T} \left( \zeta \frac{d(\mathbf{y}_{t+1}, (1-\delta)\mathcal{K})^2 - d(\tilde{\mathbf{y}}_{t+1}, (1-\delta)\mathcal{K})^2}{\delta^2 \bar{r}^2} + 1 \right) \\
&\le \sum_{t=1}^{T} \left( \zeta \frac{d(\tilde{\mathbf{y}}_t, (1-\delta)\mathcal{K})^2 + 2R\delta\eta G + \eta^2 G^2 - d(\tilde{\mathbf{y}}_{t+1}, (1-\delta)\mathcal{K})^2}{\delta^2 \bar{r}^2} + 1 \right) \\
&\le \zeta \left( \frac{2RG\eta T}{\bar{r}^2 \delta} + \frac{G^2 \eta^2 T}{\bar{r}^2 \delta^2} \right) + T \\
&= \left( \frac{8R^2\sqrt{\zeta}}{\bar{r}^2} + \frac{16R^2}{\bar{r}^2} + 1 \right) T.
\end{aligned} \tag{10}$$

where we use $\tilde{\mathbf{y}}_1 = \mathbf{p} \in (1-\delta)\mathcal{K}$ and thus $d(\tilde{\mathbf{y}}_1, (1-\delta)\mathcal{K}) = 0$ to derive the third inequality. $\qquad\square$

### B.4 Proof of Theorem 2

We first prove Lemma 7, which characterizes the relation between $\delta'$ and $\tau$.

**Lemma 7.** *For a gsc-convex subset $\mathcal{K} \subseteq \mathcal{M}$ where $\mathcal{M}$ is Hadamard, any point $\mathbf{y} \in \mathcal{K}$ and $\mathbb{B}_{\mathbf{p}}(r) \subseteq \mathcal{K} \subseteq \mathbb{B}_{\mathbf{p}}(R)$, we have*

$$\mathbb{B}_{\mathbf{y}}\left( \frac{\sqrt{-\kappa}(R+r)}{\sinh\left(\sqrt{-\kappa}(R+r)\right)} \right) \subset (1+\tau)\mathcal{K}.$$

*Proof.* The proof of this proposition takes inspiration from Lemma 19 (Wang et al., 2023), but with several significant adjustments. Notably, Lemma 19 hinges on the gsc-convexity of $\mathcal{K}$ to bound the radius of the geodesic ball $\mathbb{B}_{\mathbf{y}}(\tilde{r})$, where $\mathbf{y} \in (1-\tau)\mathcal{K}$, which resides within $\mathcal{K}$. However, the gsc-convexity of $(1+\tau)\mathcal{K}$ is unknown.[5] Therefore, we must explore alternative strategies that leverage the gsc-convexity of $\mathcal{K}$ to meet our objectives.

Let $\mathbf{v} \in T_{\mathbf{y}}\mathcal{M}$ be a unit vector such that $\|\mathbf{v}\| = 1$ and $\mathbf{z} = \mathrm{Exp}_{\mathbf{y}}(\theta\tau r\mathbf{v})$. The goal is to find the maximum $\theta$ which ensures $\mathbf{z} \in (1+\tau)\mathcal{K}$ for any $\mathbf{v}$. Let $\mathbf{y}' = \mathrm{Exp}_{\mathbf{p}}\left( \frac{\mathrm{Exp}_{\mathbf{p}}^{-1}\mathbf{y}}{1+\tau} \right)$ and $\mathbf{z}' = \mathrm{Exp}_{\mathbf{p}}\left( \frac{\mathrm{Exp}_{\mathbf{p}}^{-1}\mathbf{z}}{1+\tau} \right)$. It is immediate to see $\mathbf{z}' \in \mathcal{K}$ iff $\mathbf{z} \in (1+\tau)\mathcal{K}$. We denote $\xi_\tau(s)$ as the geodesic satisfying $\xi_\tau(0) = \mathbf{y}'$ and $\xi_\tau(1) = \mathbf{z}'$.

---

[5] We conjecture that on Hadamard manifolds, $(1+\tau)\mathcal{K}$ is gsc-convex for any $\tau \ge 0$.

We define an admissible family of curves:

$$\Gamma : \left[0, \frac{1+\tau}{\tau}\right] \times [0,1] \to \mathcal{M} \qquad (t,s) \to \mathrm{Exp}_{\mathbf{y}}(t(\mathrm{Exp}_{\mathbf{y}}^{-1}\xi_\tau(s))).$$

We can verify that $\Gamma(0,1) = \mathbf{y}, \Gamma(1,0) = \mathbf{y}', \Gamma\left(\frac{1+\tau}{\tau},0\right) = \mathrm{Exp}_{\mathbf{y}}\left(\frac{1+\tau}{\tau} \cdot \mathrm{Exp}_{\mathbf{y}}^{-1}\mathbf{y}'\right) = \mathbf{p}$, and $\Gamma(1,1) = \mathbf{z}'$. We also denote $\mathbf{w} := \Gamma\left(\frac{1+\tau}{\tau},1\right)$. The idea of the proof is to show $d(\mathbf{p},\mathbf{w}) \leq r$ by Lemma 15, which implies $\mathbf{w} \in \mathcal{K}$ since $\mathbb{B}_{\mathbf{p}}(r) \subseteq \mathcal{K}$. Combining with the fact that $\mathbf{y} \in \mathcal{K}$ and $\mathbf{z}'$ is on the geodesic connecting $\mathbf{y}$ and $\mathbf{w}$, we have $\mathbf{z}' \in \mathcal{K}$. Thus $\mathbf{z} \in (1+\tau)\mathcal{K}$.

We notice that $v(t,s) = \frac{\partial \Gamma(s,t)}{\partial s}$ is a Jacobi field since it is a variation field of $\gamma_s(t) = \mathrm{Exp}_{\mathbf{y}}(t\mathrm{Exp}_{\mathbf{y}}^{-1}\xi_\tau(s))$. Let $R(s) = d(\Gamma(0,s),\Gamma(1,s))$. To apply Lemma 15, we need to normalize the geodesic $\tilde{\gamma}_s(t) = \gamma_s\left(\frac{t}{R(s)}\right)$. Since $\mathcal{M}$ is Hadamard, by Lemma 15

$$\|v(1,s)\| \geq R(s)\|\nabla_{\dot{\tilde{\gamma}}_s} v(0,s)\|. \tag{11}$$

Remind that $\|v(1,s)\| = \left\|\frac{\partial \xi_\tau(s)}{\partial s}\right\| = \|\dot{\xi}_\tau(s)\| = d(\mathbf{y}',\mathbf{z}')$. Now we use Lemma 20 to bound $d(\mathbf{y}',\mathbf{z}')$. We construct $\Delta(\bar{\mathbf{p}},\bar{\mathbf{y}},\bar{\mathbf{z}})$ in Euclidean space, a comparison triangle of $\Delta(\mathbf{p},\mathbf{y},\mathbf{z})$ with comparison points $\bar{\mathbf{y}}' \in [\bar{\mathbf{p}},\bar{\mathbf{y}}]$ and $\bar{\mathbf{z}}' \in [\bar{\mathbf{p}},\bar{\mathbf{z}}]$. We restrict $d(\mathbf{p},\mathbf{y}') = d_{\mathbb{E}}(\bar{\mathbf{p}},\bar{\mathbf{y}}')$ and $d(\mathbf{p},\mathbf{z}') = d_{\mathbb{E}}(\bar{\mathbf{p}},\bar{\mathbf{z}}')$, then by Lemma 20, we have

$$d(\mathbf{y}',\mathbf{z}') \leq d_{\mathbb{E}}(\bar{\mathbf{y}}',\bar{\mathbf{z}}') \tag{12}$$

On the other hand, it is immediate to verify $\Delta(\bar{\mathbf{p}},\bar{\mathbf{y}},\bar{\mathbf{z}})$ is similar to $\Delta(\bar{\mathbf{p}},\bar{\mathbf{y}}',\bar{\mathbf{z}}')$ by considering

$$\frac{d_{\mathbb{E}}(\bar{\mathbf{p}}',\bar{\mathbf{z}}')}{d_{\mathbb{E}}(\bar{\mathbf{p}}',\bar{\mathbf{z}}')} = \frac{d(\mathbf{p},\mathbf{z}')}{d(\mathbf{p},\mathbf{z})} = \frac{1}{1+\tau} = \frac{d(\mathbf{p},\mathbf{y}')}{d(\mathbf{p},\mathbf{y})} = \frac{d_{\mathbb{E}}(\bar{\mathbf{p}}',\bar{\mathbf{y}}')}{d_{\mathbb{E}}(\bar{\mathbf{p}}',\bar{\mathbf{y}}')}.$$

Thus we have

$$d_{\mathbb{E}}(\bar{\mathbf{y}}',\bar{\mathbf{z}}') = \frac{1}{1+\tau}d_{\mathbb{E}}(\bar{\mathbf{y}},\bar{\mathbf{z}}) = \frac{1}{1+\tau}d(\mathbf{y},\mathbf{z}) = \frac{\theta\tau r}{1+\tau}, \tag{13}$$

where the first equation is due to the property of similar triangles, the second one follows from the definition of the comparison triangle, and the last one is due to the definition of $\mathbf{z}$.

Combining Equations (11), (12) and (13), we have

$$\|\nabla_{\dot{\tilde{\gamma}}_s} v(0,s)\| \leq \frac{\|v(1,s)\|}{R(s)} \leq \frac{\theta\tau r}{(1+\tau)R(s)}. \tag{14}$$

We also need to apply Lemma 15 at $t = \frac{1+\tau}{\tau}$. Since the sectional curvature of $\mathcal{M}$ is lower bounded by $\kappa$, we have

$$\left\|v\left(\frac{1+\tau}{\tau},s\right)\right\| \leq \frac{1}{\sqrt{-\kappa}}\sinh\left(\sqrt{-\kappa}R(s)\left(\frac{1+\tau}{\tau}\right)\right) \cdot \|\nabla_{\dot{\tilde{\gamma}}_s} v(0,s)\| \tag{15}$$

Putting Equations (14) and (15) together, we have

$$\begin{aligned}\left\|v\left(\frac{1+\tau}{\tau},s\right)\right\| &\leq \frac{\sinh\left(\sqrt{-\kappa}R(s)\left(\frac{1+\tau}{\tau}\right)\right)}{\sqrt{-\kappa}R(s)\left(\frac{1+\tau}{\tau}\right)} \cdot R(s)\frac{1+\tau}{\tau} \cdot \frac{\theta\tau r}{(1+\tau)R(s)}\\ &= \frac{\sinh\left(\sqrt{-\kappa}R(s)\left(\frac{1+\tau}{\tau}\right)\right)}{\sqrt{-\kappa}R(s)\left(\frac{1+\tau}{\tau}\right)} \cdot \theta r.\end{aligned} \tag{16}$$

Now we can bound $\theta$. Note that

$$R(s) = d(\Gamma(0,s),\Gamma(1,s)) = d(\mathbf{y},\xi_\tau(s))$$

$$\leq d(\mathbf{y},\mathbf{y}') + d(\mathbf{y}',\mathbf{z}') \leq \frac{\tau}{1+\tau}R + \frac{\theta\tau r}{1+\tau} \leq \frac{\tau}{1+\tau}(R+r).$$

By Lemma 21, we have

$$\frac{\sinh\left(\sqrt{-\kappa}R(s)\left(\frac{1+\tau}{\tau}\right)\right)}{\sqrt{-\kappa}R(s)\left(\frac{1+\tau}{\tau}\right)} \leq \frac{\sinh\left(\sqrt{-\kappa}(R+r)\right)}{\sqrt{-\kappa}(R+r)}.$$

Thus we can take

$$\theta = \frac{\sqrt{-\kappa}(R+r)}{\sinh\left(\sqrt{-\kappa}(R+r)\right)}$$

to ensure $\left\|v\left(\frac{1+\tau}{\tau}, s\right)\right\| \leq r$. The length of the curve $v\left(\frac{1+\tau}{\tau}, s\right)$ can be bounded by

$$\int_0^1 \left\|v\left(\frac{1+\tau}{\tau}\right)\right\| ds \leq r.$$

This implies $d(\mathbf{p}, \mathbf{w}) \leq r$ and thus $\mathbf{w} \in \mathcal{K}$. Then we know $\mathbf{z}' \in \mathcal{K}$ because $\mathbf{z}'$ lies on the geodesic connecting $\mathbf{w}$ and $\mathbf{y}$, with both endpoints in $\mathcal{K}$. This finally leads to the fact that $\mathbf{z} \in (1+\tau)\mathcal{K}$. $\quad\square$

Now we give the proof of Theorem 2.

*Proof of Theorem 2.* Algorithm 4 always plays feasible points because by

$$\delta' = \frac{\sqrt{-\kappa}(R+r)}{\sinh\left(\sqrt{-\kappa}(R+r)\right)}\tau r$$

and Lemma 7, we have $\mathbf{z}_t \in (1+\tau)\mathcal{K}$ and $\tilde{\mathbf{z}}_t \in \mathcal{K}$.

By the first item of Lemma 23,

$$\mathbb{E}\left[\left\langle \mathbf{g}_t, \mathrm{Exp}_{\tilde{\mathbf{y}}_t}^{-1}\mathbf{x}\right\rangle\right] = \mathbb{E}\left[\left\langle \mathbb{E}\left[\mathbf{g}_t|\tilde{\mathbf{y}}_t\right], \mathrm{Exp}_{\tilde{\mathbf{y}}_t}^{-1}\mathbf{x}\right\rangle\right] = \frac{V_{\delta'}}{S_{\delta'}}\mathbb{E}\left[\left\langle \nabla\hat{f}_t(\tilde{\mathbf{y}}_t), \mathrm{Exp}_{\tilde{\mathbf{y}}_t}^{-1}\mathbf{x}\right\rangle\right] \quad (17)$$

where $\hat{f}_t(\mathbf{x})$ is a smoothed version of $f_t(\mathbf{x})$. More specifically,

$$\hat{f}_t(\mathbf{x}) := \frac{1}{V_{\delta'}}\int_{\mathbb{B}_{\mathbf{x}}(\delta')} f_t(\mathbf{u})\omega$$

where $\omega$ is the volume form.

Applying Lemma 4 to Algorithm 4, we know $\tilde{\mathbf{y}}_t$ is an infeasible projection of $\mathbf{y}_t$ over $(1-\delta)\mathcal{K}$, which means

$$d(\tilde{\mathbf{y}}_{t+1}, \mathbf{x})^2 \leq d(\mathbf{y}_{t+1}, \mathbf{x})^2 \leq d(\tilde{\mathbf{y}}_t, \mathbf{x})^2 + \zeta\eta^2\|\mathbf{g}_t\|^2 - 2\eta\left\langle \mathbf{g}_t, -\mathrm{Exp}_{\tilde{\mathbf{y}}_t}^{-1}\mathbf{x}\right\rangle \quad (18)$$

holds for any $\mathbf{x} \in (1-\delta)\mathcal{K}$, where the second inequality is due to Lemma 29.

Equation (18) is equivalent to

$$\left\langle \mathbf{g}_t, -\mathrm{Exp}_{\tilde{\mathbf{y}}_t}^{-1}\mathbf{x}_t\right\rangle \leq \frac{d(\tilde{\mathbf{y}}_t, \mathbf{x})^2 - d(\tilde{\mathbf{y}}_{t+1}, \mathbf{x})^2}{2\eta} + \frac{\eta\zeta}{2}\|\mathbf{g}_t\|^2. \quad (19)$$

By the third item of Lemma 23, we have

$$\begin{aligned}
\hat{f}(\tilde{\mathbf{y}}_t) - \hat{f}(\mathbf{x}) &\leq \left\langle \nabla\hat{f}(\tilde{\mathbf{y}}_t), -\mathrm{Exp}_{\tilde{\mathbf{y}}_t}^{-1}\mathbf{x}\right\rangle + 2\delta'\rho G \\
&= \frac{S_{\delta'}}{V_{\delta'}}\mathbb{E}\left[\left\langle \mathbf{g}_t, -\mathrm{Exp}_{\tilde{\mathbf{y}}_t}^{-1}\mathbf{x}\right\rangle\right] + 2\delta'\rho G \\
&\leq \frac{S_{\delta'}}{V_{\delta'}}\cdot\frac{d(\tilde{\mathbf{y}}_t, \mathbf{x})^2 - d(\tilde{\mathbf{y}}_{t+1}, \mathbf{x})^2}{2\eta} + \frac{\eta\zeta}{2}\cdot\frac{S_{\delta'}}{V_{\delta'}}\mathbb{E}\left[\|\mathbf{g}_t\|^2\right] + 2\delta'\rho G.
\end{aligned} \quad (20)$$

where $\rho$ is a constant solely depends on $\mathcal{K}$.

Combining Equations (17), (19), and (20), we have

$$\mathbb{E}\left[\sum_{t=s}^{e}\left(\hat{f}_t(\tilde{\mathbf{y}}_t) - \hat{f}_t(\mathbf{x})\right)\right] \leq \frac{2R^2}{\eta}\cdot\frac{S_{\delta'}}{V_{\delta'}} + \frac{\eta\zeta S_{\delta'}}{2V_{\delta'}}\sum_{t=s}^{e}\mathbb{E}\left[\|\mathbf{g}_t\|^2\right] + 2\delta'\rho GT \quad (21)$$

holds for any $\mathbf{x} \in (1-\delta)\mathcal{K}$, where we use the fact that $d(\tilde{\mathbf{y}}_s, \mathbf{x}) \leq 2R$. We also need to bound $\mathbb{E}\left[f_t(\tilde{\mathbf{z}}_t) - \hat{f}_t(\tilde{\mathbf{y}}_t)\right]$ and $\mathbb{E}\left[\hat{f}_t(\tilde{\mathbf{x}}_I) - f_t(\mathbf{x}_I)\right]$. For the former term,

$$\begin{aligned}
\mathbb{E}\left[f_t(\tilde{\mathbf{z}}_t) - \hat{f}_t(\tilde{\mathbf{y}}_t)\right] &= \mathbb{E}\left[f_t(\tilde{\mathbf{z}}_t) - f_t(\mathbf{z}_t)\right] + \mathbb{E}\left[f_t(\mathbf{z}_t) - f_t(\tilde{\mathbf{y}}_t)\right] + \mathbb{E}\left[f_t(\tilde{\mathbf{y}}_t) - \hat{f}_t(\tilde{\mathbf{y}}_t)\right] \\
&\leq G\cdot d(\tilde{\mathbf{z}}_t, \mathbf{z}_t) + G\delta' + G\delta' \leq G\tau R + 2G\delta',
\end{aligned} \quad (22)$$

where we use Lemma 22 and the gradient Lipschitzness. Similarly, for the latter term,

$$\mathbb{E}\left[\hat{f}_t(\tilde{\mathbf{x}}_t) - f_t(\mathbf{x}_I)\right] = \mathbb{E}\left[\hat{f}_t(\tilde{\mathbf{x}}_I) - f_t(\tilde{\mathbf{x}}_I)\right] + \mathbb{E}\left[f_t(\tilde{\mathbf{x}}_I) - f_t(\mathbf{x}_I)\right]$$
$$\leq G\delta' + G \cdot d(\tilde{\mathbf{x}}_I, \mathbf{x}_I) \leq G\delta' + G\delta R. \tag{23}$$

By the second term of Lemma 23, we have

$$\frac{S_{\delta'}^2}{V_{\delta'}^2}\|\mathbf{g}_t\|^2 \leq \left(\frac{n}{\delta'} + n|\kappa|\right)^2 M^2. \tag{24}$$

Combining Equations (21), (22), (23) and (24), summing from $t = s$ to $e$, we get

$$\mathbb{E}\left[\sum_{t=s}^{e}(f_t(\widetilde{\mathbf{z}}_t) - f_t(\mathbf{x}_I))\right]$$

$$= \mathbb{E}\left[\sum_{t=s}^{e}\left(f_t(\widetilde{\mathbf{z}}_t) - \hat{f}_t(\tilde{\mathbf{y}}_t)\right)\right] + \mathbb{E}\left[\sum_{t=s}^{e}\left(\hat{f}_t(\tilde{\mathbf{y}}_t) - \hat{f}_t(\mathbf{x}_I)\right)\right] + \mathbb{E}\left[\sum_{t=s}^{e}\left(\hat{f}_t(\mathbf{x}_I) - f_t(\mathbf{x}_I)\right)\right]$$

$$\leq (G\tau R + 2G\delta')T + \left(\frac{2R^2}{\eta} \cdot \frac{S_{\delta'}}{V_{\delta'}} + \frac{\eta\zeta M^2}{2}\frac{S_{\delta'}}{V_{\delta'}}T + 2\rho\delta'GT\right) + (G\delta + G\delta R)T$$

$$\leq (G\tau R + 2G\delta')T + \left(\frac{2R^2}{\eta} \cdot \left(\frac{n}{\delta'} + n|\kappa|\right) + \frac{\eta\zeta M^2}{2}\left(\frac{n}{\delta'} + n|\kappa|\right)T + 2\rho\delta'GT\right) + (G\delta + G\delta R)T. \tag{25}$$

After plugging in $\eta = T^{-\frac{1}{2}}$, $\delta = T^{-\frac{1}{4}}$, $\tau = T^{-\frac{1}{4}}$ and $\delta' = \frac{\sqrt{-\kappa}(R+r)}{\sinh\left(\sqrt{-\kappa}(R+r)\right)}T^{-\frac{1}{4}}r$, we see $\delta' \leq T^{-\frac{1}{4}}r$ and

$$\frac{1}{\delta'} \leq \frac{\sinh\left(\sqrt{-\kappa}(R+r)\right)}{\sqrt{-\kappa}(R+r)} \cdot \frac{T^{\frac{1}{4}}}{r}.$$

Then

$$\mathbb{E}\left[\sum_{t=s}^{e}(f_t(\widetilde{\mathbf{z}}_t) - f_t(\mathbf{x}_I))\right] = O\left(T^{\frac{3}{4}}\right)$$

as claimed.

Denote $\mathbf{x}_t^* \in \operatorname{argmin}_{\mathbf{x}\in(1-\delta)\mathcal{K}} d(\mathbf{x}, \tilde{\mathbf{y}}_t)$. We now bound the number of calls to the separation oracle. Due to $\mathbf{y}_{t+1} = \operatorname{Exp}_{\tilde{\mathbf{y}}_t}(-\eta\mathbf{g}_t)$,

$$d(\mathbf{y}_{t+1}, (1-\delta)\mathcal{K}) \leq d(\mathbf{x}_t^*, \mathbf{y}_{t+1}) \leq d(\mathbf{x}_t^*, \tilde{\mathbf{y}}_t) + d(\tilde{\mathbf{y}}_t, \mathbf{y}_{t+1}) \leq d(\tilde{\mathbf{y}}_t, (1-\delta)\mathcal{K}) + \eta\|\mathbf{g}_t\|.$$

Squaring both sides, we have

$$d(\mathbf{y}_{t+1}, (1-\delta)\mathcal{K})^2 \leq d(\tilde{\mathbf{y}}_t, (1-\delta)\mathcal{K})^2 + 2d(\tilde{\mathbf{y}}_t, (1-\delta)\mathcal{K})\eta M + \eta^2 M^2$$
$$\leq d(\tilde{\mathbf{y}}_t, (1-\delta)\mathcal{K})^2 + 2\delta R\eta M + \eta^2 M^2, \tag{26}$$

where the second inequality is because

$$d(\tilde{\mathbf{y}}_t, (1-\delta)\mathcal{K}) \leq d(\tilde{\mathbf{y}}_t, \operatorname{Exp}_{\mathbf{p}}((1-\delta)\operatorname{Exp}_{\mathbf{p}}^{-1}\tilde{\mathbf{y}}_t)) = d(\tilde{\mathbf{y}}_t, \operatorname{Exp}_{\tilde{\mathbf{y}}_t}(\delta\operatorname{Exp}_{\tilde{\mathbf{y}}_t}^{-1}\mathbf{p})) = \delta d(\tilde{\mathbf{y}}_t, \mathbf{p}) \leq \delta R$$

and $\|\mathbf{g}_t\| = |f_t(\mathbf{z}_t)| \leq M$. Combining Equation (26) with Lemma 4, we can bound the number of separation oracle calls as:

$$N_{calls} \leq \sum_{t=1}^{T}\left(\zeta\frac{d(\mathbf{y}_{t+1}, (1-\delta)\mathcal{K})^2 - d(\tilde{\mathbf{y}}_{t+1}, (1-\delta)\mathcal{K})^2}{\delta^2\bar{r}^2} + 1\right)$$

$$\leq \sum_{t=1}^{T}\left(\zeta\frac{d(\tilde{\mathbf{y}}_t, (1-\delta)\mathcal{K})^2 + 2R\delta\eta M + \eta^2 M^2 - d(\tilde{\mathbf{y}}_{t+1}, (1-\delta)\mathcal{K})^2}{\delta^2\bar{r}^2} + 1\right)$$

$$\leq \frac{2RM\zeta\eta T}{\bar{r}^2\delta} + \frac{M^2\zeta\eta^2 T}{\bar{r}^2\delta^2} + T \tag{27}$$

$$= \frac{2RM\zeta T}{\bar{r}^2} \cdot \frac{\eta}{\delta} + \frac{M^2\zeta T}{\bar{r}^2} \cdot \frac{\eta^2}{\delta^2} + T$$

$$= T + \frac{2RM\zeta}{\bar{r}^2} \cdot T^{\frac{3}{4}} + \frac{M^2\zeta}{\bar{r}^2} \cdot T^{\frac{1}{2}}$$

where the last inequality follows from $\tilde{\mathbf{y}}_1 = \mathbf{p} \in (1-\delta)\mathcal{K}$ and thus $d(\tilde{\mathbf{y}}_1, (1-\delta)\mathcal{K}) = 0$. $\qquad\square$

## B.5 Proof of Theorem 3

*Proof.* We maintain the loop invariant $\mathbf{x}_t \in \beta\mathcal{K}$. So by Lemma 19, $\mathbb{B}(\mathbf{x}_t, \delta') \subseteq \mathcal{K}$, which means $\mathbf{z}_t$ and $\mathbf{z}'_t$ are feasible points.

For the two-point feedback setting, by Lemma 23,

$$
\begin{aligned}
\hat{f}_t(\mathbf{x}_t) - \hat{f}_t(\mathbf{x}) &\leq \left\langle \nabla\hat{f}_t(\mathbf{x}_t), -\mathrm{Exp}_{\mathbf{x}_t}^{-1}\mathbf{x} \right\rangle + 2\delta'\rho G \\
&= \frac{S_{\delta'}}{V_{\delta'}}\mathbb{E}\left[\left\langle \mathbf{g}_t, -\mathrm{Exp}_{\mathbf{x}_t}^{-1}\mathbf{x}\right\rangle\right] + 2\delta'\rho G \\
&= \frac{S_{\delta'}}{V_{\delta'}}\mathbb{E}\left[\left\langle \mathbf{g}_t, -\Gamma_{\mathbf{y}_t}^{\mathbf{x}_t}\mathrm{Exp}_{\mathbf{y}_t}^{-1}\mathbf{x}\right\rangle + \left\langle \mathbf{g}_t, \Gamma_{\mathbf{y}_t}^{\mathbf{x}_t}\mathrm{Exp}_{\mathbf{y}_t}^{-1}\mathbf{x} - \mathrm{Exp}_{\mathbf{x}_t}^{-1}\mathbf{x}\right\rangle\right] + 2\delta'\rho G.
\end{aligned}
\tag{28}
$$

By Lemma 17 of Wang et al. (2023),

$$
\frac{S_{\delta'}}{V_{\delta'}}\mathbb{E}\left[\|\mathbf{g}_t\|\right] \leq nG(1 + |\kappa|\delta'^2).
$$

Thus,

$$
\begin{aligned}
\frac{S_{\delta'}}{V_{\delta'}}\mathbb{E}\left[\left\langle \mathbf{g}_t, \Gamma_{\mathbf{y}_t}^{\mathbf{x}_t}\mathrm{Exp}_{\mathbf{y}_t}^{-1}\mathbf{x} - \mathrm{Exp}_{\mathbf{x}_t}^{-1}\mathbf{x}\right\rangle\right] &\leq \frac{S_{\delta'}}{V_{\delta'}}\mathbb{E}\left[\|\mathbf{g}_t\|\right] \cdot \zeta d(\mathbf{x}_t, \mathbf{y}_t) \\
\leq nG(1 + |\kappa|\delta'^2) \cdot \zeta(1-\beta)d(\mathbf{y}_t, \mathbf{p}) &\leq nGR\zeta(1-\beta)(1 + |\kappa|\delta'^2) \\
= nGR\zeta \cdot \frac{\sinh(\sqrt{-\kappa}(R+r))}{\sqrt{-\kappa}(R+r)} \cdot \frac{\delta'}{r}\left(1 + \|\kappa|\delta'^2\right) &= O(\delta'),
\end{aligned}
\tag{29}
$$

where we use the $\zeta$-smoothness of $\frac{1}{2}d(\mathbf{x}, \mathbf{y})^2$ and the definition of $\delta'$.

Since $\mathbf{y}_{t+1}$ is an infeasible projection of $\mathbf{y}'_{t+1}$ onto $(1-\delta)\mathcal{K}$,

$$
\begin{aligned}
d(\mathbf{y}_{t+1}, \mathbf{x})^2 &\leq d(\mathbf{y}'_{t+1}, \mathbf{x})^2 \leq d(\mathbf{y}_t, \mathbf{x})^2 + \zeta d(\mathbf{y}_t, \mathbf{y}'_{t+1})^2 - 2\left\langle \mathrm{Exp}_{\mathbf{y}_t}^{-1}\mathbf{y}'_{t+1}, \mathrm{Exp}_{\mathbf{y}_t}^{-1}\mathbf{x}\right\rangle \\
&= d(\mathbf{y}_t, \mathbf{x})^2 + \zeta\eta^2\|\mathbf{g}_t\|^2 - 2\left\langle -\eta\Gamma_{\mathbf{x}_t}^{\mathbf{y}_t}\mathbf{g}_t, \mathrm{Exp}_{\mathbf{y}_t}^{-1}\mathbf{x}\right\rangle \\
&= d(\mathbf{y}_t, \mathbf{x})^2 + \zeta\eta^2\|\mathbf{g}_t\|^2 - 2\eta\left\langle \mathbf{g}_t, -\Gamma_{\mathbf{y}_t}^{\mathbf{x}_t}\mathrm{Exp}_{\mathbf{y}_t}^{-1}\mathbf{x}\right\rangle
\end{aligned}
$$

holds for any $\mathbf{x} \in (1-\delta)\mathcal{K}$, which means

$$
\left\langle \mathbf{g}_t, -\Gamma_{\mathbf{y}_t}^{\mathbf{x}_t}\mathrm{Exp}_{\mathbf{y}_t}^{-1}\mathbf{x}\right\rangle \leq \frac{d(\mathbf{y}_t, \mathbf{x})^2 - d(\mathbf{y}_{t+1}, \mathbf{x})^2}{2\eta} + \frac{\eta\zeta\|\mathbf{g}_t\|^2}{2}
\tag{30}
$$

Taking expectation and using $\frac{S_{\delta'}}{V_{\delta'}} \leq \frac{n}{\delta'} + n|\kappa|$, we have

$$
\begin{aligned}
\frac{S_{\delta'}}{V_{\delta'}}\mathbb{E}\left[\left\langle \mathbf{g}_t, -\Gamma_{\mathbf{y}_t}^{\mathbf{x}_t}\mathrm{Exp}_{\mathbf{y}_t}^{-1}\mathbf{x}\right\rangle\right] &\leq \frac{S_{\delta'}}{2\eta V_{\delta'}}\left(d(\mathbf{y}_t, \mathbf{x})^2 - d(\mathbf{y}_{t+1}, \mathbf{x})^2\right) + \frac{\eta\zeta S_{\delta'}}{2V_{\delta'}}\mathbb{E}\left[\|\mathbf{g}_t\|^2\right] \\
&\leq \frac{1}{2\eta}\left(\frac{n}{\delta'} + n|\kappa|\right)\left(d(\mathbf{y}_t, \mathbf{x})^2 - d(\mathbf{y}_{t+1}, \mathbf{x})^2\right) + \left(\frac{n}{2\delta'} + \frac{n|\kappa|}{2}\right)\eta\zeta \cdot \delta'^2 G^2.
\end{aligned}
\tag{31}
$$

Combining Equations (28), (29), (30) and (31), and summing from $t = s$ to $e$, we have

$$
\mathbb{E}\left[\sum_{t=s}^{e}\left(\hat{f}_t(\mathbf{x}_t) - \hat{f}_t(\mathbf{x})\right)\right] = O\left(\frac{1}{\eta\delta'} + \delta'T + \eta\delta'T\right)
$$

holds for any $\mathbf{x} \in (1-\delta)\mathcal{K}$.

Following Equations (22) and (23), and taking $\mathbf{x} = \mathrm{Exp}_{\mathbf{p}}\left((1-\delta)\mathrm{Exp}_{\mathbf{p}}^{-1}\mathbf{x}^*\right)$, we have

$$
\mathbb{E}\left[f_t(\mathbf{z}_t) - \hat{f}_t(\mathbf{x}_t)\right] = O(\delta')
$$

and

$$
\mathbb{E}\left[\hat{f}_t(\mathbf{x}) - f_t(\mathbf{x}^*)\right] = O(\delta + \delta').
$$

In sum,
$$\sum_{t=s}^{e} \mathbb{E}\left[f_t(\mathbf{z}_t) - f_t(\mathbf{x}^*)\right] = O\left(\frac{1}{\eta\delta'} + (\delta + \delta')T + \eta\delta'T\right).$$

By choosing $\eta = 1, \delta = 1 - \beta = T^{-\frac{1}{2}}$, then $\delta' = O\left(T^{-\frac{1}{2}}\right)$ and we can get $O(\sqrt{T})$ regret.

Denote $\mathbf{x}_t^* \in \operatorname{argmin}_{\mathbf{x}\in(1-\delta)\mathcal{K}} d(\mathbf{x}, \mathbf{y}_t)$ and we again bound the number of calls to the separation oracle. Due to $\mathbf{y}_{t+1}' = \operatorname{Exp}_{\mathbf{y}_t}\left(-\eta\Gamma_{\mathbf{x}_t}^{\mathbf{y}_t}\mathbf{g}_t\right)$,

$$d(\mathbf{y}_{t+1}', (1-\delta)\mathcal{K}) \le d(\mathbf{x}_t^*, \mathbf{y}_{t+1}') \le d(\mathbf{x}_t^*, \mathbf{y}_t) + d(\mathbf{y}_t, \mathbf{y}_{t+1}')$$
$$\le d(\tilde{\mathbf{y}}_t, (1-\delta)\mathcal{K}) + \eta\|\mathbf{g}_t\| \le d(\tilde{\mathbf{y}}_t, (1-\delta)\mathcal{K}) + \eta\delta'G$$

where we use $\|\mathbf{g}_t\| = \frac{1}{2}|f_t(\mathbf{z}_t) - f_t(\tilde{\mathbf{z}}_t)| \le \delta'G$ by Assumption 4. Squaring both sides, we have

$$d(\mathbf{y}_{t+1}', (1-\delta)\mathcal{K})^2 \le d(\mathbf{y}_t, (1-\delta)\mathcal{K})^2 + 2d(\mathbf{y}_t, (1-\delta)\mathcal{K})\eta\delta'G + \eta^2\delta'^2G^2$$
$$\le d(\mathbf{y}_t, (1-\delta)\mathcal{K})^2 + 2\delta R\eta\delta'G + \eta^2\delta'^2G^2, \tag{32}$$

where the second inequality is again due to $\mathbf{y}_t \in \mathcal{K}$ and

$$d(\mathbf{y}_t, (1-\delta)\mathcal{K}) \le d(\mathbf{y}_t, \operatorname{Exp}_{\mathbf{p}}((1-\delta)\operatorname{Exp}_{\mathbf{p}}^{-1}\mathbf{y}_t)) = d(\mathbf{y}_t, \operatorname{Exp}_{\mathbf{y}_t}(\delta\operatorname{Exp}_{\mathbf{y}_t}^{-1}\mathbf{p})) = \delta d(\mathbf{y}_t, \mathbf{p}) \le \delta R.$$

Combining Equation (32) with Lemma 4, we can bound the number of separation oracle calls as:

$$N_{calls} \le \sum_{t=1}^{T}\left(\zeta\frac{d(\mathbf{y}_{t+1}', (1-\delta)\mathcal{K})^2 - d(\mathbf{y}_{t+1}, (1-\delta)\mathcal{K})^2}{\delta^2\bar{r}^2} + 1\right)$$
$$\le \sum_{t=1}^{T}\left(\zeta\frac{d(\mathbf{y}_t, (1-\delta)\mathcal{K})^2 + 2R\delta\eta\delta'G + \eta^2\delta'^2G^2 - d(\mathbf{y}_{t+1}, (1-\delta)\mathcal{K})^2}{\delta^2\bar{r}^2} + 1\right) \tag{33}$$
$$= \zeta \cdot \frac{2\delta'\eta RG}{\delta\bar{r}^2} + \zeta \cdot \frac{\eta^2\delta'^2G^2}{\delta^2\bar{r}^2}T + T$$

where the last inequality follows from $\tilde{\mathbf{y}}_1 = \mathbf{p} \in (1-\delta)\mathcal{K}$ and thus $d(\tilde{\mathbf{y}}_1, (1-\delta)\mathcal{K}) = 0$. Since $\eta = 1, \delta = 1 - \beta = T^{-\frac{1}{2}}$ and

$$\delta' = (1-\beta)\frac{\sqrt{-\kappa}(R+r)}{\sinh\left(\sqrt{-\kappa}(R+r)\right)}r = \delta\frac{\sqrt{-\kappa}(R+r)}{\sinh\left(\sqrt{-\kappa}(R+r)\right)}r \le \delta r.$$

Plugging these parameters into Equation (33), we have

$$N_{calls} \le \zeta \cdot \frac{2rRGT}{\bar{r}^2} + \zeta \cdot \frac{r^2G^2T}{\bar{r}^2} + T = O(T).$$

$\square$

## C   Omitted Proofs in Section 6

We have the following convergence guarantee for RFW.

### C.1   Proof of Lemma 5

*Proof.* Since Algorithm 6 is indeed RFW with line-search (Algorithm 9) when $f(\mathbf{x}) = d(\mathbf{x}, \mathbf{y})^2/2$, which is $\zeta$ gsc-smooth by Lemma 27 and $\nabla f(\mathbf{x}) = -\operatorname{Exp}_{\mathbf{x}}^{-1}\mathbf{y}$, the upper bound on the number of iterations follows from Lemma 26 with $L = \zeta$ and $D = 2R$. Item 1 follows from the line-search because $f(\mathbf{x}_i) = d(\mathbf{x}_i, \mathbf{y})^2/2$ does not increase as $i$ increases.

For item 2, Algorithm 6 stops when either $d(\tilde{\mathbf{x}}, \mathbf{y})^2 \le 3\epsilon$ or $d(\tilde{\mathbf{x}}, \mathbf{y})^2 > 3\epsilon$ and $\langle\operatorname{Exp}_{\tilde{\mathbf{x}}}^{-1}\mathbf{y}, \operatorname{Exp}_{\tilde{\mathbf{x}}}^{-1}\mathbf{x}\rangle \le \epsilon$. For the first case, item 2 obviously holds. Now we consider the second case. We first show $\langle\operatorname{Exp}_{\mathbf{y}}^{-1}\mathbf{z}, \operatorname{Exp}_{\mathbf{y}}^{-1}\tilde{\mathbf{x}}\rangle + \langle\operatorname{Exp}_{\tilde{\mathbf{x}}}^{-1}\mathbf{z}, \operatorname{Exp}_{\tilde{\mathbf{x}}}^{-1}\mathbf{y}\rangle \ge d(\tilde{\mathbf{x}}, \mathbf{y})^2$ as follows. By Lemma 30, we have

$$\langle\operatorname{Exp}_{\mathbf{y}}^{-1}\mathbf{z}, \operatorname{Exp}_{\mathbf{y}}^{-1}\tilde{\mathbf{x}}\rangle \ge \frac{1}{2}(d(\mathbf{y}, \mathbf{z})^2 + d(\mathbf{y}, \tilde{\mathbf{x}})^2 - d(\tilde{\mathbf{x}}, \mathbf{z})^2)$$

$$\langle\operatorname{Exp}_{\tilde{\mathbf{x}}}^{-1}\mathbf{z}, \operatorname{Exp}_{\tilde{\mathbf{x}}}^{-1}\mathbf{y}\rangle \ge \frac{1}{2}(d(\tilde{\mathbf{x}}, \mathbf{y})^2 + d(\tilde{\mathbf{x}}, \mathbf{z})^2 - d(\mathbf{y}, \mathbf{z})^2).$$

Adding the above two inequalities we have $\langle \mathrm{Exp}_{\mathbf{y}}^{-1}\mathbf{z}, \mathrm{Exp}_{\mathbf{y}}^{-1}\tilde{\mathbf{x}}\rangle + \langle \mathrm{Exp}_{\tilde{\mathbf{x}}}^{-1}\mathbf{z}, \mathrm{Exp}_{\tilde{\mathbf{x}}}^{-1}\mathbf{y}\rangle \geq d(\tilde{\mathbf{x}}, \mathbf{y})^2$. Now item 2 follows from

$$\langle \mathrm{Exp}_{\mathbf{y}}^{-1}\mathbf{z}, \mathrm{Exp}_{\mathbf{y}}^{-1}\tilde{\mathbf{x}}\rangle \geq d(\tilde{\mathbf{x}}, \mathbf{y})^2 - \langle \mathrm{Exp}_{\tilde{\mathbf{x}}}^{-1}\mathbf{z}, \mathrm{Exp}_{\tilde{\mathbf{x}}}^{-1}\mathbf{y}\rangle > 3\epsilon - \langle \mathrm{Exp}_{\tilde{\mathbf{x}}}^{-1}\mathbf{v}_i, \mathrm{Exp}_{\tilde{\mathbf{x}}}^{-1}\mathbf{y}\rangle > 2\epsilon.$$

For the third item, denote $\mathbf{x}^* = \mathrm{argmin}_{\mathbf{x}\in\mathcal{K}} d(\mathbf{x}, \mathbf{y})^2$, Suppose $d(\mathbf{y}, \mathcal{K})^2 = d(\mathbf{x}^*, \mathbf{y})^2 < \epsilon$ and $d(\tilde{\mathbf{x}}, \mathbf{y})^2 > 3\epsilon$. Denote $f(\mathbf{x}) = d(\mathbf{x}, \mathbf{y})^2/2$ and $\nabla f(\mathbf{x}) = -\mathrm{Exp}_{\mathbf{x}}^{-1}\mathbf{y}$. For the last iteration,

$$\langle \mathrm{Exp}_{\tilde{\mathbf{x}}}^{-1}\mathbf{v}_i, \mathrm{Exp}_{\tilde{\mathbf{x}}}^{-1}\mathbf{y}\rangle = \max_{\mathbf{v}\in\mathcal{K}} \langle \mathrm{Exp}_{\tilde{\mathbf{x}}}^{-1}\mathbf{v}, -\nabla f(\tilde{\mathbf{x}})\rangle \leq \epsilon,$$

which implies

$$d(\tilde{\mathbf{x}}, \mathbf{y})^2 - d(\mathbf{y}, \mathcal{K})^2 = 2f(\tilde{\mathbf{x}}) - 2f(\mathbf{x}^*) \leq 2\langle -\mathrm{Exp}_{\tilde{\mathbf{x}}}^{-1}\mathbf{x}^*, \nabla f(\tilde{\mathbf{x}})\rangle$$
$$\leq 2\max_{\mathbf{v}\in\mathcal{K}} \langle -\mathrm{Exp}_{\tilde{\mathbf{x}}}^{-1}\mathbf{v}, \nabla f(\tilde{\mathbf{x}})\rangle \leq 2\epsilon.$$

And thus

$$d(\tilde{\mathbf{x}}, \mathbf{y})^2 \leq d(\mathbf{y}, \mathcal{K})^2 + 2\epsilon \leq 3\epsilon$$

as claimed. □

### C.2 Proof of Lemma 6

Before proving Lemma 6, we need the following lemma.

**Lemma 8.** *Consider Algorithm 7 and fix some $\epsilon$ such that $0 < 3\epsilon < d(\mathbf{x}_0, \mathbf{y}_0)^2$. Setting $\gamma = \frac{2\epsilon}{d(\mathbf{x}_0,\mathbf{y}_0)^2}$, we have $d(\mathbf{x}_i, \mathbf{y}_i) \leq d(\mathbf{x}_0, \mathbf{y}_0)$ for any $i$.*

*Proof.* For all $i > 1$, we define the sequence $\mathbf{y}_i$ by the relation $\mathbf{y}_i = \mathrm{Exp}_{\mathbf{x}_{i-1}}((1-\gamma)\mathrm{Exp}_{\mathbf{x}_{i-1}}^{-1}\mathbf{y}_{i-1})$, and $\gamma \in [0, 1)$. From this, we can deduce the following sequence of inequalities:

$$d(\mathbf{x}_{i-1}, \mathbf{y}_i) = \|\mathrm{Exp}_{\mathbf{x}_{i-1}}^{-1}\mathbf{y}_i\|$$
$$= (1-\gamma)\|\mathrm{Exp}_{\mathbf{x}_{i-1}}^{-1}\mathbf{y}_{i-1}\|$$
$$\leq d(\mathbf{x}_{i-1}, \mathbf{y}_{i-1}).$$

The inequality follows since $\gamma \in (0, 1)$.

By the first item of Lemma 5, for any $i \geq 1$, we have $d(\mathbf{x}_i, \mathbf{y}_i) \leq d(\mathbf{x}_{i-1}, \mathbf{y}_i)$. Combining this with the previous inequality, we get

$$d(\mathbf{x}_i, \mathbf{y}_i) \leq d(\mathbf{x}_{i-1}, \mathbf{y}_{i-1})$$
$$\leq \ldots$$
$$\leq d(\mathbf{x}_1, \mathbf{y}_1)$$
$$\leq d(\mathbf{x}_0, \mathbf{y}_1)$$
$$\leq d(\mathbf{x}_0, \mathbf{y}_0),$$

where the last inequality follows from the fact that $\mathbf{x}_0 \in \mathcal{K}$ and $\mathbf{y}_1$ is the projection of $\mathbf{y}_0$ onto $\mathbb{B}_{\mathbf{p}}(R)$, while $\mathcal{K} \subseteq \mathbb{B}_{\mathbf{p}}(R)$. □

*Proof of Lemma 6.* Given that $\mathbf{y}_1$ is the projection of $\mathbf{y}_0$ onto $\mathbb{B}_{\mathbf{p}}(R)$ and $\mathcal{K} \subseteq \mathbb{B}_{\mathbf{p}}(R)$, we have $\forall \mathbf{z} \in \mathcal{K} : d(\mathbf{y}_1, \mathbf{z})^2 \leq d(\mathbf{y}_0, \mathbf{z})^2$. The lemma trivially holds when $d(\mathbf{x}_0, \mathbf{y}_0)^2 \leq 3\epsilon$ or $d(\mathbf{x}_1, \mathbf{y}_1)^2 \leq 3\epsilon$.

Without loss of generality, we assume that $d(\mathbf{x}_1, \mathbf{y}_1) > 3\epsilon$. Let $k > 1$ be the number of iterations in Algorithm 7. This means that $d(\mathbf{x}_k, \mathbf{y}_k)^2 \leq 3\epsilon$ and $d(\mathbf{x}_i, \mathbf{y}_i)^2 > 3\epsilon$ for any $i < k$. According to the second item of Lemma 5, we obtain $\langle \mathrm{Exp}_{\mathbf{y}_i}^{-1}\mathbf{z}, \mathrm{Exp}_{\mathbf{y}_i}^{-1}\mathbf{x}_i\rangle \geq 2\epsilon$ for all $i < k$ and $\mathbf{z} \in \mathcal{K}$. Lemma 8 also gives us $d(\mathbf{x}_i, \mathbf{y}_i) \leq d(\mathbf{x}_0, \mathbf{y}_0)$ for all $i < k$.

Applying Lemma 2 with $\mathbf{g} = -\text{Exp}_{\mathbf{y}_i}^{-1}\mathbf{x}_i$, $C = d(\mathbf{x}_0, \mathbf{y}_0)$, and $Q = 2\epsilon$, we deduce that for every $1 \le i < k$,

$$\forall \mathbf{z} \in \mathcal{K}: \quad d(\mathbf{z}, \mathbf{y}_{i+1})^2 \le d(\mathbf{z}, \mathbf{y}_i)^2 - \frac{4\epsilon^2}{\zeta d(\mathbf{x}_i, \mathbf{y}_i)^2} \le d(\mathbf{z}, \mathbf{y}_i)^2 - \frac{4\epsilon^2}{\zeta d(\mathbf{x}_0, \mathbf{y}_0)^2}. \tag{34}$$

Consequently, for any $\mathbf{z} \in \mathcal{K}$, the returned point $\mathbf{y}$ satisfies $d(\mathbf{y}, \mathbf{z})^2 \le d(\mathbf{y}_1, \mathbf{z})^2 \le d(\mathbf{y}_0, \mathbf{z})^2$. Because $\mathbf{p} \in \mathcal{K}$, we have $d(\mathbf{y}, \mathbf{p}) \le d(\mathbf{y}_1, \mathbf{p}) \le R$, which implies $\mathbf{y} \in \mathbb{B}_{\mathbf{p}}(R)$. Note that $\mathbf{x} \in \mathcal{K}$ as it is the output of Algorithm 6. Thus, we know Algorithm 7 returns $(\mathbf{x}, \mathbf{y}) \in \mathcal{K} \times \mathbb{B}_{\mathbf{p}}(R)$.

Now we bound the number of iterations in Algorithm 7. Denote $\mathbf{x}_i^* = \arg\min_{\mathbf{x} \in \mathcal{K}} d(\mathbf{x}, \mathbf{y}_i)^2$ for any $i < k$. Using Equation (34),

$$d(\mathbf{y}_{i+1}, \mathcal{K})^2 = d(\mathbf{x}_{i+1}^*, \mathbf{y}_{i+1})^2 \le d(\mathbf{x}_i^*, \mathbf{y}_{i+1})^2$$

$$\le d(\mathbf{x}_i^*, \mathbf{y}_i)^2 - \frac{4\epsilon^2}{\zeta d(\mathbf{x}_0, \mathbf{y}_0)^2} = d(\mathbf{y}_i, \mathcal{K})^2 - \frac{4\epsilon^2}{\zeta d(\mathbf{x}_0, \mathbf{y}_0)^2}.$$

By applying the inequality $d(\mathbf{y}_1, \mathcal{K})^2 \le d(\mathbf{y}_0, \mathcal{K})^2$ and subsequently performing a telescoping sum, we can demonstrate that

$$d(\mathbf{y}_{i+1}, \mathcal{K})^2 \le d(\mathbf{y}_1, \mathcal{K})^2 - \frac{4i\epsilon^2}{\zeta d(\mathbf{x}_0, \mathbf{y}_0)^2} \le d(\mathbf{y}_0, \mathcal{K})^2 - \frac{4i\epsilon^2}{\zeta d(\mathbf{x}_0, \mathbf{y}_0)^2} \le d(\mathbf{y}_0, \mathbf{x}_0)^2 - \frac{4i\epsilon^2}{\zeta d(\mathbf{x}_0, \mathbf{y}_0)^2}.$$

After at most $k - 1 = (\zeta d(\mathbf{x}_0, \mathbf{y}_0)^2(d(\mathbf{x}_0, \mathbf{y}_0)^2 - \epsilon))/4\epsilon^2$ iterations, we have $d(\mathbf{y}_k, \mathcal{K})^2 \le \epsilon$. By the third item of Lemma 5, the next iteration will be the last one, and the returned points $\mathbf{x}, \mathbf{y}$ satisfy $d(\mathbf{x}, \mathbf{y})^2 \le 3\epsilon$, as claimed. $\qquad\square$

### C.3 Proof of Theorem 4

We first prove Lemma 9, which bounds the regret of the infeasible projection for gsc-convex losses.

**Lemma 9.** *With some fixed block size $B$, $\eta_i = \eta$ and $\epsilon_i = \epsilon$ for any $i = 1, \ldots, \frac{T}{B}$. Assume all losses are gsc-convex. We use $i(t)$ to denote the block index of round $t$. Then the regret of playing $\tilde{\mathbf{y}}_{i(t)-1}$ as defined in Algorithm 8 can be bounded by*

$$\sup_{I=[s,e]\subseteq[T]} \left\{ \sum_{t=s}^{e} f_t(\tilde{\mathbf{y}}_{i(t)-1}) - \min_{\mathbf{x}_I \in \mathcal{K}} \sum_{t=s}^{e} f_t(\mathbf{x}_I) \right\} \le \frac{\zeta \eta B G^2 T}{2} + 4RKG + \frac{4R^2}{\eta}.$$

*Proof.* For convenience, we assume $B$ divides $T$. Let $\mathcal{T}_i = \{(i-1)B + 1, \ldots, iB\}$ be the set of rounds in the $i$-th block. By Lemma 6, $\tilde{\mathbf{y}}_{i+1}$ is the infeasible projection of $\mathbf{y}_{iB+1}$ onto $\mathcal{K}$:

$$d(\tilde{\mathbf{y}}_{i+1}, \mathbf{x})^2 \le d(\mathbf{y}_{iB+1}, \mathbf{x})^2$$

holds for any $\mathbf{x} \in \mathcal{K}$. By Algorithm 8, we have

$$\mathbf{y}_{iB+1} = \text{Exp}_{\tilde{\mathbf{y}}_{i-1}}\left(-\eta \sum_{s=1}^{B} \nabla f_{(i-1)B+s}(\tilde{\mathbf{y}}_{i-1})\right) = \text{Exp}_{\tilde{\mathbf{y}}_{i-1}}\left(-\eta \sum_{t \in \mathcal{T}_i} \nabla f_t(\tilde{\mathbf{y}}_{i(t)-1})\right).$$

Thus

$$d(\tilde{\mathbf{y}}_{i+1}, \mathbf{x})^2 \le d(\mathbf{y}_{iB+1}, \mathbf{x})^2$$

$$\le d(\tilde{\mathbf{y}}_{i-1}, \mathbf{x})^2 + \zeta \eta^2 B^2 G^2 - 2\eta \sum_{t \in \mathcal{T}_i} \left\langle \nabla f_t(\tilde{\mathbf{y}}_{i-1}), -\text{Exp}_{\tilde{\mathbf{y}}_{i-1}}^{-1}\mathbf{x} \right\rangle$$

where the last inequality is due to Lemma 29 and $\|\nabla f_t(\mathbf{x})\| \le G$. This is equivalent to

$$\sum_{t \in \mathcal{T}_i} \left\langle \nabla f_t(\tilde{\mathbf{y}}_{i-1}), -\text{Exp}_{\tilde{\mathbf{y}}_{i-1}}^{-1}\mathbf{x} \right\rangle \le \frac{d(\tilde{\mathbf{y}}_{i-1}, \mathbf{x})^2 - d(\tilde{\mathbf{y}}_{i+1}, \mathbf{x})^2}{2\eta} + \frac{\zeta \eta B^2 G^2}{2}. \tag{35}$$

For a certain interval $[s, e] \subseteq [T]$, let $i_s$ and $i_e$ be the smallest and the largest block indices such that $\mathcal{T}_{i_s}$ and $\mathcal{T}_{i_e}$ are fully contained in $[s, e]$. Then we have

$$
\begin{aligned}
\sum_{t=s}^{e} \left\langle \nabla f_t(\tilde{\mathbf{y}}_{i(t)-1}), -\mathrm{Exp}^{-1}_{\tilde{\mathbf{y}}_{i(t)-1}} \mathbf{x} \right\rangle &\leq \sum_{t=s}^{(i_s-1)B} \left\langle \nabla f_t(\tilde{\mathbf{y}}_{i_s-2}), -\mathrm{Exp}^{-1}_{\tilde{\mathbf{y}}_{i_s-2}} \mathbf{x} \right\rangle \\
&+ \sum_{i=i_s}^{i_e} \left\langle \nabla f_t(\tilde{\mathbf{y}}_{i-1}), -\mathrm{Exp}^{-1}_{\tilde{\mathbf{y}}_{i-1}} \mathbf{x} \right\rangle + \sum_{t=i_eB+1}^{e} \left\langle \nabla f_t(\tilde{\mathbf{y}}_{i_e}), -\mathrm{Exp}^{-1}_{\tilde{\mathbf{y}}_{i_e}} \right\rangle .
\end{aligned}
\tag{36}
$$

There are some undesirable terms in blocks $i_s - 1$ and $i_e + 1$, and we can bound them by noticing

$$
\left\langle \nabla f_t(\tilde{\mathbf{y}}_{i(t)-1}), -\mathrm{Exp}^{-1}_{\tilde{\mathbf{y}}_{i(t)-1}} \mathbf{x} \right\rangle \leq \|\nabla f_t(\tilde{\mathbf{y}}_{i(t)-1})\| \cdot \|\mathrm{Exp}^{-1}_{\tilde{\mathbf{y}}_{i(t)-1}} \mathbf{x}\| \leq G \cdot 2R = 2GR. \tag{37}
$$

Combining Equations (35), (36) and (37), we have

$$
\begin{aligned}
\sum_{t=s}^{e} \left( f_t(\tilde{\mathbf{y}}_{i(t)-1}) - f_t(\mathbf{x}) \right) &\leq \sum_{t=s}^{e} \left\langle \nabla f_t(\tilde{\mathbf{y}}_{i(t)-1}), -\mathrm{Exp}^{-1}_{\tilde{\mathbf{y}}_{i(t)-1}} \mathbf{x} \right\rangle \\
&\leq \sum_{i=i_s}^{i_e} \left( \frac{d(\tilde{\mathbf{y}}_{i-1}, \mathbf{x})^2 - d(\tilde{\mathbf{y}}_{i+1}, \mathbf{x})^2}{2\eta} + \frac{\zeta\eta B^2 G^2}{2} \right) + 2GR \cdot 2B \\
&\leq \frac{4R^2}{\eta} + \frac{\zeta\eta BG^2 T}{2} + 4BGR.
\end{aligned}
\tag{38}
$$

where the last inequality is due to the number of blocks in $[s, e]$ is upper bounded by $\frac{T}{B}$. $\qquad \square$

Now we give the proof of Theorem 4.

*Proof of Theorem 4.* We still use $i(t)$ to denote the block index of the $t$-th round. For an interval $[s, e] \subseteq [T]$, we can do the following decomposition

$$
\begin{aligned}
&\sum_{t=s}^{e} \left( f_t(\mathbf{x}_{i(t)-1}) - f_t(\mathbf{x}) \right) \\
&= \sum_{t=s}^{e} \left( f_t(\mathbf{x}_{i(t)-1}) - f_t(\tilde{\mathbf{y}}_{i(t)-1}) \right) + \sum_{t=s}^{e} \left( f_t(\tilde{\mathbf{y}}_{i(t)-1}) - f_t(\mathbf{x}) \right) \\
&\leq \sum_{t=s}^{e} \left\langle \nabla f_t(\mathbf{x}_{i(t)-1}), -\mathrm{Exp}^{-1}_{\mathbf{x}_{i(t)-1}} \tilde{\mathbf{y}}_{i(t)-1} \right\rangle + \sum_{t=s}^{e} \left( f_t(\tilde{\mathbf{y}}_{i(t)-1}) - f_t(\mathbf{x}) \right)
\end{aligned}
\tag{39}
$$

Algorithm 8 calls Algorithm 7 in each block to compute an infeasible projection. By Lemma 6 and Algorithm 8, $(\mathbf{x}_i, \tilde{\mathbf{y}}_i) \in \mathcal{K} \times \mathbb{B}_{\mathbf{p}}(R)$, which satisfies $d(\mathbf{x}_i, \tilde{\mathbf{y}}_i)^2 \leq 3\epsilon_i = 3\epsilon$. Thus

$$
\left\langle \nabla f_t(\mathbf{x}_{i(t)-1}), -\mathrm{Exp}^{-1}_{\mathbf{x}_{i(t)-1}} \tilde{\mathbf{y}}_{i(t)-1} \right\rangle \leq Gd(\mathbf{x}_{i(t)-1}, \tilde{\mathbf{y}}_{i(t)}) \leq \sqrt{3\epsilon} G. \tag{40}
$$

Combining Equations (39), (40) and Lemma 9, we have

$$
\sum_{t=s}^{e} \left( f_t(\mathbf{x}_{i(t)-1}) - f_t(\mathbf{x}) \right) \leq 4RGB + \frac{4R^2}{\eta} + \frac{\zeta\eta BG^2 T}{2} + GT\sqrt{3\epsilon}
$$

holds for any $\mathbf{x} \in \mathcal{K}$. By plugging in $\eta$, $\epsilon$, and $B$, we get the claimed regret bound.

We also need to bound the number of calls to the linear optimization oracle. Note that $d(\mathbf{x}_1, \tilde{\mathbf{y}}_1) = 0$ due to the initialization, and for any $i \geq 2$, we have $d(\mathbf{x}_i, \tilde{\mathbf{y}}_i)^2 \leq 3\epsilon$ due to Lemma 6. By Algorithm 8, we have

$$
\mathbf{y}_{iB+1} = \mathrm{Exp}_{\tilde{\mathbf{y}}_{i-1}} \left( -\eta \sum_{t=(i-1)B+1}^{iB} \nabla f_t(\tilde{\mathbf{y}}_{i-1}) \right) \qquad \forall i = 1, \ldots, \frac{T}{B}.
$$

Therefore
$$d(\mathbf{x}_{i-1}, \mathbf{y}_{iB+1}) \le d(\mathbf{x}_{i-1}, \tilde{\mathbf{y}}_{i-1}) + d(\tilde{\mathbf{y}}_{i-1}, \mathbf{y}_{iB+1}) \le \sqrt{3\epsilon} + \eta BG. \tag{41}$$
Squaring on both sides, we have

$$d(\mathbf{x}_{i-1}, \mathbf{y}_{iB+1})^2 \le \left(\sqrt{3\epsilon} + \eta BG\right)^2 \le 6\epsilon + 2B^2 G^2 \eta^2, \tag{42}$$

where $(a + b)^2 \le 2(a^2 + b^2)$ is used.

By Lemma 6, on the $i$-th block, Algorithm 7 stops after at most

$$\max\left\{\frac{\zeta d(\mathbf{x}_{i-1}, \mathbf{y}_{iB+1})^2 \left(d(\mathbf{x}_{i-1}, \mathbf{y}_{iB+1})^2 - \epsilon\right)}{4\epsilon^2} + 1, 1\right\}$$

iterations. According to Lemma 5, each iteration calls the linear optimization oracle for at most $\zeta\lceil\frac{27R^2}{\epsilon^2} - 2\rceil$ times. Thus in the $i$-th block, the number of calls to the linear optimization oracle is bounded by

$$N_{calls} := \max\left\{\frac{\zeta d(\mathbf{x}_{i-1}, \mathbf{y}_{iB+1})^2 \left(d(\mathbf{x}_{i-1}, \mathbf{y}_{iB+1})^2 - \epsilon\right)}{4\epsilon^2} + 1, 1\right\} \cdot \frac{27\zeta R^2}{\epsilon}.$$

There are $\frac{T}{B}$ blocks in total, so the total number of calls to the linear optimization oracle is at most

$$N_{calls} \le \frac{T}{B}\left(1 + \frac{15}{2}\zeta + \left(\frac{11B^2 G^2 \eta^2}{2\epsilon} + \frac{B^4 G^4 \eta^4}{\epsilon^2}\right)\zeta\right) \cdot \frac{27\zeta R^2}{\epsilon}.$$

Plugging in the values of $B, \eta$ and $\epsilon$, we find $N_{calls} \le T$ as claimed. $\qquad\square$

## C.4   Proof of Theorem 5

*Proof.* Based on a similar argument as in Equation (35), we have

$$
\begin{aligned}
&\sum_{i=1}^{\frac{T}{B}} \sum_{t \in \mathcal{T}_i} (f_t(\tilde{\mathbf{y}}_{i-1}) - f_t(\mathbf{x})) \\
&\le \sum_{i=1}^{\frac{T}{B}} \left(\sum_{t \in \mathcal{T}_i} \left\langle \nabla f_t(\tilde{\mathbf{y}}_{i-1}), -\mathrm{Exp}_{\tilde{\mathbf{y}}_{i-1}}^{-1}\mathbf{x}\right\rangle - \frac{\alpha B}{2} d(\tilde{\mathbf{y}}_{i-1}, \mathbf{x})^2\right) \\
&\le \sum_{i=1}^{\frac{T}{B}} \left(\frac{d(\tilde{\mathbf{y}}_{i-1}, \mathbf{x})^2 - d(\tilde{\mathbf{y}}_{i+1}, \mathbf{x})^2}{2\eta_i} + \frac{\zeta\eta B^2 G^2}{2} - \frac{\alpha B}{2} d(\tilde{\mathbf{y}}_{i-1}, \mathbf{x})^2\right) \\
&\le \sum_{i=1}^{\frac{T}{B}} \frac{\zeta BG^2}{\alpha i},
\end{aligned}
\tag{43}
$$

where we apply the strong gsc-convexity on the $i$-th block under the help of Lemma 28; the last inequality follows from the definition of $\eta_i$. If we consider the sum of regret over each block as in Equation (43),

$$\sum_{t=1}^{T} \left(f_t(\tilde{\mathbf{y}}_{i(t)-1}) - f_t(\mathbf{x})\right) \le \frac{\zeta BG^2}{\alpha} \sum_{i=1}^{\frac{T}{B}} \frac{1}{i}. \tag{44}$$

On the other hand, since $f_t(\cdot)$ is $G$-Lipschitz, by Lemma 6,

$$
\begin{aligned}
&\sum_{t=1}^{T} \left(f_t(\mathbf{x}_{i(t)-1}) - f_t(\tilde{\mathbf{y}}_{i(t)-1})\right) \\
&\le \sum_{i=1}^{\frac{T}{B}} \sum_{t \in \mathcal{T}_i} \left\langle \nabla f_t(\mathbf{x}_{i(t)-1}), -\mathrm{Exp}_{\mathbf{x}_{i(t)-1}}^{-1}\tilde{\mathbf{y}}_{i(t)-1}\right\rangle \le \sqrt{3}BG \sum_{i=2}^{\frac{T}{B}} \sqrt{\epsilon_{i-1}},
\end{aligned}
\tag{45}
$$

where we use the fact that for the first block, $i(t) = 1$ and thus $\mathbf{x}_{i(t)-1} = \tilde{\mathbf{y}}_{i(t)-1}$. Now as we combine Equations (44) and (45), we have

$$\sum_{t=1}^{T} \left( f_t(\mathbf{x}_{i(t)-1}) - f_t(\mathbf{x}) \right) \leq \sqrt{3}BG \sum_{i=1}^{\frac{T}{B}} \sqrt{\epsilon_m} + \frac{\zeta BG^2}{\alpha} \sum_{i=1}^{\frac{T}{B}} \frac{1}{i}.$$

By plugging in $\{\epsilon_i\}_{i=1}^{\frac{T}{B}}$, we get the stated regret guarantee.

Now it remains to bound the number of calls to the linear optimization oracle. Note that for each block $i \in \left[\frac{T}{B}\right]$, by Lemma 6, $d(\mathbf{x}_i, \tilde{\mathbf{y}}_i)^2 \leq 3\epsilon_i$. In view of Algorithm 8, we also have

$$\mathbf{y}_{(i+1)K+1} = \mathrm{Exp}_{\tilde{\mathbf{y}}_i} \left( -\eta_{i+1} \sum_{t=iB+1}^{(i+1)B} \nabla f_t(\tilde{\mathbf{y}}_i) \right).$$

By the triangle inequality,

$$d(\mathbf{x}_{i-1}, \mathbf{y}_{iB+1}) \leq d(\mathbf{x}_{i-1}, \tilde{\mathbf{y}}_{i-1}) + d(\tilde{\mathbf{y}}_{i-1}, \mathbf{y}_{iB+1}) \leq \sqrt{3\epsilon_{i-1}} + \eta_i BG. \tag{46}$$

Squaring both sides and making use of $(a+b)^2 \leq 2(a^2 + b^2)$, we have

$$d(\mathbf{x}_{i-1}, \mathbf{y}_{iB+1})^2 \leq 6\epsilon_{i-1} + 2B^2 G^2 \eta_i^2. \tag{47}$$

By Lemma 6, the number of iterations in Algorithm 7 for the $i+1$-th block is at most

$$\begin{aligned}
&\max\left\{ \frac{\zeta d(\mathbf{x}_{i-1}, \mathbf{y}_{iB+1})^2 (d(\mathbf{x}_{i-1}, \mathbf{y}_{iB+1})^2 - \epsilon_{i+1})}{4\epsilon_{i+1}^2} + 1, 1 \right\} \\
&\leq \max\left\{ \zeta \cdot \frac{30\epsilon_{i-1}^2 + 22B^2 G^2 \eta_i^2 \epsilon_{i-1} + 4B^4 G^4 \eta_i^4}{4\epsilon_{i+1}^2} + 1, 1 \right\} \\
&\leq \max\left\{ \zeta \frac{61\epsilon_{i-1}^2 + 8B^4 G^4 \eta_i^4}{4\epsilon_{i+1}^2} + 1, 1 \right\} \\
&\leq \max\left\{ \frac{16\epsilon_{i-1}^2}{\epsilon_{i+1}^2}\zeta + \frac{2B^4 G^4 \eta_i^4}{\epsilon_{i+1}^4}\zeta + 1, 1 \right\}
\end{aligned} \tag{48}$$

Thus, the number of calls to the linear optimization oracle is bounded by

$$\begin{aligned}
N_{calls} &:= \sum_{i=2}^{\frac{T}{B}} \left( \frac{16\epsilon_{i-1}^2}{\epsilon_{i+1}^2}\zeta + \frac{2B^4 G^4 \eta_i^4}{\epsilon_{i+1}^4}\zeta + 1 \right) \cdot \frac{27R^2\zeta}{\epsilon_i} \\
&\leq \sum_{i=2}^{\frac{T}{B}} \left( \frac{16(i+3)^4}{(i+1)^4} + \frac{2(i+3)^4}{400^2(i-1)^4\zeta^2} + \frac{1}{\zeta} \right) \cdot \left( \frac{R\alpha}{20G} \right)^2 (i+3)^2 \zeta \\
&\leq 0.32 \left( \frac{R\alpha}{G} \right)^2 \sum_{i=2}^{\frac{T}{B}} (i+3)^2 \zeta
\end{aligned} \tag{49}$$

where the last line follows from $\zeta \geq 1$, $\frac{i+3}{i+1} \leq \frac{5}{3}$ and $\frac{i+3}{i-1} \leq 5$ holds for any $i \geq 2$. On the other hand, we have

$$\sum_{i=2}^{\frac{T}{B}} (i+3)^2 \leq \sum_{i=5}^{\frac{T}{B}+3} i^2 \leq \sum_{i=5}^{\frac{2T}{B}} i^2 \leq \frac{8}{3}\left( \frac{T}{B} \right)^2 \tag{50}$$

Combining Equations (49), (50) and plugging in $B = \left( \frac{\alpha R}{G} \right)^{\frac{2}{3}} T^{2/3}$, we get the claimed $N_{calls} \leq \zeta T$. $\qquad\square$

# D  Some Basic Facts about Convex Sets on Hadamard Manifolds

In this section, we highlight two significant characteristics of convex sets in Euclidean space which do not apply to Hadamard manifolds, emphasizing the fundamental differences between these two spaces. Our counterexamples are constructed within the Poincaré half-plane, $\mathbb{H}^2$. Therefore, understanding some basic properties of this unique manifold will prove beneficial.

**Lemma 10.** *(Udriste, 2013; Wang et al., 2016; Kristály et al., 2016) The Poincaré half plane $\mathbb{H}^2$ can be described by*

$$\mathcal{M} := \{(t_1, t_2) \in \mathbb{R}^2 | t_2 > 0\}$$

*with metric*

$$g_{ij}(u, v) = \frac{\delta_{ij}}{v^2} \qquad i = 1, 2 \quad j = 1, 2.$$

*We have the following conclusions about geodesics and the distance on the Poincaré half-plane.*

*1) For the geodesic connecting $x = (t_1, t_2)$ and $y = (s_1, s_2)$, let*

$$b_{xy} := \frac{s_1^2 + s_2^2 - t_1^2 - t_2^2}{2(s_1 - t_1)} \qquad r_{xy} = \sqrt{(s_1 - b_{xy})^2 + s_2^2},$$

*then the geodesic $\gamma_{xy} := (\gamma_{xy}^1, \gamma_{xy}^2)$ connecting $x$ and $y$ is,*

$$\gamma_{xy}^1(s) := \begin{cases} t_1, & \text{if } t_1 = s_1 \\ b_{xy} - r_{xy} \tanh\left((1-s) \cdot \text{artanh}\frac{b_{xy}-t_1}{r_{xy}} + s \cdot \text{artanh}\frac{b_{xy}-s_1}{r_{xy}}\right), & \text{if } t_1 \neq s_1 \end{cases} \tag{51}$$

$$\gamma_{xy}^2(s) := \begin{cases} e^{(1-s)\cdot \ln t_2 + s \cdot \ln s_2}, & \text{if } t_1 = s_1 \\ \dfrac{r_{xy}}{\cosh\left((1-s)\cdot\text{artanh}\frac{b_{xy}-t_1}{r_{xy}}+s\cdot\text{artanh}\frac{b_{xy}-s_1}{r_{xy}}\right)}, & \text{if } t_1 \neq s_1. \end{cases} \tag{52}$$

*Indeed, we have $(\gamma_{xy}^1(s) - b_{xy})^2 + (\gamma_{xy}^2(s))^2 = r_{xy}^2$ for any $s \in [0, 1]$ when $t_1 \neq s_1$.*

*2) The distance between two points $(u_1, v_1)$ and $(u_2, v_2)$ can be calculated by*

$$d_{\mathbb{H}}((u_1, v_1), (u_2, v_2)) = \text{arcosh}\left(1 + \frac{(u_2 - u_1)^2 + (v_2 - v_1)^2}{2v_1 v_2}\right).$$

### D.1 Shrinking Gsc-convex Sets on Hadamard Manifolds Does Not Preserve Convexity

In the context of projection-free online learning on manifolds, the set $c\mathcal{K} = \text{Exp}_{\mathbf{p}}(c\text{Exp}_{\mathbf{p}}^{-1}\mathbf{x})|\mathbf{x} \in \mathcal{K}$ takes on a pivotal role, particularly when $c < 1$. Consider a convex set $\mathcal{K}$ in Euclidean space, then the convexity of $c\mathcal{K}$ is guaranteed for any positive $c$. Nonetheless, this property does not necessarily hold on Hadamard manifolds. We can construct a counterexample within the Poincaré half-plane $\mathbb{H}^2$. To achieve this, we rely on the following lemma.

**Lemma 11.** *On the Poincaré half-plane $\mathbb{H}^2$, the mid-point of the geodesic $\gamma_{xy}$ which connects $x = (-\lambda, \mu)$ and $y = (\lambda, \mu)$ is $(0, \sqrt{\lambda^2 + \mu^2})$.*

*Proof.* By symmetry we know $\gamma_{xy}^1(\frac{1}{2}) = 0$. To compute $\gamma_{xy}^2(\frac{1}{2})$, note that $b_{xy} = 0$ and $r_{xy} = \sqrt{\lambda^2 + \mu^2}$, then by Lemma 10,

$$\gamma_{xy}^2\left(\frac{1}{2}\right) = \frac{\sqrt{\lambda^2 + \mu^2}}{\cosh\left(\frac{1}{2}\text{arctanh}\frac{\lambda}{\sqrt{\lambda^2+\mu^2}} + \frac{1}{2}\text{arctanh}\frac{-\lambda}{\sqrt{\lambda^2+\mu^2}}\right)} = \sqrt{\lambda^2 + \mu^2},$$

where we use the fact that $\text{arctanh}(\cdot)$ is an odd function and $\cosh(0) = 1$. $\qquad\square$

Now we prove the following theorem.

**Theorem 6.** *There exists a counter-example on the Poincaré half plane $\mathbb{H}^2$ such that the radially contracted set $(1 - \alpha)\mathcal{K}$ where $\alpha \in (0, 1)$ is non-convex.*

*Proof.* Let $p = (0, 1)$, $u = (-1, \sqrt{2})$, $v = (1, \sqrt{2})$, $w = (0, \sqrt{3})$. Observe that $\gamma_{uv}$ is an arc satisfying $\{(\lambda, \mu) | \lambda^2 + \mu^2 = 3\}$ and, by Lemma 11, $w$ is its midpoint. Following this, we calculate $\gamma_{pu}, \gamma_{pw}$, and $\gamma_{pv}$, and shrink these three geodesics by a factor of $(1 - \alpha)$. Let's denote the resulting geodesics as $\gamma_{pu'}, \gamma_{pw'}$, and $\gamma_{pv'}$. Ultimately, we demonstrate that for some $\alpha$, $w'$ is positioned beneath $\gamma_{u'v'}$, implying that the radial contraction of $\mathcal{K}$ does not encompass a geodesic, hence proving

its non-convexity. We first compute $\gamma_{pu}$ and $\gamma_{pv}$ to facilitate this. Note that $b_{pu} = -1$ and $r_{pu} = \sqrt{2}$, by Lemma 10, we can write down

$$\gamma_{pu}^1(s) = -1 - \sqrt{2}\tanh\left((1-s)\operatorname{arctanh}\left(-\frac{1}{\sqrt{2}}\right)\right)$$

and

$$\gamma_{pu}^2(s) = \frac{\sqrt{2}}{\cosh\left((1-s)\operatorname{arctanh}\left(-\frac{1}{\sqrt{2}}\right)\right)}.$$

By symmetry, we have

$$\gamma_{pv}^1(s) = 1 + \sqrt{2}\tanh\left((1-s)\operatorname{arctanh}\left(-\frac{1}{\sqrt{2}}\right)\right)$$

and

$$\gamma_{pv}^2(s) = \frac{\sqrt{2}}{\cosh\left((1-s)\operatorname{arctanh}\left(-\frac{1}{\sqrt{2}}\right)\right)}.$$

We can also compute $\gamma_{pw}(s) = \left(0, e^{s\ln\sqrt{3}}\right)$. Now we have

$$u' = \gamma_{pu}(1-\alpha) = \left(-1 - \sqrt{2}\tanh\left(\alpha\operatorname{arctanh}\left(-\frac{1}{\sqrt{2}}\right)\right), \frac{\sqrt{2}}{\cosh\left(\alpha\operatorname{arctanh}\left(-\frac{1}{\sqrt{2}}\right)\right)}\right)$$

$$v' = \gamma_{pv}(1-\alpha) = \left(1 + \sqrt{2}\tanh\left(\alpha\operatorname{arctanh}\left(-\frac{1}{\sqrt{2}}\right)\right), \frac{\sqrt{2}}{\cosh\left(\alpha\operatorname{arctanh}\left(-\frac{1}{\sqrt{2}}\right)\right)}\right)$$

and $w' = \left(0, e^{(1-\alpha)\ln\sqrt{3}}\right)$. We just need to demonstrate that for some $\alpha \in (0,1)$, $w'$ is positioned below $\gamma_{u'v'}\left(\frac{1}{2}\right)$, where

$$\gamma_{u'v'}\left(\frac{1}{2}\right) = \left(0, \sqrt{3 + 2\sqrt{2}\tanh\left(\alpha\operatorname{arctanh}\left(-\frac{1}{\sqrt{2}}\right)\right)}\right).$$

by Lemma 11. Specifically, we need to show $\exists\alpha \in (0,1)$,

$$e^{(1-\alpha)\ln\sqrt{3}} < \sqrt{3 + 2\sqrt{2}\tanh\left(\alpha\operatorname{arctanh}\left(-\frac{1}{\sqrt{2}}\right)\right)}$$

We set $\alpha = .1$, then

$$\sqrt{3 + 2\sqrt{2}\tanh\left(\alpha\operatorname{arctanh}\left(-\frac{1}{\sqrt{2}}\right)\right)} - e^{(1-\alpha)\ln\sqrt{3}} = 0.036\cdots > 0.$$

Therefore, the set $(1-\alpha)\mathcal{K}$ fails to contain the geodesic $\gamma_{u'v'}(s)$ and is thus non-convex.

$\square$

### D.2 Non-convexity of the Minkowski Functional on Hadamard Manifolds

In this part, we show the Minkowski functional (Gauge function), which is defined as

$$\gamma_{\mathcal{K}}(\mathbf{x}) = \inf\left\{\lambda > 0 : \operatorname{Exp}_{\mathbf{p}}\left(\frac{\operatorname{Exp}_{\mathbf{p}}^{-1}\mathbf{z}}{\lambda}\right) \in \mathcal{K}\right\} \tag{53}$$

can be non-convex on Hadamard manifolds, where $\mathcal{K}$ is a gsc-convex set of a Hadamard manifold $\mathcal{M}$ and $\mathbf{p}$ is an interior point in $\mathcal{K}$.

**Theorem 7.** *The Minkowski functional defined in Equation* (53) *can be non-convex on Hadamard manifolds.*

*Proof.* We again construct a counterexample on the Poincaré plane $\mathbb{H}^2$:

$$\mathcal{M} := \{(t_1, t_2) \in \mathbb{R}^2 | t_2 > 0\}.$$

To verify the geodesic convexity, we can verify an equivalent statement: a function $f$ is gsc-convex iff $f$ is convex when restricted to any geodesic. That is $f(\gamma(t)) \leq (1-t)f(\gamma(0)) + tf(\gamma(1))$ holds for any geodesic $\gamma(t) : [0,1] \to \mathcal{M}$. The counterexample is as follows.

Assume $\mathcal{K}$ is the gsc-convex set consists of all points $(u_1, u_2)$ such that $u_1^2 + u_2^2 \leq 3$ and $u_2 > 0$. $x = (-1, \sqrt{3})$, $y = (-2, 2)$ and $p = (0, 1)$. We show that $\gamma_{\mathcal{K}}(x)$ is non-convex on the geodesic $\gamma_{xy}$. We first compute $b_{xy} = -2$ and $r_{xy} = 2$. By Lemma 10, we have

$$\gamma_{xy}(s) = \left(-2 - 2\tanh\left((1-s)\operatorname{arctanh}\left(-\frac{1}{2}\right)\right), \frac{2}{\cosh\left((1-s)\operatorname{arctanh}\left(-\frac{1}{2}\right)\right)}\right) \quad (54)$$

$$:= (-2 - 2\cos\theta(s), 2\sin\theta(s)),$$

where we let $\theta(s) := \tanh\left((1-s)\operatorname{arctanh}\left(-\frac{1}{2}\right)\right)$ to ease the elaboration. We denote $z := \gamma_{xy}(s) = (-2 - 2\cos\theta(s), 2\sin\theta(s))$. Then we compute the geodesic $\gamma_{pz}$ connecting $z$ and $p = (0,1)$, and we have

$$b_{pz}(s) = \frac{(2 + 2\cos\theta(s))^2 + (2\sin\theta(s))^2 - 1}{2(-2 - 2\cos\theta(s))} = -\frac{7 + 8\cos\theta(s)}{4(1 + \cos\theta(s))}$$

and $r_{pz}(s) = \sqrt{b_{pz}(s)^2 + 1}$. Now we can compute the intersection of $\gamma_{pz}(s)$ with $\mathcal{K}$ as the solution $(q_1, q_2)$ of the following system of equations

$$\begin{cases} (q_1 - b_{pz}(s))^2 + q_2^2 = b_{pz}(s)^2 + 1 \\ q_1^2 + q_2^2 = 3 \end{cases}$$

The intersection point is $\left(\frac{1}{b_{pz}(s)}, \sqrt{3 - \frac{1}{b_{pz}(s)^2}}\right)$. For each $s \in [0,1]$, we can evaluate the Minkowski functional at $\gamma_{xy}(s)$ w.r.t. $\mathcal{K}$ as

$$h(s) = d_{\mathbb{H}}((-2 - 2\cos\theta(s), 2\sin\theta(s)), (0,1)) / d_{\mathbb{H}}\left(\left(\frac{1}{b_{pz}(s)}, \sqrt{3 - \frac{1}{b_{pz}(s)^2}}\right), (0,1)\right),$$

where $d_{\mathbb{H}}$ is the hyperbolic distance defined in the second item of Lemma 10. If the Minkowski functional is gsc-convex, then we should have $h(s) \leq (1-s)h(0) + s \cdot h(1)$ for any $s \in [0,1]$. However, by numerical computation, we find

$$h(1/2) - \frac{1}{2}h(0) - \frac{1}{2}h(1) = 0.0057\cdots > 0,$$

which refutes the assumption that $h(s)$ is convex. Thus our conclusion is established. $\qquad \square$

# E   Technical Lemmas

## E.1   Background on the Jacobi Field

The role of the Jacobi field as an essential tool for understanding the impact of curvature on the behavior of neighboring geodesics cannot be overstated. To make our discussion more comprehensive and self-contained, we will introduce related definitions and properties sourced from Lee (2006) and Lee (2018).

We define an admissible curve as a continuous map, $\gamma(t) : [a,b] \to \mathcal{M}$, representing a piecewise regular curve segment. An admissible family of curves, on the other hand, is a continuous map $\Gamma(t,s) : [a,b] \times (-\epsilon, \epsilon) \to \mathcal{M}$ which is smooth on each rectangle $[a_{i-1}, a_i] \times (-\epsilon, \epsilon)$ for a finite subdivision $a = a_0 < \cdots < a_k = b$. Here, $\Gamma_s(t) := \Gamma(t,s)$ is recognized as an admissible curve for each $s \in (-\epsilon, \epsilon)$. When $\gamma(t)$ is admissible and $\Gamma(t,0) = \gamma(t)$, we refer to $\Gamma$ as a variation of $\gamma(t)$.

Upon an admissible curve $\Gamma$, we can delineate two sets of curves: the principal curves $\Gamma_s(t)$, which hold $s$ as constant, and the transverse curves $\Gamma^{(t)}(s) := \Gamma(t,s)$, which consider $t$ as constant. We

consider $\Gamma$ as a variation through geodesics if each principal curve $\Gamma_s(t) = \Gamma(t, s)$ is a geodesic segment. For an admissible family of curves $\Gamma$, we define a vector field along $\Gamma$ as a map $V : [a, b] \times (-\epsilon, \epsilon) \to T\mathcal{M}$ such that $V(t, s) \in T_{\Gamma(t,s)}\mathcal{M}$ and for each $s$, $V(s, t)$ is piecewise smooth. The Jacobi field is a vector field along a variation through geodesics and is formally defined as follows.

**Lemma 12.** *(Lee, 2018, Theorem 10.1, The Jacobi Equation) Let $(\mathcal{M}, g)$ be a Riemannian manifold, $\gamma$ be a geodesic in $\mathcal{M}$, and $J$ be a vector field along $\gamma$. If $J$ is the variation field of a variation through geodesics, then $J$ needs to satisfy the following* Jacobi Equation*:*

$$D_t^2 J + R(J, \dot{\gamma})\dot{\gamma} = 0,$$

*where $R$ is the Riemann curvature endomorphism. A smooth vector field $J$ satisfying the Jacobi Equation is called a Jacobi field.*

Given initial values of $J$ and $D_t J$ at one point, the Jacobi Equation can be uniquely determined by the following proposition.

**Lemma 13.** *(Lee, 2018, Prop. 10.2, Existence and Uniqueness of Jacobi Fields) Let $(\mathcal{M}, g)$ be a Riemannian manifold and $\gamma : I \to \mathcal{M}$ is a geodesic where $I$ is an interval. Fix $a \in I$, let $p = \gamma(a)$, $u, v \in T_p\mathcal{M}$, there is a unique Jacobi field $J$ along $\gamma$ which satisfies*

$$J(a) = u, \qquad D_t J(a) = v.$$

The Jacobi field seems to be a complicated object to study, but the following lemma shows we can write down the initial conditions of the Jacobi field for certain variations through geodesics.

**Lemma 14.** *(Lee, 2018, Lemma 10.9) Let $(\mathcal{M}, g)$ be a Riemannian manifold, $I$ be an interval containing 0, and $\gamma : I \to \mathcal{M}$ be a geodesic. Suppose $J : I \to \mathcal{M}$ be a Jacobi field with $J(0) = 0$. If $\mathcal{M}$ is geodesically complete or $I$ is compact, then $J$ is the variation field of the following variation of $\gamma$:*

$$\Gamma(t, s) = \mathrm{Exp}_p(t(u + sv)),$$

*where $p = \gamma(0)$, $u = \dot{\gamma}(0)$, and $v = D_t J(0)$.*

The advantage of introducing the Jacobi field is the norm of $J(t)$ can be bounded by the initial value $\|D_t J(0)\|$, as shown in the following Lemma 15.

**Definition 4.** *We define*

$$\mathbf{s}(\kappa, t) = \begin{cases} t, & \text{if } \kappa = 0 \\ \frac{1}{\sqrt{\kappa}} \sin(\sqrt{\kappa}t), & \text{if } \kappa > 0 \\ \frac{1}{\sqrt{-\kappa}} \sinh(\sqrt{-\kappa}t), & \text{if } \kappa < 0. \end{cases}$$

**Lemma 15.** *(Lee, 2018, Theorem. 11.9 ) Suppose $(\mathcal{M}, g)$ is a Riemannian manifold, $\gamma : [0, b] \to \mathcal{M}$ is a unit-speed geodesic segment, and $J$ is any Jacobi field along $\gamma$ such that $J(0) = 0$. For each $c \in \mathbb{R}$, let $s(\cdot, \cdot)$ be the function defined by Definition 4.*

*1) If all sectional curvatures of $M$ are bounded above by a constant $\kappa_2$, then*

$$\|J(t)\| \geq s(\max\{0, \kappa_2\}, t) \|D_t J(0)\|$$

*for all $t \in [0, b_1]$, where $b_1 = b$ if $\kappa_2 \leq 0$, and $b_1 = \min\left(b, \frac{\pi}{\sqrt{\kappa_2}}\right)$ if $\kappa_2 > 0$.*

*2) If all sectional curvatures of $M$ are bounded below by a constant $\kappa_1$, then*

$$\|J(t)\| \leq s(\min\{0, \kappa_1\}, t) \|D_t J(0)\|$$

*for all $t \in [0, b_2]$, where $b_2$ is chosen so that $\gamma(b_2)$ is the first conjugate point to $\gamma(0)$ along $\gamma$ if there is one, and otherwise $b_2 = b$.*

## E.2 Miscellaneous Technical Lemmas

**Lemma 16.** *(Bacák, 2014, Section 2.1) Let $\mathcal{M}$ be a Hadamard manifold, $f : \mathcal{M} \to (-\infty, \infty)$ be a gsc-convex lower semicontinuous function. Then any $\beta$-sublevel set of $f$:*

$$\{\mathbf{x} \in \mathcal{M} : f(\mathbf{x}) \leq \beta\}$$

*is a closed gsc-convex set.*

**Lemma 17.** *(Bacák, 2014, Section 2.2) Let $\mathcal{M}$ be a Hadamard manifold, $f_i : \mathcal{M} \to (-\infty, \infty)$ for $i = 1, \ldots, m$ be a series of gsc-convex lower semicontinuous functions. Then $\sup_{i \in [m]} f_i(\mathbf{x})$ is gsc-convex.*

**Lemma 18.** *For a set $\mathcal{K} := \{\mathbf{x} | \max_{1 \leq i \leq m} h_i(\mathbf{x}) \leq 0\} \subseteq \mathcal{M}$ where each $h_i(\mathbf{x})$ is gsc-convex, $\mathcal{M}$ is Hadamard and any $\mathbf{y} \notin \mathcal{K}$, there exists $\mathbf{g} \in T_{\mathbf{y}}\mathcal{M}$ such that*

$$\left\langle -\mathrm{Exp}_{\mathbf{y}}^{-1}\mathbf{x}, \mathbf{g} \right\rangle > 0, \quad \forall \mathbf{x} \in \mathcal{K}.$$

*Proof.* By Lemma 17, $\max_{1 \leq i \leq m} h_i(\mathbf{x})$ is gsc-convex. Further, by Lemma 16, $\mathcal{K}$ is a gsc-convex subset of $\mathcal{M}$. For some $\mathbf{y} \notin \bar{\mathcal{K}}$, there exists $j \in [m]$ such that $h_j(\mathbf{y}) > 0$. If not, we have $h_i(\mathbf{y}) \leq 0$ holds for any $i = 1, \ldots, m$, which implies $\mathbf{y} \in \mathcal{K}$ and leads to a contradictory. Applying the gsc-convexity for any $\mathbf{x} \in \mathcal{K}$, we have

$$-\left\langle \mathrm{Exp}_{\mathbf{y}}^{-1}\mathbf{x}, \nabla h_j(\mathbf{y}) \right\rangle \geq h_j(\mathbf{y}) - h_j(\mathbf{x}) > 0,$$

where the second inequality is because $h_j(\mathbf{y}) > 0$ and $h_j(\mathbf{x}) \leq 0$. This is a valid separation oracle between $\mathbf{y}$ and $\mathcal{K}$. $\qquad\square$

**Remark 4.** *Given $\mathcal{K} = \{\mathbf{x} | \max_{1 \leq i \leq m} h_i(\mathbf{x}) \leq 0\}$ where each $h_i(\mathbf{x})$ is gsc-convex and $\mathbf{y} \notin \mathcal{K}$, we can check the sign of $h_i(\mathbf{y})$ for $i = 1, \ldots, m$ until we find $i^*$ such that $h_{i^*}(\mathbf{y}) > 0$, then using the proof of Lemma 18, we obtain*

$$-\left\langle \mathrm{Exp}_{\mathbf{y}}^{-1}\mathbf{x}, \nabla h_{i^*}(\mathbf{y}) \right\rangle > 0$$

*for every $\mathbf{x} \in \mathcal{K}$. This gives us a separation oracle between $\mathbf{y}$ and $\mathcal{K}$. The computation of this separation oracle requires no more than $m$ function value evaluations and a single gradient evaluation. This is efficiently implementable when $m$ is of moderate size.*

**Remark 5.** *Using Gauss's Lemma (Lee, 2006, Theorem 6.8), we can efficiently compute the projection onto a geodesic ball. Consider $\mathbf{x} \notin \mathbb{B}_{\mathbf{p}}(r)$ for which we aim to compute its projection onto $\mathbb{B}_{\mathbf{p}}(r)$. First, we determine the geodesic segment, denoted as $\gamma(t)$, connecting $\mathbf{x}$ and $\mathbf{p}$. This segment intersects the geodesic ball at point $\mathbf{z}$. Gauss's Lemma then tells us that the tangent vector of $\gamma(t)$ at $\mathbf{z}$ is orthogonal to the geodesic sphere $\mathbb{S}_p(r)$. This indicates that $\mathbf{z}$ is the projection of $\mathbf{x}$ onto $\mathbb{B}_{\mathbf{p}}(r)$. By performing $O(\log\left(\frac{1}{\epsilon}\right))$ rounds of binary search on $\gamma(t)$, we can approximate $\mathbf{z}$ to within an $\epsilon$ precision. In each round, it suffices to verify if the condition $d(\mathbf{x}_t, \mathbf{p}) \geq r$ is met.*

**Lemma 19.** *(Wang et al., 2023, Lemma 45) Suppose the sectional curvature of $\mathcal{M}$ is in $[\kappa_1, \kappa_2]$ where $\kappa_1 \leq 0$ and $\kappa_2 \geq 0$. Assume a gsc-convex set $\mathcal{K}$ satisfies $\mathbb{B}_{\mathbf{p}}(r) \subseteq \mathcal{K} \subseteq \mathbb{B}_{\mathbf{p}}(R)$ where $R \leq \frac{\pi}{2\sqrt{\kappa_2}}$ when $\kappa_2 > 0$. Then for any $\mathbf{y} \in (1 - \tau)\mathcal{K}$, the geodesic ball $\mathbb{B}_{\mathbf{y}}\left(\frac{s(\kappa_2, R+r) \cdot \tau r}{s(\kappa_1, R+r)}\right) \in \mathcal{K}$, where $s(\cdot, \cdot)$ is defined in Definition 4.*

**Lemma 20.** *(Bridson & Haefliger, 2013, Chapter II.1) Suppose $\Delta(\mathbf{p}, \mathbf{q}, \mathbf{r})$ is a triangle on a Hadamard manifold $\mathcal{M}$ and $\Delta(\bar{\mathbf{p}}, \bar{\mathbf{q}}, \bar{\mathbf{r}})$ is a comparison triangle in Euclidean space such corresponding side lengths are the same for both triangles. Then for any $\mathbf{x} \in [\mathbf{p}, \mathbf{q}]$, $\mathbf{y} \in [\mathbf{p}, \mathbf{r}]$, $\bar{\mathbf{x}} \in [\bar{\mathbf{p}}, \bar{\mathbf{q}}]$, $\bar{\mathbf{y}} \in [\bar{\mathbf{p}}, \bar{\mathbf{r}}]$ satisfying $d(\mathbf{p}, \mathbf{x}) = d_{\mathbb{E}}(\bar{\mathbf{p}}, \bar{\mathbf{x}})$ and $d(\mathbf{p}, \mathbf{y}) = d_{\mathbb{E}}(\bar{\mathbf{p}}, \bar{\mathbf{y}})$, we have $d(\mathbf{x}, \mathbf{y}) \leq d_{\mathbb{E}}(\bar{\mathbf{x}}, \bar{\mathbf{y}})$.*

**Lemma 21.** *The function $\frac{\sinh x}{x}$ is increasing for any $x \in [0, \infty]$.*

*Proof.* We denote $g(x) = \frac{\sinh x}{x}$, then

$$g'(x) = \frac{x \cosh x - \sinh x}{x^2}.$$

It suffices to show $h(x) := x \cosh x - \sinh x \geq 0$ holds for any $x \in [0, \infty)$. This is obvious because $h(0) = 0$ and $h'(x) = x \sinh x \geq 0$ for any $x \in [0, \infty)$. It remains to compute $g'(0)$. By L'Hôpital's rule,

$$g'(0) = \lim_{x \to 0} \frac{x \cosh x - \sinh x}{x^2} = \lim_{x \to 0} \frac{x \sinh x}{2x} = 0.$$

$\qquad\square$

**Lemma 22.** *Let $f : \mathcal{M} \to \mathbb{R}$ be gsc-convex and $G$-Lipschitz on a gsc-convex set $\mathcal{K} \subseteq \mathcal{M}$. Then*

$$\hat{f}(\mathbf{x}) := \frac{1}{V_\delta} \int_{\mathbb{B}_{\mathbf{x}}(\delta)} f(\mathbf{u})\omega$$

*satisfies*

$$|\hat{f}(\mathbf{x}) - f(\mathbf{x})| \leq G\delta.$$

*Proof.* We have

$$
\begin{aligned}
|\hat{f}(\mathbf{x}) - f(\mathbf{x})| = \left|\mathbb{E}_{\|\mathbf{v}\|=1}\left[f(\mathrm{Exp}_\mathbf{x}(\delta\mathbf{v}) - f(\mathbf{x})]\right]\right| \\
\leq \mathbb{E}_{\|\mathbf{v}\|=1}\left[f(\mathrm{Exp}_\mathbf{x}(\delta\mathbf{v})) - f(\mathbf{x})\right] \\
\leq \mathbb{E}_{\|\mathbf{v}\|=1}\left[G\|\delta\mathbf{v}\|\right] \leq G\delta.
\end{aligned}
\tag{55}
$$

where $\mathbf{v} \in T_\mathbf{x}\mathcal{M}$. The first inequality is due to Jensen's inequality, and the second one is by the gradient Lipschitzness of $f$. $\qquad\square$

**Definition 5.** *Given a homogeneous Riemannian manifold $\mathcal{M}$, we use $S_\delta$ and $V_\delta$ to denote the surface area and the volume of a geodesic ball with radius $\delta$.*

**Remark 6.** *We note that on homogeneous Riemannian manifolds, the volume and the area of a geodesic ball only depend on its radius (Kowalski & Vanhecke, 1982), so $S_\delta$ and $V_\delta$ are well-defined.*

**Lemma 23.** *(Wang et al., 2023, Lemmas 11 and 13) For a homogeneous Hadamard manifold $(\mathcal{M}, g)$ with sectional curvature lower bounded by $\kappa \leq 0$, $f$ be a $C^1$ and gsc-convex function defined on $\mathcal{M}$ such that $|f| \leq C$ for any $x \in \mathcal{M}$. We define*

$$
\hat{f}(\mathbf{x}) = \frac{1}{V_\delta} \int_{\mathbb{B}_\mathbf{x}(\delta)} f(\mathbf{u})\omega
$$

*where $\omega$ is the volume element, and*

$$
\mathbf{g}(\mathbf{u}) = f(\mathbf{u})\frac{\mathrm{Exp}_\mathbf{x}^{-1}(\mathbf{u})}{\|\mathrm{Exp}_\mathbf{x}^{-1}(\mathbf{u})\|}.
$$

*is a corresponding gradient estimator. Then we have*

*1) $\frac{S_\delta}{V_\delta}\mathbf{g}(\mathbf{u})$ is an unbiased estimator of $\nabla\hat{f}(\mathbf{x})$. More specifically, for any $\mathbf{x} \in \mathcal{M}$,*

$$
\mathbb{E}_{\mathbf{u}\in\mathbb{S}_\mathbf{x}(\delta)}\left[\frac{S_\delta}{V_\delta}\mathbf{g}(\mathbf{u})\Big|\mathbf{x}\right] = \nabla\hat{f}(\mathbf{x}).
$$

*2) The expected norm of $\frac{S_\delta}{V_\delta}\mathbf{g}(\mathbf{u})$ can be bounded by*

$$
\mathbb{E}_{\mathbf{u}\in\mathbb{S}_\mathbf{x}(\delta)}\left[\left\|\frac{S_\delta}{V_\delta}\mathbf{g}(\mathbf{u})\Big|\mathbf{x}\right\|\right] \leq \frac{S_\delta}{V_\delta}C \leq \left(\frac{n}{\delta} + n|\kappa|\delta\right)C.
$$

*3) Suppose $\mathcal{K} \in \mathcal{M}$ is a gsc-convex set and $\|\nabla f(\mathbf{x})\| \leq G$, then $\hat{f}$ satisfies*

$$
\hat{f}(\mathbf{y}) - \hat{f}(\mathbf{x}) - \left\langle\nabla\hat{f}(\mathbf{x}), \mathrm{Exp}_\mathbf{x}^{-1}(\mathbf{y})\right\rangle \geq -2\rho\delta G,
$$

*where $\rho$ solely depends on the set $\mathcal{K}$.*

**Remark 7.** *From the proof of Lemma 23, we can figure out the formal definition of $\rho$ is*

$$
\rho = \sup_{\mathbf{x},\mathbf{y},\mathbf{u}\in\mathcal{K}}\left\|\frac{1}{\sqrt{G}}\frac{\partial}{\partial x_i}\left(\sqrt{G}\mathrm{Exp}_\mathbf{u}^{-1}\phi(\mathbf{u})\right)^i\right\| \quad s.t. \quad \phi(\mathbf{x}) = \mathbf{y}.
$$

*By Wang et al. (2023), this quantity can be $O\left(e^D\right)$ even on a 2-dimensional Poincaré disk.*

---

**Algorithm 9:** Riemannian Frank-Wolfe with line-search

**Data:** feasible gsc-convex set $\mathcal{K}$, initial point $\mathbf{x}_0 \in \mathcal{K}$, gsc-convex objective $f(\cdot)$.
**for** $i = 0, \ldots$ **do**
$\quad\mathbf{v}_i = \mathrm{argmin}_{\mathbf{x}\in\mathcal{K}}\{\langle\nabla f(\mathbf{x}_i), \mathrm{Exp}_{\mathbf{x}_i}^{-1}\mathbf{x}\rangle\}$
$\quad\sigma_i = \mathrm{argmin}_{\sigma\in[0,1]}\{f(\mathrm{Exp}_{\mathbf{x}_i}(\sigma\mathrm{Exp}_{\mathbf{x}_i}^{-1}\mathbf{v}_i))\}$
$\quad\mathbf{x}_{i+1} = \mathrm{Exp}_{\mathbf{x}_i}(\sigma_i\mathrm{Exp}_{\mathbf{x}_i}^{-1}\mathbf{v}_i)$
**end**

---

**Lemma 24.** *(Weber & Sra, 2022b, Theorem 1) Suppose the diameter of the gsc-convex set $\mathcal{K} \subseteq \mathcal{M}$ is upper bounded by $D$ and $f$ is gsc-convex as well as $L$ gsc-smooth on $\mathcal{K}$. With a linear optimization oracle in hand, the primal convergence rate of Algorithm 9 can be described by*

$$f(\mathbf{x}_k) - f(\mathbf{x}^*) \leq \frac{2LD^2}{k+2}.$$

**Lemma 25.** *(Weber & Sra, 2022b, Lemma 2) Under the same assumptions as in Lemma 24, for a step $\mathbf{x}_{k+1} = \mathrm{Exp}_{\mathbf{x}_k}(\eta \mathrm{Exp}_{\mathbf{x}_k}^{-1} \mathbf{v})$ with $\eta \in [0, 1]$,*

$$f(\mathbf{x}_{k+1}) \leq f(\mathbf{x}_k) + \eta \left\langle \nabla f(\mathbf{x}_k), \mathrm{Exp}_{\mathbf{x}_k}^{-1} \mathbf{v} \right\rangle + \frac{\eta^2 LD^2}{2}.$$

**Lemma 26.** *Under the same assumptions as in Lemma 24, denote*

$$g(\mathbf{x}) := \max_{\mathbf{v} \in \mathcal{K}} \left\langle -\mathrm{Exp}_{\mathbf{x}}^{-1} \mathbf{v}, \nabla f(\mathbf{x}) \right\rangle.$$

*If Algorithm 9 is run for $K \geq 2$ rounds, then there exists $k_*$ such that $1 \leq k_* \leq K$ and*

$$g(\mathbf{x}_{k_*}) \leq \frac{27LD^2}{4(K+2)}.$$

*Proof.* The proof is an adaptation of Jaggi (2013, Theorem 2). We first assume the duality gap is large in the last one third of $K$ iterations, then get a contradictory to prove conclusion. For simplicity, we define $h_k := f(\mathbf{x}_k) - f(\mathbf{x}^*)$ and $g_k := g(\mathbf{x}_k)$.

To ease the elaboration, we denote $C := 2LD^2$, $\alpha := \frac{2}{3}$, $b := \frac{27}{8}$ and $d := K + 2$. We assume $g_k > \frac{bC}{K+2} = \frac{bC}{d}$ holds for the last one third of the $K$ iterations. More specifically,

$$g_k > \frac{bC}{d} \quad \text{for} \quad k \in \{\lceil \alpha d \rceil - 2, \ldots, K\}.$$

By Lemma 25 with $\eta = \frac{1}{k+2}$, we have

$$\begin{aligned}
h_{k+1} &\leq h_k - \frac{2}{k+2} g_k + \frac{2LD^2}{(k+2)^2} \\
&= h_k - \frac{2}{k+2} g_k + \frac{C}{(k+2)^2}
\end{aligned} \tag{56}$$

Plugging in the assumption that the duality gap is large, we arrive at

$$\begin{aligned}
h_{k+1} &< h_k - \frac{2}{k+2} \frac{bC}{d} + \frac{C}{(k+2)^2} \\
&\leq h_k - \frac{2bC}{d^2} + \frac{C}{\alpha^2 d^2} \\
&= h_k - \frac{2bC - C/\alpha^2}{d^2}
\end{aligned} \tag{57}$$

for all $k = \lceil \alpha d \rceil - 2, \ldots, K$, when the second inequality follows from $\lceil \alpha d \rceil - 2 \leq k \leq K$ implies $\alpha d \leq k + 2 \leq d$.

Denote $k_{min} := \lceil \alpha d \rceil - 2$ and $h_{min} = h_{k_{min}}$, then there are $K - k_{min} + 1 = K - (\lceil \alpha d \rceil - 2) + 1 \geq (1 - \alpha)d$ steps in the last one third iterations. Thus summing Equation (57) from $k = \lceil \alpha d \rceil - 2$ to $K$, we have

$$\begin{aligned}
h_{K+1} &< h_{min} - (1 - \alpha)d \frac{2bC - C/\alpha^2}{d^2} \\
&\leq \frac{C}{\alpha d} - (1 - \alpha)d \frac{2\alpha b - 1/\alpha}{d} \cdot \frac{C}{\alpha d} \\
&= \frac{C}{\alpha d} \left(1 - (1 - \alpha) \cdot (2\alpha b - 1/\alpha)\right) \\
&\leq 0,
\end{aligned} \tag{58}$$

where $h_{min} \leq \frac{C}{\alpha d}$ is due to Lemma 24, while for the last step, we plug in the values $\alpha = \frac{2}{3}$ and $b = \frac{27}{8}$. However, $h_{K+1} < 0$ is impossible because $h_{K+1} = f(\mathbf{x}_{K+1}) - f(\mathbf{x}^*)$. Thus, there always exists $k_*$ in the last one third iterations of $K$ such that

$$g(\mathbf{x}_{k_*}) \leq \frac{bC}{d} = \frac{\frac{27}{8} \cdot 2LD^2}{K+2} = \frac{27LD^2}{4(K+2)}.$$

$\square$

**Lemma 27.** *(Ahn & Sra, 2020, Prop. H.1) Let $\mathcal{M}$ be a Riemannian manifold with sectional curvatures lower bounded by $\kappa \leq 0$ and the distance function $d(\mathbf{x}) = \frac{1}{2}d(\mathbf{x}, \mathbf{p})^2$ where $\mathbf{p} \in \mathcal{M}$. For $D \geq 0$, $d(\cdot)$ is $\zeta$ gsc-smooth within the domain $\{\mathbf{u} \in \mathcal{M} : d(\mathbf{u}, \mathbf{p}) \leq D\}$ where $\zeta = \sqrt{-\kappa}D \coth(\sqrt{-\kappa}D)$ is the geometric constant defined in Definition 1.*

**Lemma 28.** *Suppose $f_i(\mathbf{x})$ is $\alpha$-strongly gsc-convex for any $i = 1, \ldots, N$, then $\sum_{i=1}^{N} f_i(\mathbf{x})$ is $\alpha N$-strongly gsc-convex.*

*Proof.* By the $\alpha$-strongly gsc-convexity of each $f_i(\mathbf{x})$,

$$f_i(\mathbf{y}) \geq f_i(\mathbf{x}) + \left\langle \nabla f_i(\mathbf{x}), \text{Exp}_{\mathbf{x}}^{-1}(\mathbf{y}) \right\rangle + \frac{\alpha}{2}d(\mathbf{x}, \mathbf{y})^2.$$

Summing from $i = 1$ to $N$, we have

$$\begin{aligned}
\sum_{i=1}^{N} f_i(\mathbf{y}) &\geq \sum_{i=1}^{N} f_i(\mathbf{x}) + \sum_{i=1}^{N} \left\langle \nabla f_i(\mathbf{x}), \text{Exp}_{\mathbf{x}}^{-1}\mathbf{y} \right\rangle + \sum_{i=1}^{N} \frac{\alpha}{2}d(\mathbf{x}, \mathbf{y})^2 \\
&= \sum_{i=1}^{N} f_i(\mathbf{x}) + \left\langle \sum_{i=1}^{N} \nabla f_i(\mathbf{x}), \text{Exp}_{\mathbf{x}}^{-1}\mathbf{y} \right\rangle + \frac{1}{2}(\alpha N)\, d(\mathbf{x}, \mathbf{y})^2.
\end{aligned}$$

(59)

This indeed implies $\sum_{i=1}^{N} f_i(\mathbf{y})$ is $\alpha N$-strongly gsc-convex. $\square$

**Lemma 29.** *(Zhang & Sra, 2016, Lemma 5). Let $\mathcal{M}$ be a Riemannian manifold with sectional curvature lower bounded by $\kappa \leq 0$. Consider a geodesic triangle fully lies within $\mathcal{M}$ with side lengths $a, b, c$, we have*

$$a^2 \leq \zeta(\kappa, c)b^2 + c^2 - 2bc \cos A$$

*where $\zeta(\kappa, c) := \sqrt{-\kappa}c \coth(\sqrt{-\kappa}c)$.*

**Lemma 30.** *(Sakai, 1996, Prop. 4.5) Let $\mathcal{M}$ be a Riemannian manifold with sectional curvature upper bounded by $\kappa \leq 0$. Consider $\mathcal{N}$, a gsc-convex subset of $\mathcal{M}$ with diameter $D$. For a geodesic triangle fully lies within $\mathcal{N}$ with side lengths $a, b, c$, we have*

$$a^2 \geq b^2 + c^2 - 2bc \cos A.$$

### E.3 Extension to CAT($\kappa$) Spaces

It is not too difficult to generalize the result in this work to CAT($\kappa$) spaces, which are simply connected manifolds with sectional curvature upper bounded by $\kappa$. To achieve this extension, the following adjustments would be necessary in our manuscript:

- We may assume that the sectional curvature lies in the range $[\kappa, K]$. In this context, we can replace Lemma 30 in our draft with Alimisis et al. (2020, Corollary 2.1). When $K > 0$, we must further assume that the diameter of the decision set is upper-bounded by $\frac{\pi}{\sqrt{K}}$.

- The separation theorem, specifically for Hadamard manifolds as outlined in Silva Louzeiro et al. (2022), would require generalization to CAT($\kappa$) spaces. This change ensures that the separation oracle remains well-defined.

- Lemma 3 requires modification to accommodate the recomputation of the Jacobi field, considering the manifold's sectional curvature. Lemmas 15 and 19 will be instrumental in achieving this.

