# OpenReview forum: "Riemannian Projection-free Online Learning"
_NeurIPS.cc/2023/Conference — NeurIPS 2023 poster_

### Official Review · Reviewer_2HSe · 2023-07-04

**Soundness:** 3 good
**Presentation:** 4 excellent
**Contribution:** 3 good
**Rating:** 8
**Confidence:** 1

**Summary:**

The authors present online algorithms for geodesically convex losses on Riemannian manifolds. Their algorithms do not call the expensive operation of projection onto a feasible set. Instead of projection, they rely on two oracles to provide a direction of descent: a separation oracle and a linear oracle. Both require an extension of the concept of hyperplanes from Euclidean space to the manifold, for which they use an inverse exponential map. This map transforms points on the manifold into tangent space vectors, enabling operations on the manifold without the need for explicit projection operations. In addition, they also consider the projection onto a geodesic ball, which is computable with high accuracy and thus effectively projection free. Their algorithms give adaptive regret guarantees which are sublinear in the horizon.

**Strengths:**

- The authors' contribution is significant as they address the challenge of projecting onto a Riemannian manifold, which is computationally expensive and can lead to geometric distortions. This area of research seems relatively less explored.

- The writing style is clear and they provide a comprehensive background on Reimannian geometry, making it easy for a read to grasp the problem and motivation.


**Weaknesses:**

As I am not familiar with this area, I am unable to identify any weaknesses or limitations.

**Questions:**

I am not sure why $\tilde{\mathcal{K}} = (1-\delta) \mathcal{K}$ is considered as the feasible set for the separation oracle.

**Limitations:**

The authors have acknowledged the limitations of their work, specifically the absence of a membership oracle, and have discussed potential enhancements to their algorithms. These improvements include utilizing the separation oracle for strongly convex losses, reducing the reliance on the number of calls regarding the set's diameter and a faster method to optimise the objective in the linear oracle.

---

> ### Author Rebuttal · Authors · 2023-08-09
>
> Thank you for the time and effort you put into understanding our work. Your supportive and constructive review means a lot to us. Below, we have provided answers to your specific questions and addressed your concerns.
> > I am not sure why $\tilde{\mathcal{K}} = (1-\delta) \mathcal{K}$ is considered as the feasible set for the separation oracle.
>
>  Thanks for your question. The main reason is we want to make sufficiently large progress for each call to the separation oracle. By Lemma 2, we need a separating hyperplane $-\left<\text{Exp}_y^{-1}z,g\right>\geq Q$ and the progress at each call is on the order of $O(Q^2)$. A separating oracle may generate a separating hyperplane in the form of $-\left<\text{Exp}_y^{-1}x,g\right> > 0$ for any $x\in\mathcal{K}$, but the corresponding $Q$ could be arbitrarily small, even as low as $T^{-100}$. This could lead to a significant number of oracle calls. By computing an infeasible projection onto $\tilde{\mathcal{K}} = (1-\delta) \mathcal{K}$, we can achieve projection-free online learning with a more reasonable $O(T)$ oracle calls. This idea is briefly outlined in our draft (Lines 218-223).

---

> > ### Comment · Reviewer_2HSe · 2023-08-21
> >
> > This was helpful, thanks.

---

> > > ### Author Response · Authors · 2023-08-21
> > > **Response**
> > >
> > > You are welcome. Thank you for your positive evaluation of our work!

---

### Official Review · Reviewer_FXr6 · 2023-07-06

**Soundness:** 3 good
**Presentation:** 4 excellent
**Contribution:** 2 fair
**Rating:** 7
**Confidence:** 3

**Summary:**

The paper considers Riemannian online optimization problems over sets of constraints and tries to tackle them by avoiding projections onto the constraint set. This is an already established line of research in Euclidean optimization and the results follow the structure of Garber and Kretzu (2022). The results of the latter are adapted to the Riemannian setting using well-known geometric bounds.

**Strengths:**

The paper is concerned with problems of profound importance for the neurips community. It is well-written and can be followed easily by people with reasonable background in Riemannian optimization. The use of geometric bounds (law of cosines in negative curvature, spread of Jacobi fields) is clearly explained. The authors pay special attention in computability issues behind the separating and linear oracles used, building upon the contributions of Weber and Sra (2022b). The convergence guarantees much the ones of the Euclidean setting in Garber and Kretzu (2022) up to constants.

**Weaknesses:**

I am not thrilled by the level of originality since the paper follows closely the one by Garber and Kretzu (2022) and used geometric bounds that are well-known in the Riemannian optimization community for quite some time. Still, it is a clearly-written concrete contribution to a problem of interest, thus I think that publication in the conference is totally justifiable.

There are some points in the paper that I find peculiar as I specify in my questions' section. This is the main reason that I give a score only slightly above the acceptance threshold (which I will revisit in case the authors answer me concerns convincingly).

I would advise the authors to revisit the title of the paper. When I see "On ..." I automatically form the impression that the paper features some vague discussion about a research topic without really useful contributions, which is not the case here.

**Questions:**

1. I see some slight discrepancy in the selection of parameters and worst-case regret bounds between this paper and Garber and Kretzu (2022) (I will refer to that as GK from now on). Could you explain why this is happening? To be more specific:

Theorem 1 in this paper and Theorem 6 in GK do not have exactly the same constants. I think you choose $c=1/2$ and $c_1=4$, but I cannot see how the regret bound and the bound in the number of oracle calls become the same for $\zeta=1$.

The parameters $\eta$ and $\delta$ in Theorem 2 of this paper are not chosen the same way as the ones in Theorem 7 of GK with respect to $T$. This is quite worrying.

Theorem 3 in both papers have the same parameters (for $\zeta=1$), but the regret bound features slightly different constants.

Discrepancies like that seem not to be the same among Lemmas 1 and Theorems 4 between the papers.


2. I struggle to understand Algorithm 5, I think that after the if statement, you should have $< \exp^{-1}_{x_i}(y),\exp^{-1}_{x_i}(v_i) > \leq \epsilon$ or $d(x_i,v_i)^2 \leq 3 \epsilon$. Is that true?

3. As I see the results, it should pretty straightforward to have similar ones for manifolds of positive curvature (with $\zeta=1$ and $\bar r=r$). Is there any specific reason that you didn't attempt that?



**Limitations:**

The authors adequately discuss some limitations of their work in the conclusion of the paper.

---

> ### Author Rebuttal · Authors · 2023-08-09
>
> We appreciate your detailed and constructive feedback. Your specific questions and concerns have been addressed as follows.
> > I see some slight discrepancy in the selection of parameters and worst-case regret bounds between this paper and Garber and Kretzu (2022) (I will refer to that as GK from now on). Could you explain why this is happening?
>
> Thanks for the question on the discrepancy between our paper and Garber and Kretzu (2022). We  clarify these concerns item by item as follows.
>
> 1)  The discrepancy between our Theorem 1 and Theorem 6 of Garber & Kretzu (2022) might stem from a misunderstanding. Upon closer examination, we can observe that Theorem 6 of Garber & Kretzu omitted the dependence on $c_1$​, so its guarantee can be rewritten as:
> $$
> \text{Reg}\_T\leq\left(GRc+\frac{Gc_1r}{4}+\frac{4GR^2}{c_1r}\right)\sqrt{T}.
> $$
> In our work, the parameters correspond to $c=\frac{1}{2}$ and $c_1=\frac{4R}{r}$. By substituting these values into Theorem 6 of Garber & Kretzu (2022), we obtain:
> $$
> \text{Reg}\_T\leq\left(GRc+\frac{Gc_1r}{4}+\frac{4GR^2}{c_1r}\right)\sqrt{T}=\frac{5GR\sqrt{T}}{2}.
> $$
> This result matches the guarantee of our Theorem 1 when setting $\zeta=1$. Similarly, we can compute the number of oracle calls in Theorem 6 of Garber & Kretzu (2022):
> $$
> N_{calls}\leq \left(\frac{c_1R}{cr}+\frac{c_1^2}{4c^2}+1\right)T=\left(\frac{8R^2}{r^2}+\frac{16R^2}{r^2}+1\right)T.
> $$
> This is in agreement with our result, once again, when setting $\zeta=1$ and $\bar{r}=r$.
> 2) We appreciate your diligence in pointing out the apparent difference between our Theorem 2 and Theorem 7 of Garber & Kretzu (2022). This discrepancy was due to a typographical error. The correct parameters were used in the proof of Theorem 2 (Line 524). We will fix this in the revised version.
> 3) The discrepancy between our Theorem 3 and Theorem 3 of Garber & Kretzu (2022) is due to numerical rounding. The regret guarantee of Theorem 3 in our paper is
> $$
> \text{Reg}_T\leq GR\left(\left(\frac{5}{2}\zeta^2+\sqrt{180}\zeta+\frac{4}{\zeta}\right)T^{\frac{3}{4}}+20T^{\frac{1}{2}}\right).
> $$
> After choosing $\zeta=1$, we find $\frac{5}{2}\zeta^2+\sqrt{180}\zeta+\frac{4}{\zeta}=19.9164\approx 20$, which leads to the same result as Theorem 7 of Garber & Kretzu (2022):
> $$
> \text{Reg}_T\leq 20GRT^{\frac{1}{2}}+20GRT^{\frac{3}{4}}.
> $$
>
>
> > I struggle to understand Algorithm 5, I think that after the if statement, you should have $< \text{Exp}^{-1}_{x_i}(y),\text{Exp}^{-1}_{x_i}(v_i) > \leq \epsilon$ or $d(x_i,v_i)^2 \leq 3 \epsilon$. Is that true?
>
>  Thanks for pointing out this. Sorry there is a typographical error and this line verifies if $\left<\text{Exp}_{x_i}^{-1}y,\text{Exp}_{x_i}^{-1}v_i\right>\leq\epsilon$ or $d(x_i,y)^2\leq 3\epsilon$ holds. If $d(x_i,y)^2 \leq 3 \epsilon$ holds, then we have completed the necessary tasks because $x_i$ is sufficiently close to the target point $y$. And if $\left<\text{Exp}_{x_i}^{-1}y,\text{Exp}_{x_i}^{-1}v_i\right>\leq\epsilon$ holds, by the definition of $v_i$, we have
> $$
> \left<\text{Exp}_{x_i}^{-1}y,\text{Exp}_{x_i}^{-1}x\right>\leq \left<\text{Exp}_{x_i}^{-1}y,\text{Exp}_{x_i}^{-1}v_i\right>\leq \epsilon
> $$
> holds for any $x\in\mathcal{K}$. This relationship can be leveraged to construct a separating hyperplane between $y$ and $\mathcal{K}$, as outlined in Lines 542-546.
>
> > I would advise the authors to revisit the title of the paper.
> >
> Thanks for your thoughtful suggestion. We will be more than happy to remove the word "On" in the title.
>
> > As I see the results, it should pretty straightforward to have similar ones for manifolds of positive curvature (with $\zeta=1$ and $\bar r=r$). Is there any specific reason that you didn't attempt that?
>
> _Repeated from response to Reviewer 4T7s, Comment 2, with brief revisions._
>
> It is not too difficult to generalize the results in our paper to manifolds of positive curvature and even to CAT$(\kappa)$ spaces. To achieve this extension, the following adjustments would be necessary in our manuscript:
>
> 1.  We may assume that the sectional curvature lies in the range $[\kappa,K]$. In this context, we can replace Lemma 30 in our draft with Corollary 2.1 from Alimisis et al. (2020). When $K>0$, we must further assume that the diameter of the decision set is upper-bounded by $\frac{\pi}{\sqrt{K}}$​.
> 2.  Lemma 3 would need to be revised to recompute the Jacobi field, taking into account the sectional curvature of the manifold. Notably, in the case of manifolds with positive curvature, an initial computation provides $\bar{r}=O\left(\frac{\sin(\sqrt{K}(R+r))}{\sqrt{K}(R+r)}\right)\cdot r$, a relation that does not exhibit exponential dependence on $(R+r)$.
> 3.  The separation theorem, specifically for Hadamard manifolds as outlined in Silva Louzeiro et al. (2022), would require generalization to CAT$(\kappa)$ spaces. This change ensures that the separation oracle remains well-defined.
>
> We discovered that delving into all of these details did not offer additional insights and might have actually diminished the readability of the text. Therefore, we chose not to include them in the current version. However, we recognize the importance of this matter and will provide further discussions on this subject in the revised version.

---

> > ### Comment · Reviewer_FXr6 · 2023-08-12
> >
> > I thank the authors for their clarifications. I am satisfied by them and increase my score by one point.

---

> > > ### Author Response · Authors · 2023-08-12
> > > **Response**
> > >
> > > Thank you so much for the helpful feedback and positive evaluation!

---

### Official Review · Reviewer_4T7s · 2023-07-07

**Soundness:** 3 good
**Presentation:** 3 good
**Contribution:** 2 fair
**Rating:** 7
**Confidence:** 4

**Summary:**

The paper focuses on constrained Riemannian online optimization on the Hadamard manifold. Existing Riemannian online optimization methods often require projections, which present computational complexity challenges in high-dimensional settings. To address this issue, the authors have developed a projection-free Riemannian online optimization structure, implemented under two scenarios - the separation oracle and the linear optimization oracle.

Initially, they explore the Infeasible R-OGD and Infeasible Riemannian bandit algorithm under a separation oracle, for which the regret bounds are proved to be $\mathcal O(\sqrt T)$ and $\mathcal O(T^{\frac{3}{4}})$ respectively for geodesically convex functions. Furthermore, they consider the Block R-OGD under a linear optimization oracle, and provide a proof for regret bounds of $\mathcal O(T^{\frac{3}{4}})$ and $\mathcal O(T^\frac{2}{3})$ for geodesically convex and geodesically strongly convex functions respectively. It's noteworthy that all these regret bounds match their respective Euclidean counterparts.

**Strengths:**

-  Originality & Significance: Althogh the paper has retricted novelty (discuss in the following), it provides, to my best of knowledge.  the first no-regret guarantee for projection-free Riemannian OCO. Also, it is nice to see that all regret bounds match their Euclidean counterparts.

- Clarity & Significance: The paper is well-organized, techinally sound and easy to follow. The assumption is standard in the literature of Riemannian optimization and Riemannian OCO.

**Weaknesses:**

- Retricted novelty: The majority of the analysis leans heavily on the Euclidean Analogue and Jacobian/Hessian comparison. Although it is the standard structure in Riemannian optimization literature, it would have been enriching to see some novel conceptual ideas that engage more specifically with the geometry of the problem.

- Positive curvature: The study focuses on the Hadamard manifold, known for its non-positive sectional curvature. Conversely, in practical scenarios, there's considerable work undertaken within spaces possessing positive sectional curvature like SO(3). The methodology and algorithms presented in this paper appear, at first glance, to be easily extendable to accommodate $cat(K)$ spaces. However, the authors have not provided any explanation or context for why such positive curvature spaces were not considered in their study.  A consideration of such spaces could have added more depth and wider applicability to the research.

- Possible typo: formula about $\bar r $ after l227: there should be a additional $r$ at RHS.
 Algorithm 5: $y$ seems to be a point rather than a tangent vector.


**Questions:**

- I'd appreciate if the authors could expound on their primary insights in this paper. Specifically, it would be beneficial to understand the significant challenges encountered while extending the Euclidean projection-free algorithm to the Riemannian manifold.

- Additionally, I'm interested in knowing whether the proposed Riemannian projection-free method could be easily extended to $cat(K)$ spaces. If this is not feasible, could the authors shed light on the primary hurdles that prevent such an extension?

---

> ### Author Rebuttal · Authors · 2023-08-09
>
> Thank you for your thoughtful and helpful feedback. We will revise typographical errors accordingly. We have addressed your specific questions and concerns below.
> > I'd appreciate if the authors could expound on their primary insights in this paper. Specifically, it would be beneficial to understand the significant challenges encountered while extending the Euclidean projection-free algorithm to the Riemannian manifold.
>
> _Repeated from response to Reviewer xiK4, Comment 4, with brief revisions._
>
> Thank you for the question. Our work builds heavily on Garber & Kretzu (2022) and Wang et al. (2021, 2023), but we have several novel contributions:
>
> 1.  We identify that the shrinking set $(1-\delta)\mathcal{K}$ is non-convex and subsequently extend the definition of infeasible projection initially presented by Garber & Kretzu (2022) to accommodate non-convex settings.
> 2.  Quantifying the progress made with each call to the separation oracle presents a highly non-trivial challenge. We overcome this in Lemma 3, where we amalgamate Lemma 19 (equivalent to Lemma 45 in Wang et al. (2023)) with meticulous computation involving Jacobi field comparisons.
> 3.  Concerning the linear optimization oracle, deriving a separating hyperplane using Riemannian Frank-Wolfe stands as the principal hurdle. We manage to overcome this through an innovative application of Riemannian cosine laws. For further insights, we refer readers to Remark 2 and the proof of Lemma 5 in our manuscript.
>
> > Additionally, I'm interested in knowing whether the proposed Riemannian projection-free method could be easily extended to CAT$(\kappa)$ spaces. If this is not feasible, could the authors shed light on the primary hurdles that prevent such an extension?
>
> It is not too difficult to generalize the results in our paper to CAT$(\kappa)$ spaces. To achieve this extension, the following adjustments would be necessary in our manuscript:
>
> 1.  We may assume that the sectional curvature lies in the range $[\kappa,K]$. In this context, we can replace Lemma 30 in our draft with Corollary 2.1 from Alimisis et al. (2020). When $K>0$, we must further assume that the diameter of the decision set is upper-bounded by $\frac{\pi}{\sqrt{K}}$​.
> 2.  Lemma 3 would need to be revised to recompute the Jacobi field, taking into account the sectional curvature of the manifold. Notably, in the case of manifolds with positive curvature, an initial computation provides $\bar{r}=O\left(\frac{\sin(\sqrt{K}(R+r))}{\sqrt{K}(R+r)}\right)\cdot r$, a relation that does not exhibit exponential dependence on $(R+r)$.
> 3.  The separation theorem, specifically for Hadamard manifolds as outlined in Silva Louzeiro et al. (2022), would require generalization to CAT$(\kappa)$ spaces. This change ensures that the separation oracle remains well-defined.
>
> We discovered that delving into all of these details did not offer additional insights and might have actually diminished the readability of the text. Therefore, we chose not to include them in the current version. However, we recognize the importance of this matter and will provide further discussions on this subject in the revised version.

---

> > ### Comment · Reviewer_4T7s · 2023-08-17
> >
> > I would like to express my appreciation for the detailed clarification provided by the author, especially in terms of the technical novelty and the extension to the CAT($\kappa$) space. This has significantly enhanced my understanding of the work presented.
> >
> > In light of this, I will raise the score by one point, moving from 6 to 7.

---

> > > ### Author Response · Authors · 2023-08-17
> > > **Response**
> > >
> > > Thanks for taking the time to provide helpful comments on the paper and for the upward revision of the evaluation score!

---

### Official Review · Reviewer_puFc · 2023-07-26

**Soundness:** 2 fair
**Presentation:** 3 good
**Contribution:** 2 fair
**Rating:** 7
**Confidence:** 5

**Summary:**

The authors study online learning on homogeneous Hadamard manifolds with geodesically-convex, Lipschitz losses. They focus on projection-free methods and in particular they develop algorithms that use either a separaton oracle or a linear optimization oracle. They study adaptive regret algorithms for the full information and the bandit feedback settings.


**Strengths:**

Besides from the strengths of having the theoretical results authors have, I want to emphasize the following.

This work usually states clearly the assumptions that is making, even though this could be improved, as I suggested. It was good to see the authors taking into account the non convexity of (1-\delta)\mathcal{K}. Some works ignore this fact, I wonder if they learn about this in the review of a paper. But the authors here are rigorous and work with the possible non-convex set.

The results with linear minimization oracle look fine to me

**Weaknesses:**

# Main points of my review

Several results in this paper follow closely Garber and Kretzu (2022). This is not a criticism, of course there are several challenges in the Hadamard case that need to be surmounted.

I have several points of criticism though:

+ The paper contains some mistakes, and I am not sure if one could fix at least one of them while keeping the stated results (see below).
+ Moreover algorithm 4 has a regret guarantee in Theorem 2 which is non standard (algorithm plays a point and feedback is given for another point but the regret is still measured with respect to the played point). This seems an artifact of the proof, as in it was the notion that made the proof work, but without any other justification for this made-up definition of regret, usefulness of it or an application of it, the result is very weak.
+ Lemmas 3 and 4 yield that the number of iterations of algorithm 2 is exponential on R. This is not desirable. I do not want to criticize this point harshly, since Hadamard is hard and sometimes the geometry makes these exponential dependences appear, and this could be a result deserving its own paper. This dependence also appears in previous Riemannian online learning works. But rather, I just want to encourage the authors, possibly for future work, to try to improve this to a polynomial dependence (see below for a few comments on this).

While I think this works has value, I also think this is not ready for publication. At the very least the errors should be corrected, and the bandit results should not be presented as results (e.g. in the table) unless the authors can justify that the modify notion or regret makes sense. And in any case this should be clearly stated for the readers that only want to firstly check the introduction out.

# Proof Errors

Correct me if I am wrong but I think I found the following errors in your proofs. I can increase my score if the technical errors can indeed be fixed.

+ The proof of Lemma 1 is wrong, although fixable. The points y_{t+1} are not guaranteed to be in B_p(R) and so you cannot use lemma 29 as you stated it. However, if you look closely at Lemma 5 of Zhang and Sra, you will realize that the lemma actually says that \zeta depends on \kappa and c, which a stronger inequality than what you stated (dependence on \kappa and D). Using that stronger statement then you can show in your Lemma 1 that the \zeta in the proof depends on d(\tilde{y}_t, x) \leq D, and so the proof follows.

+ However, your lemma 2 has the same problem, and when you use it in the proof of lemma 4, you use it with the point y_i, which as you point out in L435, it is y_i \not \in \mathcal{K}. Therefore, the geometric deformation is not \zeta (which was defined as depending on D), and this does not seem to be so easily fixable.


# Other comments / suggestions

The results with linear minimization oracle look fine to me.

You should specify in lemma 5, that you always use y\in B_p(R), since you use f(x) = d(x y)^2/2 and you claim that it is \zeta smooth, which you can have because when you use lemma 5 in alg 6 it is always y\in B_p(R). But this is not stated and it was confusing for me.

This work usually states clearly the assumptions that is making, even though this could be improved, as I suggested. It was good to see the authors taking into account the non convexity of (1-\delta)\mathcal{K}. Some works ignore this fact, I wonder if they learn about this in the review of a paper. But the authors here are rigorous and work with the possible non-convex set.

"(pinpointing a compact set that contains the true optimum can improve time complexity) This makes metric projection onto a subset of Riemannian manifold seemingly indispensable, yet this operation is not only computationally taxing but can also lead to unwanted geometric distortion, which can undermine convergence. Certain works have depended on simplifying assumptions to prove results..." I disagree with the message that is given here. For several reasons:
    + Firstly, in many situations, one can "pinpoint" a compact set where the optimum is by just estimating the initial distance to minimizer and imposing ball constraints (and using a doubling trick or other tricks if necessary to avoid assuming knowledge of this distance ). In that case, the operation is "projection-free" according to your own definition in line 178, due to the simplicity of the operation, so not necessarily computationally taxing.
    + Secondly, while it is true that using projections can lead to greater geometric distortion that can undermine convergence (e.g. see https://arxiv.org/pdf/2305.16186.pdf where the authors make projected gradient descent work for smooth functions while quantifying the distortion), several works make use of projections to obtain algorithms that previously had to make the assumption of iterates staying in some set, and at the same time the geometric dependence on the convergence rates does not worsen. See for instance https://arxiv.org/pdf/2211.14645.pdf and theorem 7 in https://arxiv.org/pdf/2305.16186.pdf, and https://arxiv.org/pdf/2111.13263v2.pdf section 6
    + Thirdly, the works that mandate the iterates to be in a feasible set by assumption do so because they do not know how to enforce or guarantee their algorithms are in some feasible set, even if they have access to projections. It is not like the assumption is made to avoid projections that are "computationally taxing and with extra geometric distortions, which can undermine convergence" by using projections. Their techniques do not allow them to use projections and as explained above and more clever algorithms do without extra distortions.


You probably cannot compute \sigma_i in Algorithm 5 in close form. You probably would need a binary search and then have the algorithm be able to account for an error. Is that right?

# Exponential dependence on the diameter

Regarding the exponential nature of the constants in lemma 3 and running time in lemma 4, this phenomenon is very similar to the exponential constants that appear in the deformations of lemma 2 in https://arxiv.org/pdf/2012.03618.pdf However, in that paper they are interested on optimization, and they show that one can get away with this exponential complexity. The trick is a reduction from global optimization to approximately implementing a ball optimization oracle, so that the algorithms would only need to run in balls where R\sqrt{-\kappa} = O(1). The reduction goes as follows: optimize in one such constant-radius ball with linear rates (regularize as it is done in reductions if necessary, although you'd need smoothness of the losses) and then optimize in another ball with center equal to the previously computed point. In the online learning setting, it is harder but you could in principle try to run Riemannian OGC by minimizing the sort of FTRL objective you have by using this reduction (you'd have to optimize with linear rates in the intersection of constant curvature balls and your sets, but every time the infeasible projection would be easy). Or just implement the separation oracle by looking at making progress in constant-curvature balls. Anyway, maybe it doesn't work for you, but I hope it is useful to you to at least know that a problem of this kind has been sort of solved in Riemannian optimization.

# Minor suggestions / typos

L62 "While a horosphere gsc-convex" ->""While a horosphere is gsc-convex"
L200 "theLipschitz" -> "the Lipschitz"
L259 Alg 5. I would not say target vector y, since y is a point.
L269 Alg 6. "initial vector y" -> "initial vector y_i" (although again, I would not say initial *vector* )

# Edit after rebuttal
The authors provided fixes for the proof errors and a new algorithm two work with two point bandit feedback. I increased my score from reject to accept (from 3 to 7)


**Questions:**

see above

**Limitations:**

see above

---

> ### Author Rebuttal · Authors · 2023-08-09
>
> Your insightful comments and constructive criticism are highly valued. We've responded to your specific questions and concerns in the following sections.
> >The paper contains some mistakes, and I am not sure if one could fix at least one of them while keeping the stated results.
>
> Thank you for bringing this matter to our attention. We have indeed found that the geometric distortion was underestimated, but we have identified a quick fix based on Lemma 5 from Zhang & Sra (2016):
> $$
> a^2 \leq \zeta(\kappa,c) b^2+c^2-2bc\cos A.
> $$
>
> First, we amend Definition 1 to express $\zeta$ as $\zeta \coloneqq \zeta(\kappa,2R)$.
>
> For the proof of Lemma 1, it suffices to ensure that the inequality $d(\tilde{y}_t,x)\leq 2R$ holds for any $\tilde{y}_t\in B_p(R)$ and $x\in\tilde{\mathcal{K}}$. Because $\tilde{\mathcal{K}}\subseteq \mathcal{K}\subseteq B_p(R){}$, by the triangle inequality:
> $$
> d(\tilde{y}_t,x)\leq d(\tilde{y}_t,p)+d(p,x)\leq 2R.
> $$
>
> In the proof of Lemma 4, we have utilized Lemma 2. To validate that $\zeta$ describes the correct geometric distortion, we need to prove that $d(y_i,z)\leq 2R$ holds for any $i\geq 1$ and $z\in(1-\delta)\mathcal{K}$. This can be achieved through induction. The case for $i=1$ is easy. As $y_1\in B_p(R)$, $z\in(1-\delta)\mathcal{K}\subseteq B_p(R)$, the following relationship holds:
> $$
> d(y_1,z)\leq d(y_1,p)+d(z,p)\leq 2R.
> $$
> Now assume $d(y_i,z)\leq 2R$ holds for some $i\geq 1$ and any $z\in(1-\delta)\mathcal{K}$, then $\zeta$ is a valid geometric distortion. By Lemma 5 of Zhang \& Sra (2016) and our Lemma 2,
>
> $$
> d(y_{i+1},z)^2\leq \zeta d(y_{i+1},y_i)^2+d(y_i,z)^2-2\left<\text{Exp}_{y_i}^{-1}y_{i+1},\text{Exp}_{y_i}^{-1}z\right>  \leq d(y_i,z)^2-\frac{\delta^2\bar{r}^2}{\zeta}\leq (2R)^2.
> $$
>
> Thus $d(y_{i},z)\leq 2R$ holds for any $i\geq 1$ and $z\in(1-\delta)\mathcal{K}$, and the mistake was fixed.
>
> > Moreover algorithm 4 has a regret guarantee in Theorem 2 which is non standard $\dots$
>
> Thank you for your keen observations. We employed this non-standard setting to circumvent a fundamental difficulty on Hadamard manifolds, where $(1-\delta)\mathcal{K}$ is non-convex. During the rebuttal phase, we discovered a method to eliminate this drawback and achieve $O(\sqrt{T})$  regret by using an SO for $\mathcal{K}$ within the two-point feedback setting. The revised algorithm is outlined below:
>
> > Algorithm: Riemannian Projection-free BCO with Two-point Feedback
> > Parameters: $\beta, \delta, \delta'$; $\delta\in(0,1)$, $\beta\in(0,1)$, $\delta'=(1-\beta)\frac{\sqrt{-\kappa}(R+r)}{\sinh(\sqrt{-\kappa}(R+r))}\cdot r$.
> > Initialize  $x_1\in\beta\mathcal{K}$, $y_1=\text{Exp}_p(\frac{\text{Exp}_p^{-1}x_1}{\beta})\quad$   //$y_1\in\mathcal{K}$
> >For $t=1,\dots,T$:
> >
> >> Sample $z_t\sim\mathbb{S}_{x_t}(\delta')$
> >>
> >> Play $z_t$ and its antipodal point
> >>
> >> Observe $f_t$ at $z_t$ and the antipodal point
> >>
> >>Construct $g_t$ by the estimator in Algorithm 3 of  Wang et al. (2023)
> >>
> >>$y_{t+1}'=\text{Exp}\_{y_t}\left( -\eta \frac{S\_{\delta'}}{V_{\delta'}} \Gamma_{x_t}^{y_t}g_t \right)$
> >>
> >>$y_{t+1}\leftarrow$Output of Algorithm 2 with $\mathcal{K},r,\delta$ and $y_{t+1}'\quad$ //$y_{t+1}\in\mathcal{K}$
> >>
> >>$x_{t+1}=\text{Exp}_p(\beta \text{Exp}\_p^{-1}y\_{t+1})\quad$ // $x_t\in\beta\mathcal{K}$
> >
> >End For
>
> Proof Sketch: following (15), (18) in our draft, we have
> $$
> E[\hat{f}\_t(x_t)-\hat{f}\_t(x)]\leq \frac{S_{\delta'}}{V_{\delta'}}E[\left< g_t,-\Gamma_{y_t}^{x_t}\text{Exp}^{-1}_{y_t}x\right>+\left< g_t,\Gamma_{y_t}^{x_t}\text{Exp}^{-1}_{y_t}x-\text{Exp}^{-1}_{x_t}x\right>]+2\delta'\rho G.
> $$
>
> For the two-point feedback model, by Lemma 17 of Wang et al. (2023), $\frac{S_{\delta'}}{V_{\delta'}}E[\\|g_t\\|]=O(1)$. Thus,
> $$
> \frac{S_{\delta'}}{V_{\delta'}}E[\left< g_t,\Gamma_{y_t}^{x_t}\text{Exp}^{-1}\_{y_t}x-\text{Exp}^{-1}_{x_t}x\right>]\leq\frac{S\_{\delta'}}{V\_{\delta'}}E[\\|g_t\\|]\cdot \zeta d(x_t,y_t)=O(1)\cdot \zeta(1-\beta)d(y_t,p)=O(\delta'),
> $$
>
> where we use the $\zeta$ smoothness of $\frac{1}{2}d(x,y)^2$ and $1-\beta=O(\delta')$.
> Since $y_{t+1}$ is an infeasible projection of $y_{t+1}'$ onto $(1-\delta)\mathcal{K}$,
> $$
> d(y_{t+1},x)^2\leq d(y_{t+1}',x)^2\leq d(y_t,x)^2+\zeta\eta^2\frac{S_{\delta'}^2}{V_{\delta'}^2}\\|g_t\\|^2-2\eta\frac{S_{\delta'}}{V_{\delta'}}\left< g_t,-\Gamma_{y_t}^{x_t}\text{Exp}^{-1}_{y_t}x\right>
> $$
> holds for any $x\in(1-\delta)\mathcal{K}$. Combining the above inequalities, we have
> $$
> E[\hat{f}_t(x_t)-\hat{f}\_t(x)]\leq\frac{d(y_t,x)^2-d(y\_{t+1},x)^2}{2\eta}+O(\eta)+O(\delta').
> $$
> holds for any $x\in(1-\delta)\mathcal{K}$. Also, following (20), (21) and taking $x=\text{Exp}_p((1-\delta)\text{Exp}^{-1}_p x^*)$, we have $E[f_t(z_t)-\hat{f}_t(x_t)]=O(\delta')$ and $E[\hat{f}_t(x)-f_t(x^*)]=O(\delta+\delta')$.
>
> In sum,
> $$
> \sum_{t=1}^TE[f_t(z_t)-f_t(x^*)]=O\left(\frac{1}{\eta}+(\eta+\delta+\delta')T\right).
> $$
> Choosing $\eta,\delta,\delta'=O(\frac{1}{\sqrt{T}})$, we get $O(\sqrt{T})$ regret.
>
> Your comments or suggestions on this idea would be greatly appreciated. We are more than willing to replace the current BCO result with this revised version, as it resolves the problem related to the non-standard setting.
>
> > Lemmas 3 and 4 yield that the number of iterations of algorithm 2 is exponential on R $\dots$
>
>  Thanks for your detailed suggestions. We will consider this interesting problem in the near future.
>
> > You should specify in lemma 5, that you always use $y\in B_p(R)$. You probably cannot compute $\sigma_i$ in Algorithm 5 in close form.
>
> Thank you for your insightful suggestion. You are indeed correct, and we appreciate your pointing this out. We will provide more detailed discussions in our revised version.
>
> > "(pinpointing a compact set that contains the true optimum can improve time complexity) $\dots$ without extra distortions.
>
> Thank you for sharing your insightful understanding of Riemannian metric projection. We will make sure to carefully revise these sentences in the updated version of our work.

---

> > ### Comment · Reviewer_puFc · 2023-08-17
> > **reply**
> >
> > Thanks for the reply. Indeed that fixes the proof.
> >
> > I would not call your new bandit algorithm "a method to eliminate this drawback" since the new result is different. It works in a different model (and as a consequence, you can get a \sqrt{T} bound instead of T^{3/4}). In any case, that result is indeed interesting and the paper has several other contributions that are interesting. Please add the new result in full detail to the paper and keep the other one. You can emphasize that the other one does not work with a standard notion of regret.
> >
> > I am increasing my score from reject to accept (3 to 7).

---

> > > ### Author Response · Authors · 2023-08-17
> > > **Response**
> > >
> > > Thank you for your valuable feedback on the paper and for increasing the evaluation score! We will carefully revise the paper based on your comments.

---

### Official Review · Reviewer_xiK4 · 2023-07-30

**Soundness:** 3 good
**Presentation:** 2 fair
**Contribution:** 3 good
**Rating:** 5
**Confidence:** 2

**Summary:**

The authors propose a generalization for online learning on the Riemannian manifolds via separation and linear optimization oracles. The core idea here is to use the oracles to construct an infeasible projection, which may not be the nearest point in the constrain/decision set $\mathcal{K}$, but are in $\mathcal{K}$ and is closer to a shrunk feasible set $(1-\delta)\mathcal{K}$ than the original point. The interesting point finding here is that because there is a buffer between $\mathcal{K}$ and $(1-\delta)\mathcal{K}$, we have a constant distance decrease of $O(\delta^2)$ (Lemma 4 eq (5) in the appendix) w.r.t. by the separating oracle direction to the shrunk set $(1-\delta)\mathcal{K}$, and hence the projection to $\mathcal{K}$ can be done in constant time to $\delta$. This means we can construct an infeasible/inaccurate projection by the separation oracle in $O(1/\delta^2)$ time, and the algorithm follows from replacing the projection part with the infeasible projection and bound the errors. The authors also consider the linear optimization oracles, and it's done by constructing the separation oracles using the linear optimization oracles in Frank-Wolfe. Actually, this draft is a combination between Garber & Kretzu (2022) and Wang et al. (2021), where the authors generalize the online learning by SO/LOO framework by Garber & Kretzu (2022) with the tools and assumptions in Wang et al. (2021).

**Strengths:**

The idea of constructing an infeasible projection (the solution is feasible to $\mathcal{K}$) and providing guarantees to a shrunk set $(1-\delta)\mathcal{K}$ is very interesting, and it's good to see it works on the Riemannian manifold. The introduction is pretty well-written, and there is no major error in the proof, and generalizing the result from Euclidean space to the Riemannian space may be challenging in some corner cases.

**Weaknesses:**

The major weakness of the paper is its complexity, the lack of experiments, and the limiting assumptions.

First, the proof in the paper is pretty complex and is not self-contained. To verify any proof in the paper, the readers usually need to go to Garber & Kretzu (2022) and Wang et al. (2021), which have another set of symbols and may again point to other papers. And this may lead to confusion for both the authors and the readers. For example, the Lemma 19 in the draft points to Wang et al. 2023 Lemma 45, which requires the curvature to be also upper-bounded, but such assumption and coefficient disappears in the draft. Garber & Kretzu (2022) assume that the feasible set contains the origin. The paper generalized it to arbitrary point $p$, but in that case, the shrinking set $\mathcal{K}$ should be redefined toward $p$ for $(1-\delta)\mathcal{K}\subset\mathcal{K}$, but it is also not done. The proof in the paper is mostly correct when self-contained, but it is very time-consuming to verify the mentioned errors from cited lemmas by chasing around the links, so I am not very sure about the correctness of the theoretical paper.

Second, I am not very sure about the practical aspect of the paper due to its limiting assumptions and lack of experiments. Surely, theoretical papers may not have been experimented with, but the paper is too complex to verify its correctness, so it would be good to show the reader that it works in a sense. I am concerned about the limitation in the assumptions because most of the examples mentioned in the draft's introduction don't fit the assumption, e.g., the spherical constraint in K-means clustering. The manifold must be bounded, containing a non-empty unit ball, and contained within a unit ball of radius $\leq \pi/\sqrt{4\kappa_2}$ (Lemma 19). And I am unsure when the separation oracle and linear optimization oracle are cheaper than the projection in a practical example. Please give the readers more examples to understand when the setting would be suitable.

Finally, the paper is a direct combination between Garber & Kretzu (2022) and Wang et al. (2021). Most of the work here looks like a Riemannian generalization of Garber & Kretzu's (2022)'s framework with Wang et al. (2021,2023)'s tools, replacing the distance with exponential maps and bound them with Wang et al. (2021,2023)'s inequalities. And there are not many new surprising results, but double the complexity. It is not necessary to be novel to publish, but given that I'm not convinced that the results are totally correct, I would recommend a weak rejection. This draft should be more suited to a journal than a conference.

**Questions:**

1. L32: Computing the orthogonal projection of "polytope" to half-spaces is NP-hard, but point projection is not. So this is unrelated to the draft.
2. L161: May you explain the existence of g in Lemma 18 implies an efficient implementation?
3. Assumptions & Lemma 19 requires $R\leq \pi/\sqrt{4\kappa_2}$, but it's not in the assumptions and coefficients.
4. Lemma 29 is different from the original paper. Please note that you use the monotonicity of $x coth(x)$ here.
5. L179: Please cite the projection algorithm in $O(\log(1/\epsilon)$
6. Algorithm 2 takes $\delta$ as an input but the $\delta$ doesn't appears in the algorithm. Please at least define $\gamma$ as a function.
7. Appendix L430 eq (3): Please annotate that the later line suffices the inequality.
8. Algorithm 2: It's also using one membership oracle at each step.
9. Appendix L456: You use the bounded gradient property, not the Lipschitzness of the gradient.
10. Algorithm 4: This looks pretty much like the R-BAN. Why R-BAN sample from a sphere but the algorithm samples from a ball?

---

> ### Author Rebuttal · Authors · 2023-08-09
>
> Thank you for your constructive comments and suggestions. We appreciate the time and effort spent in reviewing our work, and we will revise the draft accordingly. Below, we address your specific questions and concerns:
>
> > For example, the Lemma 19 in the draft points to Wang et al. 2023 Lemma 45, which requires the curvature to be also upper-bounded, but such assumption and coefficient disappears in the draft. Garber & Kretzu (2022) assume that the feasible set contains the origin. The paper generalized it to arbitrary point $p$, but in that case, the shrinking set $\mathcal{K}$ should be redefined toward $p$ for $(1-\delta)\mathcal{K}\subset \mathcal{K}$, but it is also not done.
>
> This might be a misunderstanding. As you pointed out, Lemma 19 requires the sectional curvature to be both lower and upper bounded. We assumed the manifold to be Hadamard and defined the curvature lower bound in Assumption 1, and the sectional curvature of Hadamard manifolds is upper bounded by $0$ (Lines 149-151), so we did not overlook this point. Additionally, we defined the shrinking set  $(1-\delta)\mathcal{K}$ for some fixed $p$ in Definition 2.
>
> > The proof of the paper is pretty complex and is not self-contained.
>
> We acknowledge that our proof is somewhat complex due to the Jacobi field technique. To make the paper more accessible, we have introduced related background in Appendix E.1. We plan to further improve readability in the revised version and welcome any comments or suggestions.
>
> > I am not very sure about the practical aspect of the paper due to its limiting assumptions and lack of experiments. $\dots$ Please give the readers more examples to understand when the setting would be suitable.
>
> We mainly consider Hadamard manifolds, a standard assumption (Zhang and Sra, 2016; Wang et al., 2021). While more restrictive than general Riemannian manifolds, practical examples exist, such as the geometric mean and the Bures-Wasserstein barycenter on the manifold of SPD matrices (Line 165). In these two specific examples, Weber & Sra (2022b) demonstrate that, WLOG, the feasible set can be assumed to be $\mathcal{K}=\\{X|L\preceq X\preceq U\\}$, where $L$ and $U$ are SPD matrices. Weber & Sra (2022b) establish that the linear optimization oracle for $\mathcal{K}$ admits a closed-form solution. We show a separation oracle for $\mathcal{K}$ can also be efficiently implemented. Under the affine-linear metric, we have $\left<A,B\right>_Y=\text{tr}(Y^{-1}AY^{-1}B)$ and $\text{Exp}_Y^{-1}X=Y^{\frac{1}{2}}\log(Y^{-\frac{1}{2}}XY^{-\frac{1}{2}})Y^{\frac{1}{2}}$. By some computation, for any $X\in\mathcal{K}$, we have:
> $$\left< -\text{Exp}_Y^{-1}X,Y \right>_Y>0\text{ when }Y\succ U$$
> and
> $$\left< -\text{Exp}_Y^{-1}X,-Y \right>_Y>0\text{ when }Y\prec L$$
> Thus both optimization oracles can be efficiently computed. We agree that numerical experiments would enhance our work and will include some experiments on the SPD manifold in the revised version.
>
> > The paper is a direct combination between Garber \& Kretzu (2022) and Wang et al. (2021).
>
> Thank you for your comment. We acknowledge the similarities between our work and the works of Garber & Kretzu (2022) and Wang et al. (2021, 2023), but it also features several significant and novel contributions:
>
> 1.  We identify that the shrinking set $(1-\delta)\mathcal{K}$ is non-convex and subsequently extend the definition of infeasible projection initially presented by Garber & Kretzu (2022) to accommodate non-convex settings.
> 2.  Quantifying the progress made with each call to the separation oracle presents a highly non-trivial challenge. We overcome this in Lemma 3, where we amalgamate Lemma 19 (equivalent to Lemma 45 in Wang et al. (2023)) with meticulous computation involving Jacobi field comparisons.
> 3.  Concerning the linear optimization oracle, deriving a separating hyperplane using Riemannian Frank-Wolfe stands as the principal hurdle. We manage to overcome this through an innovative application of Riemannian cosine laws. For further insights, we refer readers to Remark 2 and the proof of Lemma 5 in our manuscript.
>
> > L161: May you explain the existence of g in Lemma 18 implies an efficient implementation?
>
> We consider a gsc-convex set $\mathcal{K}=\\{x|\max_{1\leq i\leq m}{h_i(x)}\leq 0\\}$ where each $h_i(x)$ is gsc-convex. Given a point $y\notin\mathcal{K}$, we can check the sign of $h_i(y)$ for $i=1,\dots,m$ until we find $i^*$ such that $h_{i^*}(y)>0$, then by Lemma 18, we have
>
> $$
> -\left<\operatorname{Exp}_{y}^{-1}x,\nabla h_{i^*}(y)\right> >0
> $$
>
> holds for any $x\in\mathcal{K}$ and thus we get a separation oracle between $y$ and $\mathcal{K}$. Computing this separation oracle requires at most $m$ function value evaluations and $1$ gradient evaluation, which can be easily implemented when $m$ is moderate.
>
> > Assumptions \& Lemma 19 requires $R\leq \pi/\sqrt{4\kappa_2}$, but it's not in the assumptions and coefficients.
>
> In Assumption 1, we assumed $\mathcal{M}$ is Hadamard, so $\kappa_2=0$. In this case, we do not need to assume the manifold is bounded because for any two different points, there exists a global length-minimizing geodesic connecting them.
>
> > L179: Please cite the projection algorithm in $O(\log(1/\epsilon))$.
>
> Let $B_p(r)$ be a geodesic ball with center $p$ and radius $r$ and $x$ be a point outside of $B_p(r)$, and we intend to compute the projection of $x$ onto $B_p(r)$. We first compute the geodesic connecting $x$ and $p$, say, $\gamma(t)$, which intersects the sphere at $z$. Then by Gauss’s Lemma (Theorem 6.8, Lee 2006), the tangent vector of the geodesic $\gamma(t)$ at $z$ is perpendicular to the geodesic sphere $S_p(r)$, which means $z$ is the projection of $x$ onto $B_p(r)$. So it suffices to do $O(\log(1/\epsilon))$ binary search on the geodesic $\gamma(t)$ to get an approximation of $z$ up to $\epsilon$ precision. For the $t$-th binary search, we merely need to verify whether $d(x_t,p)\geq r$ is true, which is computationally efficient.

---

> > ### Comment · Reviewer_xiK4 · 2023-08-16
> >
> > > Definition of shrinking set
> >
> > Sorry to skipped the definition and the non-positive curvature of the Hadamard space. For clarity, I think it would be better to put the later near the assumptions.
> >
> > After reading the authors' rebuttal, I decide to bump my score. However, I am still not confident about the correctness of the draft because it is not self-contained and think the draft is more suitable for a journal than a conference.

---

> > > ### Author Response · Authors · 2023-08-16
> > > **Response**
> > >
> > > Thank you for your constructive feedback and for adjusting the evaluation score upwards! We will clarify the non-positive curvature characteristic of Hadamard manifolds near the assumptions.

---

### Decision · Program_Chairs · 2023-09-21

**Decision:**

Accept (poster)

**Comment:**

I thank the reviewers for their comments. I took the discussion as well as the author feedback into consideration. there seems to be general consensus among the reviewers that the paper is interesting (now that the error has been fixed). I am a bit unhappy over the rushed presentation and the generally low quality of the paper. i will recommend acceptance - this is an optimistic descent step - and I strongly urge the authors to revise their work thoroughly for the camera-ready version, taking the comments of the reviewers into account.